# ON THE HARDNESS OF LEARNING UNDER SYMMETRIES

**Bobak T. Kiani**[†12]**, Thien Le**[†1]**, Hannah Lawrence**[†1]**, Stefanie Jegelka**[13]**, Melanie Weber**[2]
[1] MIT EECS, [2] Harvard SEAS, [3] TU Munich, [†] co-first author

## ABSTRACT

We study the problem of learning equivariant neural networks via gradient descent. The incorporation of known symmetries ("equivariance") into neural networks has empirically improved the performance of learning pipelines, in domains ranging from biology to computer vision. However, a rich yet separate line of learning theoretic research has demonstrated that actually learning shallow, fully-connected (i.e. non-symmetric) networks has exponential complexity in the correlational statistical query (CSQ) model, a framework encompassing gradient descent. In this work, we ask: are known problem symmetries sufficient to alleviate the fundamental hardness of learning neural nets with gradient descent? We answer this question in the negative. In particular, we give lower bounds for shallow graph neural networks, convolutional networks, invariant polynomials, and frame-averaged networks for permutation subgroups, which all scale either superpolynomially or exponentially in the relevant input dimension. Therefore, in spite of the significant inductive bias imparted via symmetry, actually learning the complete classes of functions represented by equivariant neural networks via gradient descent remains hard.

## 1 INTRODUCTION

In recent years, the purview of machine learning has expanded to non-traditional domains with geometric inputs, from graphs to sets to point clouds. Correspondingly, it is now common practice to tailor neural architectures to the symmetries of the input – for example, graph neural networks are invariant to permutations of the input nodes, while networks operating on molecules and point clouds are invariant to permutation, translation, and rotation. Empirically, encoding such structure has led to computational benefits in applications (Wang et al., 2021; Batzner et al., 2022; Bronstein et al., 2021). From the perspective of generalization, previous research has quantified the benefits of imposing symmetry during learning (Long & Sedghi, 2019; Bietti et al., 2021; Sannai et al., 2021; Weber et al., 2020; Mei et al., 2021), often achieving rather tight bounds for simple models (Elesedy, 2021; Tahmasebi & Jegelka, 2023). At their core, these results are bounds on sample complexity, i.e. how much data is needed to learn a given task. In contrast, the effect of symmetries on the **computational complexity** of learning algorithms has not been previously studied. Broadly speaking, generalization bounds are necessary but not sufficient to show efficient learnability, as there can be exponentially large gaps between sample complexity and runtime lower bounds.

Indeed, an active line of research in learning theory studies the hardness of learning fully-connected neural networks via "correlational statistical query" (CSQ) algorithms (defined in Sec. 2.1), which notably encompass gradient descent. In particular, for a Gaussian data distribution, Diakonikolas et al. (2020) proved exponential lower bounds for learning shallow neural networks. Such works provide valuable impossibility results, demonstrating that one cannot hope to efficiently learn *any* function represented by a small neural networks under simple data distributions.

In this work, we ask: does invariance provide a restrictive enough subclass of neural network functions to circumvent these impossibility results? Our main finding, shown in various settings, is that an inductive bias towards symmetry is not strong enough to alleviate the fundamental hardness of learning neural nets via gradient descent. We view this result as a guide for future theoretical work: additional function structure is generally needed, even in the invariant setting, to achieve guarantees of efficient learnability. Our proof techniques largely rely on extending techniques from the general

setting to symmetric function classes. In many of our lower bounds, imposition of symmetry on a function class reduces the computational complexity of learning by a factor proportional to the size of the group. Some of our lower bounds use simple symmetrizations of existing hard families, while others require bona fide new classes of hard invariant networks. For example, intuition from message passing GNN architectures seems to indicate that complexity would grow exponentially with the number of such message passing steps; however, our results show that even a single message passing layer gathers sufficient features to form exponentially hard functions (Liao et al., 2020).

**Our Contributions**    We consider various interrelated questions on the computational hardness of learning, focusing primarily on how hard it is to learn data generated by invariant architectures.

**Question** (primary). *Given Gaussian i.i.d. inputs in $n$ variables, how hard is it to learn the class of shallow single hidden layer invariant architectures in the correlational statistical query model?*

We answer this question in various settings. As a warm-up, we give general (SQ) lower bounds for learning invariant Boolean functions in Sec. 3. Next, we show that graph neural networks with a single message passing layer are exponentially hard to learn, either in the number of nodes or feature dimension (Theorem 3 and Theorem 4, respectively). Hardness in the feature dimension results from careful symmetrization of the hard functions in Diakonikolas et al. (2020). This technique did not extend to the node dimension hardness result in Theorem 3, for which we instead constructed a customized set of hard functions whose outputs are parities in the degree counts of the graph. Since adjacency matrices of graphs are often Boolean valued, this setting also offers general SQ lower bounds scaling exponentially with the number of nodes $n$. For subgroups of permutations, we prove CSQ hardness results for "symmetrized" (frame-averaged) neural network classes encapsulating CNNs (Theorem 7 of Sec. 5). The proofs here are based on constructions in Goel et al. (2020) and Diakonikolas et al. (2020), ensuring that properly chosen frames retain the underlying structure necessary (e.g., sign invariance) to guarantee orthogonality of hard functions.

These lower bounds constitute our main results, but we provide two complementary results to provide an even more complete picture. First, we show that classes of sparse invariant polynomials can be learned efficiently, but not by gradient descent, via a straightforward extension of the algorithm in Andoni et al. (2014) for arbitrary polynomials (Sec. 6). This learning algorithm is SQ, and we give various $n^{\Omega(\log n)}$ CSQ lower bounds for learning over $n^{\Omega(\log n)}$ many orthogonal invariant polynomials of degree at most $O(\log(n))$. Second, we take a classical computational complexity perspective to establish the NP hardness of learning GNNs. The work of Blum & Rivest (1988) established that even 3-node feedforward networks are NP hard to train, and later expanded to average case complexity for improper learning by various works (Daniely et al., 2014; Daniely, 2016; Klivans & Kothari, 2014). Although it is perhaps not surprising that these hardness results can be extended to invariant architectures, for the sake of completeness (and since we were unaware of an extension in the literature), in Sec. 4.3 we give an NP hardness result for proper learning of the weights of a GNN architecture. Finally, Sec. 7 provides a few experiments verifying that the hard classes of invariant functions we propose are indeed difficult to learn.

## 1.1 Related work

A long line of research studies the hardness of learning neural networks. We briefly review the most relevant works here, and refer the reader to App. A for a more holistic review.

Our work is largely motivated to extend hardness results for feedforward networks to equivariant and symmetric neural networks. Early results showed that there exist rather artificial distributions of data for which feedforward networks are hard to learn in worst-case settings (Judd, 1987; Blum & Rivest, 1988; Livni et al., 2014). Interest later grew into studying hardness for more natural settings via the statistical query framework (Kearns, 1998; Shamir, 2018). Goel et al. (2020) showed superpolynomial CSQ lower bounds for learning single hidden layer ReLU networks, subsequently strengthened to exponential lower bounds in Diakonikolas et al. (2020). We use the hard class of networks in Diakonikolas et al. (2020) as a basis for the frame averaged networks in our work. Hardness results for learning feedforward neural networks have also been shown by proving reductions to average case or cryptographically hard problems (Chen et al., 2022a; Daniely & Vardi, 2020).

Though our work is focused on runtime lower bounds, some algorithms exist for learning restricted classes of geometric networks efficiently such as CNNs (Brutzkus & Globerson, 2017; Du et al.,

2017; 2018a). Zhang et al. (2020); Li et al. (2021) give algorithms learning one hidden layer GNNs via gradient based algorithms, which are efficient when certain factors such as the condition number of the weights are bounded. We should also note that for practical separations between SQ and CSQ algorithms, Andoni et al. (2014) show that learning sparse polynomials of degree $d$ in $n$ variables requires $\Omega(n^d)$ CSQ queries, but often only $\tilde{O}(nd)$ SQ queries. We extend this to polynomials in the invariant polynomial ring in Sec. 6.

## 2 BACKGROUND AND NOTATION

We use $a$, $\boldsymbol{a}$, and $\boldsymbol{A}$ to denote scalars, vectors, and matrices. $\boldsymbol{I}_n$ denotes the identity matrix of dimension $n$. We denote groups by $G$ and graphs by $\mathcal{G}$. We denote the normal distribution supported over $\mathbb{R}^n$ with mean $\boldsymbol{v} \in \mathbb{R}^n$ and covariance matrix $\boldsymbol{M} \in \mathbb{R}^{n \times n}$ as $\mathcal{N}(\boldsymbol{v}, \boldsymbol{M})$, often abbreviated as $\mathcal{N}$ for $\mathcal{N}(\boldsymbol{0}, \boldsymbol{I}_n)$. We also denote by $\mathbb{G}_n$ the set of graphs on $n$ vertices and, when clear from context, $\mathcal{E}$ an arbitrary distribution over $\mathbb{G}_n$. To ease notation, we often use $[n]$ to denote the set $\{1, \ldots, n\}$.

Let $\mathcal{X}$ be an input space and $\mathcal{D}$ a distribution on $\mathcal{X}$. Given functions $f, g : \mathcal{X} \to \mathbb{R}$, their inner product is $\langle f, g \rangle_{\mathcal{D}} = \mathbb{E}_{\mathcal{D}}[fg]$ with corresponding norm $\|f\|_{\mathcal{D}} = \sqrt{\langle f, f \rangle_{\mathcal{D}}}$. For functions $f, g : \mathcal{X} \to \{-1, +1\}$ whose outputs are Boolean, the classification error is $\mathbb{P}_{\boldsymbol{x} \sim \mathcal{D}}[f(\boldsymbol{x}) \neq g(\boldsymbol{x})]$.

### 2.1 SQ LEARNING FRAMEWORK

The well-studied statistical query (SQ) model offers a restricted query complexity based model for proving hardness, which encapsulates most algorithms in practice (Kearns, 1998; Reyzin, 2020). Given a joint distribution $\mathcal{D}$ on input/output space $\mathcal{X} \times \mathcal{Y}$, any SQ algorithm is composed of a set of queries. Each query takes as input a function $g : \mathcal{X} \times \mathcal{Y} \to [-1, 1]$ and tolerance parameter $\tau > 0$, and returns a value $\mathrm{SQ}(g, \tau)$ in the range:

$$\mathbb{E}_{(\boldsymbol{x},y) \sim \mathcal{D}}[g(\boldsymbol{x}, y)] - \tau \leq \mathrm{SQ}(g, \tau) \leq \mathbb{E}_{(\boldsymbol{x},y) \sim \mathcal{D}}[g(\boldsymbol{x}, y)] + \tau. \tag{1}$$

A special class of queries are **correlational statistical queries (CSQ)**, where the query function $g : \mathcal{X} \to [-1, 1]$ is only a function of $\boldsymbol{x}$, and the oracle returns $\mathbb{E}_{(\boldsymbol{x},y) \sim \mathcal{D}}[g(\boldsymbol{x}) \cdot y]$ the correlation of $g$ with $y$ up to error $\tau$.

Hardness is quantified as the number of queries required to learn a function $y = c^*(x)$, drawn from a function class $\mathcal{C}$ up to a desired maximal error. Notably, SQ hardness results imply hardness for any gradient based algorithm, as gradients with respect to a loss can be captured by statistical queries. Furthermore, only CSQ oracles are required for the mean-squared error (MSE) loss.

**Example 1** (GD from CSQ). Given a function $N_\theta : \mathcal{X} \to \mathbb{R}$ with parameters $\theta$, gradients of the MSE loss with respect to $\theta$ can be estimated with a correlational statistical query for each parameter:

$$\mathbb{E}_{(\boldsymbol{x},y) \sim \mathcal{D}}\left[\nabla_\theta \tfrac{1}{2}(y - N_\theta(\boldsymbol{x}))^2\right] = \underbrace{\mathbb{E}[N_\theta(\boldsymbol{x})\nabla_\theta N_\theta(\boldsymbol{x})]}_{y- \text{ independent}} - \underbrace{\mathbb{E}[y \nabla_\theta N_\theta(\boldsymbol{x})]}_{\text{CSQ with} \nabla_\theta N_\theta}. \tag{2}$$

Access to the distribution $\mathcal{D}$ is typically provided through samples, and the tolerance $\tau$ in part captures the error in statistical sampling of quantities such as the gradient. In fact, by drawing $O(\log(1/\delta)/\tau^2)$ samples, standard Hoeffding bounds guarantee estimates of a given query are within tolerance $\tau$ with probability $1 - \delta$. In the SQ (CSQ) framework, the complexity of learning is determined by the tolerance bound and number of queries (loosely corresponding to "steps" in an algorithm) needed to learn a function class.

**Definition 1** (SQ (CSQ) Learning). Given a function class $\mathcal{C}$ and distribution $\mathcal{D}$ over input space $\mathcal{X}$, an algorithm SQ (CSQ) learns $\mathcal{C}$ up to classification error ($\ell_2$ squared loss) $\epsilon$ if, given only SQ (CSQ) oracle access to $(\boldsymbol{x}, c^*(\boldsymbol{x})), \boldsymbol{x} \sim \mathcal{D}$ for some unknown $c^* \in \mathcal{C}$, the algorithm outputs a function $f$ such that $\mathbb{P}_{\boldsymbol{x} \sim \mathcal{D}}[f(\boldsymbol{x}) \neq c^*(\boldsymbol{x})] \leq \epsilon$ (i.e. $\|f - c^*\|_{\mathcal{D}} \leq \epsilon$).

SQ lower bounds for Boolean functions are typically shown by finding a set of roughly orthogonal functions that lower bound the so-called statistical dimension. For real-valued function classes, lower bounds on the statistical dimension imply CSQ lower bounds. Applying more general SQ lower bounds is challenging, unless one reduces the problem of learning a real-valued function to that of learning a subset therein of (say) Boolean functions within the class (Chen et al., 2022a). We cover this in more detail and also discuss other learning theoretic frameworks in App. B.

## 3 WARM UP: INVARIANT BOOLEAN FUNCTIONS

Before detailing our main results concerning real-valued functions, we explore the foundational context of Boolean functions where the SQ formalism originated (Kearns, 1998; Blum et al., 1994). This illustrative analysis will serve as a warm-up for the more complex scenarios later. It is known that many classes of Boolean functions have exponential query complexity (e.g., the class of parity functions) arising from the orthogonality properties of Boolean functions. Enforcing invariance under a given group $G$, the number of orbits of the $2^n$ bitstrings gives an analogous hardness metric, though care must be taken in dealing with the distribution of these orbits.

More formally, in the Boolean setting, let input distribution $\mathcal{D}$ be uniform over $\mathcal{X} = \{-1, +1\}^n$. For a group $G$ with representation $\rho$, let $\mathcal{O}_\rho = \{\{\rho(g) \cdot \boldsymbol{x} : g \in G\} : \boldsymbol{x} \in \{-1, +1\}^n\}$ denote the orbits of the inputs under the representation. For any given orbit $O \in \mathcal{O}_\rho$, let $\boldsymbol{x}_O$ denote an orbit representative (i.e., some fixed element in the set $O$). We can represent any symmetric function $f : \mathcal{O}_n \to \{-1, +1\}$ as a function of these orbits, where the correlation of two functions $f, g$ is

$$\langle f, g \rangle_\mathcal{D} = \mathbb{E}[fg] = 2^{-n} \sum_{O_k \in \mathcal{O}_\rho} |O_k| f(\boldsymbol{x}_{O_k}) g(\boldsymbol{x}_{O_k}). \quad (3)$$

The probability $p_{\mathcal{O}_\rho} : \mathcal{O}_\rho \to [0, 1]$ of a random bitstring falling in orbit $O_K \in \mathcal{O}_\rho$ is equal to $p_{\mathcal{O}_\rho}(O_k) = |O_k|/2^n$. With this notation, we have a general lower bound on the SQ hardness of learning invariant Boolean functions.

**Theorem 2** (Boolean SQ hardness). *For a given symmetry group $G$ with representation $\rho : G \to GL(\{-1, +1\}^n)$, let $\|p_{\mathcal{O}_\rho}\| := \left(\sum_{O_k \in \mathcal{O}_\rho} \left(\frac{|O_k|}{2^n}\right)^2\right)^{1/2}$ and let $\mathcal{H}_\rho$ be the class of symmetric Boolean functions, defined as*

$$\mathcal{H}_\rho = \{f : \{-1, +1\}^n \to \{-1, +1\} : \forall g \in G, \forall \boldsymbol{x} \in \{-1, +1\}^n : f(\rho(g) \cdot \boldsymbol{x}) = f(\boldsymbol{x})\}. \quad (4)$$

*Any SQ learner capable of learning $\mathcal{H}_\rho$ up to sufficiently small classification error probability $\epsilon$ ($\epsilon < 1/4$ suffices) with queries of tolerance $\tau$ requires at least $\tau^2 \|p_{\mathcal{O}_\rho}\|^{-2}/2$ queries.*

*Proof sketch.* We show that $1 - \|p_{\mathcal{O}_\rho}\|^2$ is the probability that, over independently drawn inputs $\boldsymbol{x}, \boldsymbol{x}'$, the distribution $(f(\boldsymbol{x}), f(\boldsymbol{x}'))$ is uniformly random over $f$ drawn uniformly from $\mathcal{H}_\rho$. From a proof technique in Chen et al. (2022a), this connects the task to one of distinguishing distributions, for which the SQ complexity is at least the stated amount. See App. C for complete proof. $\quad \square$

For a more direct lower bound, we can use Hölder's inequality such that $\|p_{\mathcal{O}_\rho}\|^2 \leq \frac{1}{2^n} \max_{O_k \in \mathcal{O}_\rho} |O_k| \leq |G| 2^{-n}$. This gives query complexity of at least $2^{n-1}|G|^{-1}\tau^2$. For commonly studied groups, assuming $\tau = O(1)$, the SQ complexity generally aligns with the number of orbits. Table 1 summarizes these hardness lower bounds for commonly studied groups[1].

| Group | $\max_{O_k \in \mathcal{O}_\rho} |O_k|$ | Query Complexity |
|---|---|---|
| Symmetric group on $n$ bits | $\frac{\binom{n}{n/2}}{2^n} = \frac{1}{\Theta(\sqrt{n})}$ | $\Theta(\sqrt{n})$ |
| Symmetric group on $n \times n$ graphs | $\frac{n!}{2^{n^2}} = 2^{-n^2 + n \log n + O(n)}$ | $\Omega(2^{O(n^2)})$ |
| Cyclic group on $n$ bits | $n$ | $\Omega(2^n/n)$ |

Table 1: Query complexity of learning common invariant Boolean function classes.

## 4 LOWER BOUNDS FOR GNNS

We now show statistical query lower bounds for graph neural networks, which are invariant to node permutations, in three settings: (1) SQ lower bounds scaling with the number of nodes $n$ for Erdős–Rényi distributed graphs with trivial node features, (2) CSQ lower bounds scaling in the node feature dimension for a fixed graph, and (3) a simple extension of NP hardness in a proper learning task (deciding if fitting a training set with a given GNN is possible).

---

[1]Certain sub-classes of $\mathcal{H}_\rho$ may suffice to attain hardness, much like how Parity functions suffice to prove exponential lower bounds in the traditional Boolean setting. We do not consider this strengthening here.

## 4.1 Hardness in number of nodes

We consider a class of two hidden layer GNNs that are only a function of the adjacency matrix, and show hardness in the number of nodes $n$. This covers a commonly used procedure where one learns node-level equivariant features in the first layer, aggregates these features, and passes them through an MLP (Dwivedi et al., 2020). We consider a Boolean function class of GNNs over the uniform distribution $\mathrm{Unif}(\{0,1\}^{n \times n})$, which can be viewed as the Erdős–Rényi random graph model with $p = 0.5.$[2] Node features $\boldsymbol{x} \in \mathbb{R}^n$ are trivial and always set to $\boldsymbol{x} = \mathbf{1}$.

**2 hidden layer GNN family**   We consider two layer GNNs $f = f^{(2)} \circ f^{(1)}$. These take the commonly used form of message passing $f^{(1)} : \{0,1\}^{n \times n} \to \mathbb{R}^{k_1}$ which aggregates $k_1$ permutation invariant features for the graph, followed by a single hidden layer ReLU MLP $f^{(2)} : \mathbb{R}^{k_1} \to \{0,1\}$.

$$[f^{(1)}_{\boldsymbol{a},\boldsymbol{b}}(\boldsymbol{A})]_i = \mathbf{1}_n^\top \sigma\left(a_i + b_i \boldsymbol{A}\boldsymbol{x}\right) \quad \text{(output of channel } i \in [k_1])$$

$$f^{(2)}_{\boldsymbol{u},\boldsymbol{v},\boldsymbol{W}}(\boldsymbol{h}) = \sum\nolimits_{i=1}^{k_2} u_i \sigma(\langle \boldsymbol{W}_{:,i}, \boldsymbol{h} \rangle + v_i),$$

(5)

with subscripts for trainable weights. For our hard class of functions, $k_1 = O(n)$ and $k_2 = O(n)$, so there are at most $O(n^2)$ parameters (proportional to the number of edges).

**Family of hard functions**   We define the hard functions in terms of degree counts $\boldsymbol{c_A} \in [n]^{n+1}$, where entry $[\boldsymbol{c_A}]_i$ counts the number of nodes that have $i - 1$ outgoing edges in $\boldsymbol{A}$:

$$[\boldsymbol{c_A}]_i = \sum\nolimits_{k=1}^n \mathbb{1}\left[[\boldsymbol{A}\mathbf{1}]_k = i - 1\right].$$

(6)

The hard functions in $\mathcal{H}_{ER,n}$ are enumerated over subsets $S \subseteq [n+1]$ and a bit $b \in \{0,1\}$:

$$\mathcal{H}_{ER,n} = \{g_{S,b} : S \subseteq [n+1], b \in \{0,1\}\}, \quad g_{S,b}(\boldsymbol{A}) = b + \sum\nolimits_{i \in S}[\boldsymbol{c_A}]_i \mod 2.$$

(7)

$g_{S,b}(\boldsymbol{A})$ is a sort of parity function supported over integers in $\boldsymbol{c_A}$ in $\mathcal{S}$. We show in Lemma 15 that the GNNs in Eq. (5) can construct these functions. Additionally, the class $\mathcal{H}_{ER,n}$ is Boolean, so hardness lower bounds apply in the general SQ model.

**Theorem 3** (SQ hardness of $\mathcal{H}_{ER,n}$). *Any SQ learner capable of learning $\mathcal{H}_{ER,n}$ up to classification error probability $\epsilon$ sufficiently small ($\epsilon < 1/4$ suffices) with queries of tolerance $\tau$ requires at least $\Omega\left(\tau^2 \exp(n^{\Omega(1)})\right)$ queries.*

*Proof sketch.* We form networks that in the message passing layer calculate $\boldsymbol{c_A}$, which is passed to the MLP calculating Eq. (7) as a sum of ReLUs via hat-like functions that mimic parities. Based on concentration properties of the Erdős–Rényi model, we show that for some $p < 0.5$, at least $\Theta(n^p)$ entries of $\boldsymbol{c_A}$ will have values scaling roughly as $O(\sqrt{n})$ and the probability that any such entry is odd converges to $1/2$. Also using concentration properties, we condition on a high probability region (where probability of these entries being odd are roughly independent from each other), and use hardness of learning parity functions to get SQ lower bounds. $\qquad\qquad\square$

## 4.2 Hardness in feature dimension

In this section, we derive correlational statistical query lower bounds (CSQ) for commonly used graph neural network architectures (GNNs), and state the corresponding learning-theoretic hardness result. The techniques in this section extend those of Diakonikolas et al. (2020), resulting in an exponential (in *feature* dimension) lower bound for learning one-hidden layer GNNs.

**1-hidden layer GNN**   For a fixed directed, unweighted graph $\mathcal{G}$ with $n \in \mathbb{N}$ vertices, let $\boldsymbol{A}(\mathcal{G}) \in \mathbb{R}^{n \times n}$ be the adjacency matrix or the Laplacian of $\mathcal{G}$. For input feature dimension $d \in \mathbb{N}$ and width parameter $k \in \mathbb{N}$, we consider the following set of functions:

$$F_{\mathcal{G}}^{d,k} := \left\{ f : \mathbb{R}^{n \times d} \to \mathbb{R}, \ f(\boldsymbol{X}) = \mathbf{1}_n^\top \sigma(\boldsymbol{A}(\mathcal{G})\boldsymbol{X}\boldsymbol{W})\boldsymbol{a} \mid \boldsymbol{W} \in \mathbb{R}^{d \times 2k}, \boldsymbol{a} \in \mathbb{R}^{2k} \right\},$$

(8)

for some nonlinearity $\sigma$. When the graph itself is part of the input, we have the function class:

$$\underline{F_n^{d,k} := \left\{ f : \mathbb{R}^{n \times d} \times \mathbb{G}_n \to \mathbb{R}, \ f(\boldsymbol{X}, \mathcal{G}) = \mathbf{1}_n^\top \sigma(\boldsymbol{A}(\mathcal{G})\boldsymbol{X}\boldsymbol{W})\boldsymbol{a} \mid \boldsymbol{W} \in \mathbb{R}^{d \times 2k}, \boldsymbol{a} \in \mathbb{R}^{2k} \right\}.}$$

(9)

---

[2]Our exponential lower bounds arise from the wide range of possible degrees of nodes in the graph. Though we do not generalize this result, this fact means that it is likely extendable to restricted sparse Erdős–Rényi models $G(n, p_n)$ where $p_n = \omega(1/n)$ (i.e. average degree grows arbitrarily with $n$).

**Family of hard functions** First, we define functions $f_\mathcal{G} : \mathbb{R}^{n\times 2} \to \mathbb{R}$ and $f_n : \mathbb{R}^{n\times 2} \times \bar{\mathcal{G}}_n \to \mathbb{R}$ as

$$f_\mathcal{G}(\boldsymbol{X}) = \mathbf{1}_n^\top \sigma\left(\boldsymbol{A}(\mathcal{G})\boldsymbol{X}\boldsymbol{W}^*\right)\boldsymbol{a}^*, \quad f_n(\boldsymbol{X}, \mathcal{G}) = \mathbf{1}_n^\top \sigma\left(\boldsymbol{A}(\mathcal{G})\boldsymbol{X}\boldsymbol{W}^*\right)\boldsymbol{a}^*, \tag{10}$$

$$\text{where} \quad \boldsymbol{W}^* := \begin{bmatrix} (\cos(\pi j/k))_{j\in[2k]}^\top \\ (\sin(\pi j/k))_{j\in[2k]}^\top \end{bmatrix} \in \mathbb{R}^{2\times 2k}, \qquad \boldsymbol{a}^* = \left((-1)^j\right)_{j\in[2k]} \in \mathbb{R}^{2k}. \tag{11}$$

Given a set of matrices $\mathcal{B} \subset \mathbb{R}^{d\times 2}$, our family of hard functions can now be written as:

$$C_\mathcal{G}^\mathcal{B} := \left\{ g_\mathcal{G}^\boldsymbol{B} : \boldsymbol{X} \mapsto \frac{f_\mathcal{G}(\boldsymbol{X}B)}{\|f_\mathcal{G}\|_\mathcal{N}} \mid B \in \mathcal{B} \right\}, \quad C_n^\mathcal{B} := \left\{ g_n^\boldsymbol{B} : (\boldsymbol{X}, \mathcal{G}) \mapsto \frac{f_n(\boldsymbol{X}B, \mathcal{G})}{\|f_n\|_{\mathcal{N}\times\mathcal{E}}} \mid B \in \mathcal{B} \right\}. \tag{12}$$

Below, we show exponential lower bounds [3] for learning $F_\mathcal{G}^{d,k}$ and $F_n^{d,k}$:

**Theorem 4** (Exponential CSQ lower bound for GNNs). *For any number of vertices $n$ independent of input dimension $d$ and width parameter $k$, let $\epsilon > 0$ be a sufficiently small error constant and $\mathcal{E}$ a distribution over $\mathbb{G}_n$, independent of $\mathcal{N}$, with $\Pr_\mathcal{E}(\{\emptyset\}) < \Omega(1) < 1$. Then there exists a set $\mathcal{B}$ of size at least $2^{\Omega(d^{\Omega(1)})}$ such that any CSQ algorithm that queries from oracles of concept $f \in C_\mathcal{G}^\mathcal{B}$ (resp. $C_n^\mathcal{B}$) and outputs a hypothesis $h$ with $\|f - h\|_\mathcal{N} \le \epsilon$ (resp. $\|f - h\|_{\mathcal{N}\times\mathcal{E}} \le \epsilon$) requires either $2^{d^{\Omega(1)}}$ queries or at least one query with precision $d^{-\Omega(k)} + 2^{-d^{\Omega(1)}}$.*

*Proof sketch.* Full details of the proof can be found in App. E. From Lemma 17 of Diakonikolas et al. (2020), there exists a set $\mathcal{B} \subset \mathbb{R}^{d\times 2}$ of size at least $2^{\Omega(d^{\Omega(1)})}$ such that

$$\|\boldsymbol{B}^\top \boldsymbol{B}'\|_2 = O(d^{-\Omega(1)}) < 1 \text{ if } \boldsymbol{B} \ne \boldsymbol{B}' \tag{13}$$

and $\boldsymbol{B}^\top \boldsymbol{B} = \mathbf{1}_2$ which we use to index $C_\mathcal{G}^\mathcal{B}$. We design a specialized class of invariant orthogonal Hermite polynomials, such that the correlation in low degree moments vanishes and is bounded as:

$$|\langle f_\mathcal{G}(\cdot\boldsymbol{B}), f_\mathcal{G}(\cdot\boldsymbol{B}')\rangle_\mathcal{N}| \le \|\boldsymbol{B}^\top\boldsymbol{B}'\|_2^k \sum_{m>k} \|f_\mathcal{G}^{[m]}\|_\mathcal{N}^2 = \|\boldsymbol{B}^\top\boldsymbol{B}'\|_2^k \|f_\mathcal{G}\|_\mathcal{N}^2. \tag{14}$$

Finally, we use Eq. (13) to get the desired almost uncorrelatedness of elements in $C_\mathcal{G}^\mathcal{B}$ and use a simple total probability argument to extend this to $C_n^\mathcal{B}$. $\qquad\square$

## 4.3 NP HARDNESS OF PROPER LEARNING OF GNNS

The classic work of Blum & Rivest (1988) showed that there exist datasets for which determining whether or not a 3-node neural network can fit that dataset is NP hard. It is straightforward to prove a similar result for invariant networks. We map the task of training a single hidden layer GNN to the NP hard problem of learning halfspaces with noise (Guruswami & Raghavendra, 2009; Feldman et al., 2012). Here, one is given a set of $N$ labeled examples $\{\boldsymbol{x}_i, y_i\}_{i=1}^N$ and must determine whether there exists a halfspace parameterized by vector $\boldsymbol{v}$ and constant $\theta$ that correctly classifies a specified fraction of the labels such that $\text{sign}\left(\langle \boldsymbol{v}, \boldsymbol{x}_i\rangle - \theta\right) = y_i$. We map a GNN learning problem to this task, as informally described below and formally detailed in App. D.

**Proposition 5** (NP hardness of GNN training; informal). *There exists a sequence of datasets indexed by the number of nodes $n$, each of the form $\{\boldsymbol{A}_i, \boldsymbol{x}_i, y_i\}_{i=1}^N$ where $y_i \in \{-1, +1\}$, such that determining whether a GNN with parameters $\boldsymbol{a}, \boldsymbol{b} \in \mathbb{R}^k$, $c \in \mathbb{R}$ and $k = O(n)$ of the form*

$$f_{\boldsymbol{a},\boldsymbol{b},c}(\boldsymbol{x}, \boldsymbol{A}) = c + \sum_{i=1}^k \mathbf{1}^\top \text{relu}\left(\boldsymbol{x} + a_i\boldsymbol{A}\boldsymbol{x}\right)b_i \tag{15}$$

*can correctly classify a specific fraction of the data is* NP *hard.*

## 5 CSQ LOWER BOUND FOR CNNS AND FRAME AVERAGING

Several commonly used invariant architectures take the form of symmetrized neural networks over permutation subgroups $G \le S_n$, which can be generalized as frame-averaging architectures (Puny et al., 2023). This includes translation invariant CNNs, group convolutional networks, and networks with preprocessing steps such as sorting. We study the hardness of learning these invariant functions.

---

[3]To have a meaningful set of functions, $g_\mathcal{G}^\boldsymbol{B}$ and $g_n^\boldsymbol{B}$ must not vanish, which holds when $\sigma$ is not a low-degree polynomial (ReLU suffices, see Remark 12 of Diakonikolas et al. (2020)) and $\mathcal{G} \not\equiv \emptyset$ (graph with no edges).

**Frame averaging and its connection with CNNs.** Let $G$ be a subgroup of the permutation group $S_n$ that acts on $\mathbb{R}^{n \times d}$ by permuting the rows. A frame is a function $\mathcal{F} : \mathbb{R}^{n \times d} \to 2^G \backslash \emptyset$ satisfying $\mathcal{F}(g \cdot \boldsymbol{X}) = g\mathcal{F}(\boldsymbol{X})$ set-wise. As noted in Puny et al. (2023), given some frame $\mathcal{F}$, one can transform an arbitrary function (or neural network) $h(\boldsymbol{X})$ into an invariant function by averaging its values as $\frac{1}{|\mathcal{F}(\boldsymbol{X})|} \sum_{g \in \mathcal{F}(\boldsymbol{X})} h(g^{-1}\boldsymbol{X})$. For instance, setting $\forall \boldsymbol{X} : \mathcal{F}(\boldsymbol{X}) = G$ recovers the Reynolds (or group-averaging) operator. Given a fixed frame $\mathcal{F}$, we will first show CSQ lower bounds for the following class of frame-averaged one-hidden-layer fully connected nets:

$$\mathcal{H}_{\mathcal{F}} := \left\{ f : \mathbb{R}^{n \times d} \to \mathbb{R}, f(\boldsymbol{X}) = \frac{1}{\sqrt{|\mathcal{F}(\boldsymbol{X})|}} \sum_{g \in \mathcal{F}(\boldsymbol{X})} \boldsymbol{a}^\top \sigma(\boldsymbol{W}^\top(g^{-1}\boldsymbol{X}))\mathbf{1}_d \mid \boldsymbol{W} \in \mathbb{R}^{n \times k}, \boldsymbol{a} \in \mathbb{R}^k \right\},$$

for some nonlinearity $\sigma$.

**Example 2** (Frame for CNN). Set $d = 1$ and let $G$ be the cyclic group acting on $\mathbb{R}^n$ via cyclic shifts of its elements with frame $\mathcal{F}(\boldsymbol{X}) = G, \forall X \in \mathbb{R}^n$. Then, $\mathcal{F}_n^d$ consists of CNNs with one convolutional layer and $k$ hidden channels.

**Remark 6** (Difficulty of frames). Since the frame $\mathcal{F}(\boldsymbol{X})$ may vary by datapoint $\boldsymbol{X}$, the distribution $\mathrm{Unif}(\mathcal{F}(\boldsymbol{X})^{-1}) \circ \boldsymbol{X}$ can be significantly different from the original distribution over $\boldsymbol{X}$, even if $\boldsymbol{X} \sim g\boldsymbol{X}$ for all $g$. For example, consider the frame $\mathcal{F}(\boldsymbol{X})$ containing all permutations that sort $\boldsymbol{X}$ lexicographically (where $|\mathcal{F}(\boldsymbol{X})| > 1$ if $\boldsymbol{X}$ contains a repeated row). Even if $\boldsymbol{X} \sim \mathcal{N}$ with $\mathcal{N}$ invariant to row-permutation, the resultant distribution is not row permutation invariant.

We focus on certain cases where such effects are not too pronounced. For instance, it is simple to check that if $\mathcal{F}(\boldsymbol{X})$ is constant over $\boldsymbol{X}$, then $\mathcal{F}(\boldsymbol{X}) = G \; \forall \boldsymbol{X}$ (Lemma 28). In the following, we assume that $G$ is a polynomial-sized (in $n$) subgroup of $S_n$.

**Family of hard functions.** Recall the low-dimensional function from the proof of Theorem 4:

$$f_{\exp}^* : \mathbb{R}^{2 \times d} \to \mathbb{R} \text{ with } f(\boldsymbol{X}) = (\boldsymbol{a}^*)^\top \sigma((\boldsymbol{W}^*)^\top \boldsymbol{X})\mathbf{1}_d \tag{16}$$

for some special parameter $\boldsymbol{a}^* \in \mathbb{R}^{2k}$ and $\boldsymbol{W}^* \in \mathbb{R}^{2 \times 2k}$ in Eq. (11). We now define the family of hard functions, indexed by a set of matrices $\mathcal{B} \subset \mathbb{R}^{n \times 2}$ obtained from Lemma 29:

$$C_{\mathcal{F}}^{\mathcal{B}} = \left\{ g_{\boldsymbol{B}} : \mathbb{R}^{n \times d} \to \mathbb{R} \text{ with } g_{\boldsymbol{B}}(\boldsymbol{X}) = \frac{\sum_{g \in G} f_{\exp}^*(\boldsymbol{B}^\top g^{-1}\boldsymbol{X})}{\sqrt{|G| \cdot \|f_{\exp}^*\|_{\mathcal{N}}}} \mid \boldsymbol{B} \in \mathcal{B} \subset \mathbb{R}^{n \times 2} \right\}. \tag{17}$$

**Theorem 7** (Exponential CSQ lower bound for polynomial-sized group averaging). *For feature dimension $d$ independent of inputs $n$ and width parameter $k = \Theta(n)$, let $\epsilon > 0$ be a sufficiently small error constant. Then there exists a set $\mathcal{B}'$ of size at least $2^{\Omega(d^{\Omega(1)})}/|G|^2$ such that: for any target $g \in C_{\mathcal{F}}^{\mathcal{B}'}$ with $\|g\|_{\mathcal{N}} = 1$, any CSQ algorithm outputting a hypothesis $h$ with $\|g - h\|_{\mathcal{N}} \leq \epsilon$ requires either $2^{n^{\Omega(1)}}/|G|^2$ queries or at least one query with precision $|G|2^{-n^{\Omega(1)}} + \sqrt{|G|}n^{-\Omega(k)}$.*

*Proof sketch.* Using a union bound argument over the original set $\mathcal{B}$ from Diakonikolas et al. (2020), and concentration of inner product between permuted unit vectors, we obtain from Lemma 29 a set of orthogonal matrices $\mathcal{B}'$ such that both $\|\boldsymbol{B}(\boldsymbol{B}')^T\|$ and $\|g\boldsymbol{B}(g'\boldsymbol{B}')^T\|$ are small $\forall \boldsymbol{B} \neq \boldsymbol{B}' \in \mathcal{B}'$, $\forall g \neq g' \in G$. This construction costs a factor of $|G|^2$ in $|\mathcal{B}'|$. The rest of the proof proceeds similarly; see App. F.1. $\square$

We also obtain superpolynomial lower bounds for more general frames using the technique of Goel et al. (2020). Here, we drop the assumption that $G$ is polynomially-sized. The hard functions are based on parity functions, similar to Goel et al. (2020) (but with an extra input dimension for $\boldsymbol{X}$):

$$f_S : \mathbb{R}^{n \times d} \to \mathbb{R} \text{ with } f_S(\boldsymbol{X}) = \mathbf{1}_d^\top \left( \sum_{\boldsymbol{w} \in \{-1,1\}^m} \chi(\boldsymbol{w})\sigma(\langle \boldsymbol{w}, \boldsymbol{X}_S \rangle / \sqrt{m}) \right), \qquad S \subseteq_m [n]. \tag{18}$$

where $\subseteq_m$ indicates a size $m$ subset, $\chi$ is the parity function $\chi(\boldsymbol{w}) = \prod_i w_i$ and $\boldsymbol{X}_S \in \mathbb{R}^{m \times d}$ denotes entries of $\boldsymbol{X}$ in $S$. Here, $\langle \boldsymbol{w}, \boldsymbol{X}_S \rangle := \sum_{i=1}^m w_i(\boldsymbol{X}_S)_{i,:}$ and $\sigma : \mathbb{R}^d \to \mathbb{R}^d$ is an activation function. The hard *invariant* function class consists of frame-averages of the functions above:

$$\mathcal{H}_{S,\mathcal{F}} := \left\{ \tilde{f}_S : \mathbb{R}^{n \times d} \to \mathbb{R} \text{ with } f(\boldsymbol{X}) = \frac{1}{\sqrt{|\mathcal{F}(\boldsymbol{X})|}} \sum_{g \in \mathcal{F}(\boldsymbol{X})} f_S(g^{-1}\boldsymbol{X}) \mid S \subset_m [n] \right\}. \tag{19}$$

Below, we show a hardness result for this class inspired by that of Goel et al. (2020), whenever the frame $\mathcal{F}$ is **sign invariant**: $\mathcal{F}(\boldsymbol{X}) = \mathcal{F}(\boldsymbol{X} \circ \boldsymbol{z})$ for almost all $\boldsymbol{X}$ and for all $\boldsymbol{z} \in \{-1, 1\}^n$.

**Theorem 8** (Superpolynomial CSQ lower bound for sign-invariant frame averaging). *If $\mathcal{F}$ is a sign-invariant frame, then for any $c > 0$, any CSQ algorithm that queries from an oracle of concept $f \in \mathcal{H}_{\mathcal{S},\mathcal{F}}$ and output a hypothesis $h$ with $\|f - h\|_{\mathcal{N}} \leq \Theta(1)$ needs at least $\Omega(n^{\Omega(\log n)}/M(n))$ queries or at least one query with precision $O(n^{-c/2})$, where $M(n)$ is the size of the largest orbit of size $m \leq \log(n)$ subsets of $[n]$ under $G$. If moreover $|\mathcal{F}(\boldsymbol{X})| = 1$ for almost all $\boldsymbol{X}$ (a "singleton" frame), then the same result holds with $M(n)$ replaced by $1$.*

*Proof sketch.* The proof hinges on the structure of $f_S$, and in particular that applying $g$ to the input of $f_S$ yields $f_{gS}$. The sign-invariance of $\mathcal{F}$ is necessary to ensure that the proof of Goel et al. (2020), which involves introducing random sign-flips, goes through. See Lemma 33 for the full proof. □

**Example 3** (Singleton frames). An example of such a frame is the lexicographical sorting of $\boldsymbol{X} \in \mathbb{R}^{n \times d}$ according to the absolute value of the entries (to make the frame sign-invariant). With probability 1, $\boldsymbol{X}$ does not have any repeated values, and thus there is a unique lexicographical sort for $\boldsymbol{X}$ almost surely. In fact, for any group $G$, a singleton frame exists if $\boldsymbol{X}$ has no self-symmetry with probability 1, i.e. $\boldsymbol{X} \neq g\boldsymbol{X}$ for all $g \in G$.

For the hard class function to be nonvanishing, we show in Corollary 39 that at least a superpolynomial-sized subset of $\mathcal{H}_{\mathcal{S},\mathcal{F}}$ has norm lower-bounded by $\Omega(\text{poly}(n)^{-1})$, assuming that $|\mathcal{F}(\boldsymbol{X})|$ is polynomial in $n$ almost surely ($|G|$ does not have to be polynomial in $n$).

# 6 SEPARATION BETWEEN SQ AND CSQ FOR INVARIANT FUNCTION CLASSES

A natural question is whether there exist real-valued function classes which are hard to learn in the CSQ setting (and hence by gradient descent with mean squared error loss), but efficient to learn via a more carefully constructed non-CSQ algorithm. In the general (non-invariant) setting, one such separation is that of learning $k$-sparse polynomials, i.e., degree $d$ polynomials in $n$ variables expressible as sums of only $k$ unique monomials over inputs drawn from a product distribution $\mathcal{D} = \bigtimes_{i=1}^{n} \mu_i$. Here, an efficient SQ (but not CSQ) algorithm called the GROWING-BASIS algorithm is known from Andoni et al. (2014). CSQ algorithms require $\Omega(n^d)$ queries corresponding to the number of degree $d$ orthogonal polynomials, but the GROWING-BASIS algorithms learns with $O(nkf(d))$ SQ queries, where $f(d)$ may be exponential in $d$ but independent of $n$. We informally illustrate the invariant extension of this here, leaving complete details to App. G.

To perform this extension, we apply GROWING-BASIS to more general spaces (so-called 'rings') of invariant polynomials whose basis consists of independent invariant functions or 'generators' $\{g_i \in \mathbb{R}[\boldsymbol{x}]^G\}_{i \in [r]}$, for some $r \in \mathbb{N}$. Independence here asserts that any $g_i$ cannot be written as a polynomial in the others. The space of all polynomials in $g_i(\boldsymbol{x})$ with degree $d$ and sparsity $k$ (both in $g_i$'s expansion) is denoted by $\mathbb{R}[g_1(\boldsymbol{x}), \ldots, g_r(\boldsymbol{x})]_{d,k}$. We have the following separation result:

**Lemma 9** (informal). *Let $\boldsymbol{x} \sim \mathcal{D}$ such that the induced distribution of $(g_i(\boldsymbol{x}))_{i \in [r]}$ is a product distribution. Then any CSQ algorithm that learns $f \in \mathbb{R}[g_1(\boldsymbol{x}), \ldots, g_r(\boldsymbol{x})]_{d,k}$ of degree at most $d = O(\log n)$ to a small constant error requires $\Omega(r^d)$ CSQ queries of bounded tolerance. However, the GROWING-BASIS algorithm learns $f$ with $O(kr^2 \log r)$ SQ queries of precision $\Omega(1/(k \, \text{poly}(r)))$.*

A more rigorous exposition and proof is prepared in App. G. Finding independent generators itself can be a challenging task; however, we provide an example below, and more in App. G, for separations over groups where $r = \Omega(n)$ generators are known (Derksen & Kemper, 2015).

**Example 4.** The sign group $G$ flips the sign of input elements and is commonly used to study eigenvectors (Lim et al., 2022). $G$ is indexed by vectors $\boldsymbol{v} \in \{-1, +1\}^n$ with representation $\rho(\boldsymbol{v}) \cdot \boldsymbol{x} = \boldsymbol{x} \odot \boldsymbol{v}$ where $\odot$ represents pointwise multiplication. Consider generators $g_1, \ldots, g_n$ where $g_i(\boldsymbol{x}) = |\boldsymbol{x}_i|$. For inputs $\boldsymbol{x} \sim \text{Unif}([-1, +1]^n)$, the distribution of $[g_1(\boldsymbol{x}), \ldots, g_n(\boldsymbol{x})]$ is a product distribution $\text{Unif}([0, 1]^n)$ with the "shifted" Legendre polynomials as the orthogonal polynomials. For $d$ at most $O(\log n)$, any CSQ algorithm requires $\Omega(n^d)$ queries of bounded tolerance to learn these polynomials, whereas the GROWING-BASIS algorithm learns in $O(kn^2 \log n)$ SQ queries.

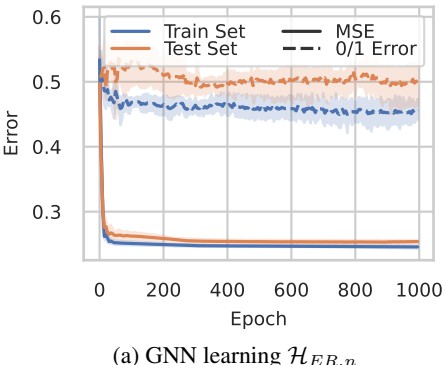 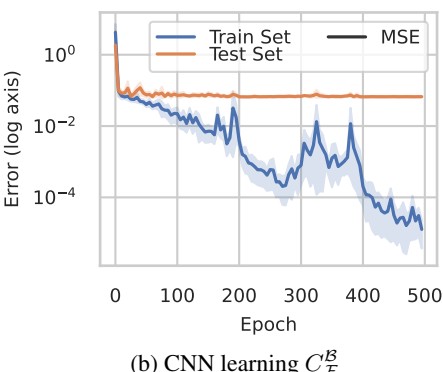

(a) GNN learning $\mathcal{H}_{ER,n}$            (b) CNN learning $C_{\mathcal{F}}^{\mathcal{B}}$

Figure 1: Overparameterized GNN (a) and CNN (b) fail to learn functions from the class $\mathcal{H}_{ER,n}$ and $C_{\mathcal{F}}^{\mathcal{B}}$ respectively by either failing to fit the training set or overfitting the data. Plots are aggregated and averaged over five random realizations.

# 7 EXPERIMENTS

We train overparameterized GNNs and CNNs on the hard functions from Sec. 4 and Sec. 5, respectively. Specifically, we attempt to learn relatively small instances drawn from the function classes $\mathcal{H}_{ER,n}$ for graphs (Theorem 3) and $C_{\mathcal{F}}^{\mathcal{B}}$ for translationally invariant data (Theorem 7). For the function class $\mathcal{H}_{ER,n}$, we consider $n = 15$ node inputs and the target function $g_{S,b}$ where $S \subseteq [n]$ is a random subset of size 7 and $b$ is either 0 or 1 at random. For the function class $C_{\mathcal{F}}^{\mathcal{B}}$, we choose a random orthogonal matrix $B$ and then symmetrize the function class as per Eq. (17).

Fig. 1a and 1b plot the performance of the GNN and CNN respectively. The GNN is unable to fit even the training data consisting of 225 $n = 15$ node graphs drawn uniformly from the Erdős–Rényi model (i.e., $p = 0.5$). The CNN trained on $n = 50$ dimension inputs fits the training set of size 500 appropriately, but fails to generalize in the mean squared error loss. The GNN/CNN were overparameterized with 3 layers of graph/cyclic convolution followed by a two layer ReLU MLP on the aggregated invariant features. We refer the reader to App. H for further details.

# 8 DISCUSSION

Our study asked how hard it is to learn relatively simple function classes constrained by symmetry. Setting aside mathematical formality, one may conjecture that such function classes are good representations or approximations of the data observed in nature. Our results indicate that such an assumption is unlikely to be a realistic one — at least in worst-case settings, where we show such functions would be exponentially hard to learn. Provable guarantees of learning will have to incorporate further assumptions in the model class or biases in training to better account for the practical success of learning algorithms (Lawrence et al., 2021; Gunasekar et al., 2018; Le & Jegelka, 2022).

We now state potential directions for future work and conjectures. It would be interesting to extend our hardness results beyond the SQ setting or to apply to continuous groups. For example, a natural question is whether symmetric functions are cryptographically hard to learn, as shown in Chen et al. (2022a) for two hidden layer ReLU networks. A line of work in probability and statistics has identified barriers to many algorithms solving average-case settings of classic optimization problems, such as max cut or largest independent set. These include the overlap gap property (Gamarnik, 2021) and bounds on low degree algorithms (Hopkins & Steurer, 2017; Brennan et al., 2020). Whether these barriers extend to the types of problems studied in geometric deep learning is an open question. We should note that there are restricted settings where learning neural networks is possible (see App. A). Recent work (Chen et al., 2022b; Chen & Narayanan, 2023) has shown that ReLU networks with $O(1)$ hidden nodes are learnable in polynomial time. Extending this result to invariant network classes with $O(1)$ channels would similarly expand the set of learnable invariant networks.

ACKNOWLEDGEMENTS

The authors thank Sitan Chen for insightful discussions and feedback. BTK and MW were supported by the Harvard Data Science Initiative Competitive Research Fund and NSF award 2112085. TL and SJ were supported by NSF awards 2134108 and CCF-2112665 (TILOS AI Institute), and Office of Naval Research grant N00014-20-1-2023 (MURI ML-SCOPE). HL is supported by the Fannie and John Hertz Foundation and the NSF Graduate Fellowship under Grant No. 1745302.

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

## TABLE OF CONTENTS

## A  EXTENDED RELATED WORKS

**Restricted learning algorithms for CNNs and GNNs**   Various works show that convolutional and graph neural networks can be efficiently learned under various assumptions on the form of the network. For given input distributions, Brutzkus & Globerson (2017); Du et al. (2017) give an efficient learning algorithm for learning a single convolution filter and Zhong et al. (2017) give a polynomial time algorithm for learning a sum of a polynomial number of convolution kernels applied to an input. These can be viewed as a single hidden layer network where the last layer weights are all equal to one. Du & Goel (2018) (improving on Goel et al. (2018)) give a polynomial time algorithm for learning single layer convolutional networks (not necessarily invariant) with filters that have stride at least half the filter width. These CNNs have a single channel but are not invariant to the cyclic group due to the presence of weights in the second layer that vary by each patch of the filter. Similarly, Du et al. (2018a) show that gradient descent can also learn networks of a similar form with non-overlapping patches in the random setting (weights are random) with high probability. Oymak & Soltanolkotabi (2018) learn convolutional networks of depth $D$ layers with a single kernel per layer and make a similar assumption that stride is at least as large as width. Zhang et al. (2020); Li et al. (2021) provide learning algorithms for a class of one hidden layer GNNs via gradient descent based algorithms whose sample complexity and runtime depend on factors like the condition number or norm of the weights of the target function. These algorithms are only guaranteed to be polynomial in runtime for the restricted instances when such factors are appropriately bounded.

**Learning feedforward neural networks**   The study of the hardness of learning feedforward nueral networks has a rich history dating back many decades. Judd (1987); Blum & Rivest (1988) show that in the proper learning setting, it is NP complete to find a set of weights of a given network that fits a given training set. These results were later expanded in (Zhang et al., 2017) to more realistic settings. Many works study the hardness of improperly learning the class of ReLU feedforward networks. In

the SQ setting, Goel et al. (2018); Diakonikolas et al. (2020); Song et al. (2017) show that at least a superpolynomial number of queries are needed to learn even single hidden layer networks in the correlational statistical query model. Chen et al. (2022a) show that networks with two hidden layers are even hard in the general SQ setting by using the networks to round inputs to boolean inputs and applying hardness results from Boolean learning theory and cryptography. We should remark that there are also other papers that reduce the task of learning feedforward neural networks to average case or cryptographically hard problems (Song et al., 2021; Daniely & Vardi, 2020). To sidestep these hardness results and provide proofs of learnability in the PAC setting, a number of works make assumptions on the networks or inputs/outputs to give efficient PAC learning algorithms for single hidden layer neural networks (Bakshi et al., 2019; Goel et al., 2018; Shamir, 2018; Sedghi et al., 2016; Vempala & Wilmes, 2019). Such assumptions include bounds on the condition number (Bakshi et al., 2019), approximation guarantees of the network by polynomials (Vempala & Wilmes, 2019), positivity of the second layer weights Diakonikolas et al. (2020), and others. In perhaps the most general setting, Chen & Narayanan (2023); Chen et al. (2023) give a polynomial time learning algorithm for learning single hidden layer ReLU networks with $O(1)$ hidden nodes (i.e. runtime is exponential in the number of nodes but polynomial in other parameters such as the input size and error).

**Random neural network learnability**    Though the class of neural networks may be challenging to learn, various works study the average case setting where one is interested in learning random neural networks or random features from neural networks. In fact, prior work has shown that the complexity of such random neural networks, measured via statistical measures, continuity, or robustness, is rather low though this is by no means a guarantee of learnability (Kalimeris et al., 2019; De Palma et al., 2019; Shah et al., 2020; Valle-Perez et al., 2018). Various hardness results exist for this setting of learning random neural networks. Das et al. (2019) gives SQ hardness results for learning neural networks with sign activation that shows query complexity increasing exponentially with the depth of the network. Daniely & Vardi (2020) show that neural networks with random weights are hard to learn if the input distribution is allowed to depend on the weights based on reductions to random K-SAT. Nevertheless, recent work by Daniely et al. (2023) shows that random constant depth neural networks with ReLU activation can be learned in error $\epsilon$ in time scaling as $d^{\mathrm{polylog}(1/\epsilon)}$ where $d$ is the size of the network.

**Sample complexity and generalization bounds**    Separate from the question of computational complexity of learning is the question of how many samples are needed to learn a function class. Specific to (exactly or approximately) invariant neural network architectures, there exist papers studying the sample complexity of learning convolutional neural networks (Du et al., 2018b; Zhou & Feng, 2018; Cao & Gu, 2019) and graph neural networks (Zhang et al., 2020).

Various papers consider how generalization bounds improve when enforcing equivariance. These include generalization bounds based on covering numbers or the complexity of the invariant function space Sokolic et al. (2017); Petrache & Trivedi (2023), based on invariant kernel method algorithms Bietti et al. (2021); Tahmasebi & Jegelka (2023); Elesedy (2021), and based on equivariant versions of norm-based PAC-Bayesian generalization bounds (Behboodi et al., 2022). For the group of translations, there also exist papers studying generalization bounds for architectures with convolutional layers though these architectures are not strictly invariant (Long & Sedghi, 2019; Zhou & Feng, 2018). Similarly, there are various generalization bounds as well for the class of graph neural networks (Lv, 2021; Verma & Zhang, 2019).

**Other related topics**    Though not directly a statement of computational hardness, various papers give no-go theorems for learning function classes by studying expressivity limitations of graph neural networks (Jegelka, 2022; Wu et al., 2020; Loukas, 2019). These expressivity results provide a set of functions that a graph neural network cannot express and thus by definition also cannot learn to arbitrary accuracy. The most well known set of results are related to limitations of graph neural networks in distinguishing non-isomorphic graphs via the Weisfeiler-Lehman hierarchy (Xu et al., 2018; Morris et al., 2019; Geerts & Reutter, 2022). Other work show expressivity limitations of GNNs by studying their power in expressing invariant polynomials (Puny et al., 2023), identifying graph biconnectivity (Zhang et al., 2023), counting substructures (Chen et al., 2020), and through various other means.

Various works study the implicit bias of neural networks primarily to help address why neural networks can even learn in overparameterized settings (Vardi, 2023; Chizat & Bach, 2020; Ji & Telgarsky, 2020). Here, the aim of these works is to show that gradient descent will converge to specific regularized functions that fit a given training set. Implicit bias results have been proven for linear but multi-layer classes of convolutional neural networks (Gunasekar et al., 2018; Yun et al., 2020; Le & Jegelka, 2022) and equivariant neural networks (Lawrence et al., 2021; Chen & Zhu, 2023). Generally, these results show that gradient descent on the parameter space of such neural networks is implicitly regularized under some norm or semi-norm depending on properties of the group. Extending these results to more realistic settings is likely challenging.

In Boolean learning theory, there are various results on the problem of learning a symmetric $k$-junta (Mossel et al., 2003; Lipton et al., 2005; Kolountzakis et al., 2005). Here, we are promised that the Boolean function only depends on $k$ variables, and within that $k$ variable subset, the function is symmetric and thus only depends on the number of $1$s in that subset. For this problem, Kolountzakis et al. (2005) achieve a runtime $n^{O(k \log k)}$ that is polynomial in $n$ for fixed $k$. Note, that the functions we study in the Boolean setting are assumed to be symmetric over the whole input space and not just on a size $k$ subset of the inputs bits which drastically simplifies the problem. In fact, the hardness of the symmetric $k$-junta learning problem largely arises from challenges in finding which $k$ bits are in the support of the Boolean function.

Finally, in quantum machine learning, there has been a recent interest in studying quantum variational algorithms or quantum neural networks that are constrained under symmetries (Meyer et al., 2023; Ragone et al., 2022; Skolik et al., 2023; Cong et al., 2019). Some recent theoretical work has quantified the hardness and sample complexity associated with these models that are generally linear though acting on high dimensional Hilbert spaces (Pesah et al., 2021; Schatzki et al., 2022; Anschuetz & Kiani, 2022; Nguyen et al., 2022; Anschuetz et al., 2022). Since these quantum models typically operate in a different regime than classical models and are typically focused on quantum tasks, they are out of the scope of our current work.

## B  EXTENDED BACKGROUND INTO LEARNING FRAMEWORKS

There are various models used to rigorously quantify the hardness of learning. We recommend the review in Reyzin (2020) for an overview of the SQ formalism and its connections to other learning models. Here, we briefly mention the PAC model and overview further the background into the SQ framework from the main text.

Perhaps the most widely used framework for quantifying hardness of learning is the provably approximately correct (PAC) model (Valiant, 1984).

**Definition 10** (PAC Learning (Mohri et al., 2018)). A concept class $\mathcal{C}$ is PAC-learnable if there exists an algorithm such that for any $\epsilon > 0$ and $\delta > 0$, for all distributions $\mathcal{D}$ on $\mathcal{X}$, and for any target concept $c^* \in \mathcal{C}$, the algorithm takes at most $m = \text{poly}(1/\epsilon, 1/\delta, n)$ samples drawn i.i.d. from $(\boldsymbol{x}, c(\boldsymbol{x}))$ with $\boldsymbol{x} \sim \mathcal{D}$, and returns a function $f$ satisfying $\|f - c^*\|_{\mathcal{D}} \leq \epsilon$ with probability at most $1 - \delta$. If the algorithm runs in time at most $\text{poly}(1/\epsilon, 1/\delta, n)$, then $\mathcal{C}$ is efficiently PAC learnable. The algorithm is a proper learner if it returns $f \in \mathcal{C}$, and otherwise denoted an improper learner.

Proving computational hardness in the PAC model typically requires reducing learning tasks to cryptographically or complexity theoretically hard problems. Unconditional hardness results for learning with neural networks would imply P $\neq$ NP: Abbe & Sandon (2023) show that the task of training neural networks via gradient descent is P-Complete, i.e. any poly-time algorithm in P can be reduced to the task of training a given poly-sized neural network with stochastic gradient descent.

Given the above, the statistical query formalism sidesteps these challenges by constraining algorithms to be composed of a set of noisy queries. Perhaps the most important restriction in the SQ framework is that algorithms do not have access to individual data points but only noisy queries averaged across the input/output distribution. The classic example of an efficient non-SQ algorithm is Gaussian elimination for learning parities (Blum et al., 1994). Nevertheless, the SQ formalism naturally captures most algorithms used in practice including gradient descent as shown for example in Example 1 of the main text for gradients over the mean squared error loss.

For boolean functions, correlational statistical queries can capture any general statistical query since any query can be split into its $+1$ or $-1$ output cases (Reyzin, 2020). For example, assume one performs an SQ query with query function $q : \mathcal{X} \times \{-1, +1\} \to [-1, +1]$, then

$$\mathbb{E}_{(\boldsymbol{x},y)\sim\mathcal{D}} \left[ g(\boldsymbol{x}, y) \right] = \mathbb{E} \left[ g(\boldsymbol{x}, +1) \frac{1+y}{2} + g(\boldsymbol{x}, -1) \frac{1-y}{2} \right], \tag{20}$$

which decomposes into two correlational statistical queries and two functions independent of the target.

We remarked in the main text that generalizing real-valued hardness results proven in the CSQ framework to the SQ framework is in general challenging. In fact, Vempala & Wilmes (2019) more rigorously underscored this challenge (see their Proposition 4.1) by providing a rather unnatural, yet efficient, SQ algorithm that learns any finite real-valued function class satisfying a non-degeneracy assumption that the set of points $f(x) = g(x)$ is measure zero for any pair of functions $f, g$ in the class. Namely, they show the following no-go theorem.

**Theorem 11** (Limitations on SQ hardness for real-valued functions (Vempala & Wilmes, 2019)). *Given a concept class $\mathcal{C}$ of functions $f : \mathcal{X} \to \mathbb{R}$ of size $|\mathcal{C}|$ where for all $f, g \in \mathcal{C}$ the set of inputs where $f(x) = g(x)$ is measure zero, the family $\mathcal{C}$ can be learned with $O(\log |\mathcal{C}|)$ queries to SQ with a constant error tolerance $\tau$.*

*Proof.* We aim to choose a query $g : \mathcal{C} \to [-1, 1]$ which maps the target function $f \in \mathcal{C}$ to a unique value in the range $[-1, 1]$. For our target function $f \in \mathcal{C}$, let us call this value $v$. To specify the form of this query, it suffices to send each pair $(x, f(x))$ to its corresponding unique value in $[-1, +1]$, i.e. output $v$ for every input $(x, f(x))$. Since outputs of disjoint functions in $\mathcal{C}$ are equal only on a support of measure zero, this mapping can be performed with probability one. Thus, we can query the value $v$ and receive an output, say $v'$ up to error $\tau$. Repeating this procedure iteratively to only include functions in the tolerance range $[v' - \tau, v' + \tau]$, we can select the target function after $O(\log |\mathcal{C}|)$ queries. $\square$

Loosely, the above theorem shows that real-valued classes of functions are only hard in SQ settings when there exist sufficiently many hard functions with finitely many outputs. Chen et al. (2022a) achieved such a result for two-hidden layer networks by constructing networks that round inputs to the nearest boolean bitstring.

## C  BOOLEAN STATISTICAL QUERY SETTING

As discussed in Sec. 3, the Boolean function setting provides a nice illustration of the types of results obtainable via SQ lower bounds. For further background into the boolean SQ setting, we recommend the review in Reyzin (2020). Here, we will prove Theorem 2 restated below for convenience.

**Theorem 2** (Boolean SQ hardness). *For a given symmetry group $G$ with representation $\rho : G \to GL(\{-1, +1\}^n)$, let $\|p_{\mathcal{O}_\rho}\| := \left( \sum_{O_k \in \mathcal{O}_\rho} \left( \frac{|O_k|}{2^n} \right)^2 \right)^{1/2}$ and let $\mathcal{H}_\rho$ be the class of symmetric Boolean functions, defined as*

$$\mathcal{H}_\rho = \{f : \{-1, +1\}^n \to \{-1, +1\} : \; \forall g \in G, \forall \boldsymbol{x} \in \{-1, +1\}^n : f(\rho(g) \cdot \boldsymbol{x}) = f(\boldsymbol{x})\}. \tag{4}$$

*Any SQ learner capable of learning $\mathcal{H}_\rho$ up to sufficiently small classification error probability $\epsilon$ ($\epsilon < 1/4$ suffices) with queries of tolerance $\tau$ requires at least $\tau^2 \|p_{\mathcal{O}_\rho}\|^{-2}/2$ queries.*

For a simple proof of the above statement, we will follow the proof technique in Appendix B of Chen et al. (2022a). The above statement will follow as a consequence of the pairwise independence of functions in $\mathcal{H}_\rho$ defined below.

**Definition 12** (Boolean Case of Definition C.1 of Chen et al. (2022a)). *Let $\mathcal{C}$ be a hypothesis class mapping $\{-1, +1\}^n$ to $\{-1, +1\}$, and let $\mathcal{D}$ be a distribution on $\mathcal{X}$. $\mathcal{C}$ is a $(1-\eta)$-pairwise independent function family if with probability $1-\eta$ over the choice of $\boldsymbol{x}, \boldsymbol{x}'$ drawn independently from $\mathcal{D}$, the distribution of $(f(\boldsymbol{x}), f(\boldsymbol{x}'))$ for $f$ drawn uniformly at random from $\mathcal{C}$ is the product distribution $\mathrm{Unif}(\{-1, +1\}^2)$.*

Pairwise independent function families offer a convenient means to bound the number of queries needed to distinguish the outputs of a given unknown function $f^*$ versus uniformly random outputs from the function class.

**Theorem 13** (Theorem C.4 of Chen et al. (2022a)). *Let function class $\mathcal{C}$ mapping $\{-1, +1\}^n$ to $\{-1, +1\}$ be a $(1 - \eta)$-pairwise independent function family w.r.t. a distribution $\mathcal{D}$ on $\{-1, +1\}^n$. For any $f \in \mathcal{C}$, let $\mathcal{D}_f$ denote the distribution of $(\boldsymbol{x}, f(\boldsymbol{x}))$ where $\boldsymbol{x} \sim \mathcal{D}$. Let $\mathcal{D}_{\mathrm{Unif}(\mathcal{C})}$ denote the distribution of $(\boldsymbol{x}, y)$ where $\boldsymbol{x} \sim \mathcal{D}$ and $y = f(\boldsymbol{x})$ for $f \sim \mathrm{Unif}(\mathcal{C})$. Any SQ learner able to distinguish the labeled distribution $\mathcal{D}_{f^*}$ for an unknown $f^* \in \mathcal{C}$ from the randomly labeled distribution $\mathcal{D}_{\mathrm{Unif}(\mathcal{C})}$ using bounded queries of tolerance $\tau$ requires at least $\frac{\tau^2}{2\eta}$ such queries.*

Finally, we are ready to prove Theorem 2.

*Proof of Theorem 2.* Since the symmetric function is constant on orbits, we can represent any symmetric function as $f : \mathcal{O}_n \to \{-1, +1\}$ and the correlation between two functions $f, g$ can be written as

$$\langle f, g \rangle_{SQ} = \mathbb{E}[fg] = 2^{-n} \sum_{O_k \in \mathcal{O}_\rho} |O_k| f(\boldsymbol{x}_{O_k}) g(\boldsymbol{x}_{O_k}). \tag{21}$$

The probability $p_{\mathcal{O}_\rho} : \mathcal{O}_\rho \to [0, 1]$ of a uniformly random bitstring falling in orbit $O_K \in \mathcal{O}_\rho$ is equal to $p_{\mathcal{O}_\rho}(O_k) = |O_k|/2^n$. Note that the distribution of $(f(\boldsymbol{x}_O), f(\boldsymbol{x}_{O'}))$ is equal to the uniform distribution whenever $\boldsymbol{x}_O \neq \boldsymbol{x}_{O'}$ since we have a uniform distribution over all possible symmetric functions. This occurs with probability

$$\eta = \sum_{O_k \in \mathcal{O}_\rho} \left( \frac{|O_k|}{2^n} \right)^2 = \|p_{\mathcal{O}_\rho}\|^2. \tag{22}$$

Therefore, $\mathcal{H}_\rho$ is a $(1 - \eta)$-pairwise independent function family with parameter $\eta$ as above.

To apply Theorem 13, note that for a given function $f^* \in \mathcal{H}_\rho$, to distinguish $\mathcal{D}_{f^*}$ from $\mathcal{D}_{\mathrm{Unif}(\mathcal{H}_\rho)}$, it suffices to return a function $f$ that has classification error at most $1/4$ with respect to $f^*$. Since distinguishing the two distributions requires $\frac{\tau^2}{2\eta}$ queries, we arrive at the final result. $\qquad \square$

## D   COMPLEXITY THEORETIC HARDNESS EXTENSION

In this section, we would like to show a reduction from the task of training an invariant neural network on a given dataset to a previously NP hard problem. This extends the classic result of Blum & Rivest (1988) showing that there exists a sequence of datasets such that determining whether or not a 3-node neural network can fit that dataset is hard. Though we consider extending this to a simple graph neural network here for simplicity, we remark that we chose this only for convenience and similar extensions can be found for other classes of networks.

The particular NP hard problem we consider is the decision version of the learning halfspaces with noise problem (Guruswami & Raghavendra, 2009; Feldman et al., 2012).

**Problem 1** (Learning halfspaces with noise (Guruswami & Raghavendra, 2009)). Given a set of $N$ labeled examples $\{\boldsymbol{x}_i, y_i\}_{i=1}^N$ where $\boldsymbol{x}_i \in \{0, 1\}^n$ and $y_i \in \{-1, +1\}$ and constants $\delta, \epsilon$ where $0 < \delta \leq \epsilon < 1$, we distinguish between the following two cases:

- **Case 1:** There exists a vector $\boldsymbol{v} \in \mathbb{R}^n$, constant $\theta \in \mathbb{R}$, and set $S \subseteq [N]$ of size at least $|S| \geq (1 - \delta)N$ such that $y_i(\langle \boldsymbol{v}, \boldsymbol{x}_i \rangle - \theta) \geq 0$ for all $i \in S$.

- **Case 2:** For all $\boldsymbol{v} \in \mathbb{R}^n$, $\theta \in \mathbb{R}$, and sets $S \subseteq [N]$ such that $|S| \geq (1 - \epsilon)N$, there exists some $i \in S$ such that $y_i(\langle \boldsymbol{v}, \boldsymbol{x}_i \rangle - \theta) < 0$.

Consider the following class of message passing GNNs with a single hidden layer and $k$ channels which acts on an adjacency matrix $\boldsymbol{A} \in \{0, 1\}^{n \times n}$ and node features $\boldsymbol{x} \in \mathbb{R}^n$:

$$f_{\boldsymbol{a}, \boldsymbol{b}, c}(\boldsymbol{x}, \boldsymbol{A}) = c + \sum_{i=1}^k \boldsymbol{1}^\top \mathrm{relu}(\boldsymbol{x} + a_i \boldsymbol{A} \boldsymbol{x}) b_i. \tag{23}$$

We will map the below problem of fitting the weights $\boldsymbol{a}, \boldsymbol{b}, c$ of the GNNs above to a given dataset consisting of a set of graphs and their corresponding outputs.

**Problem 2.** Given a labeled set of graphs, node features, and outputs $\{\boldsymbol{A}_i, \boldsymbol{x}_i, y_i\}_{i=1}^{N}$ where $\boldsymbol{A}_i \in \{0,1\}^{n \times n}$, $\boldsymbol{x}_i \in \mathbb{R}^n$ and $y_i \in \{-1, +1\}$ and constants $\delta, \epsilon$ where $0 < \delta \leq \epsilon < 1$, distinguish between the following two cases:

- **Case 1:** There exists weights $a_1, \ldots, a_k, b_1, \ldots, b_k, c \in \mathbb{R}$ and set $S \subseteq [N]$ of size at least $|S| \geq (1 - \delta)N$ such that $y_i f_{\boldsymbol{a}, \boldsymbol{b}, c}(\boldsymbol{x}_i, \boldsymbol{A}_i) \geq 0$ for all $i \in S$.

- **Case 2:** For all weights $a_1, \ldots, a_k, b_1, \ldots, b_k, c \in \mathbb{R}$, and sets $S \subseteq [N]$ such that $|S| \geq (1 - \epsilon)N$, there exists some $i \in S$ such that $y_i f_{\boldsymbol{a}, \boldsymbol{b}, c}(\boldsymbol{x}_i, \boldsymbol{A}_i) < 0$.

**Proposition 14** (NP hardness of GNN training). *Training the GNN in Eq.* (23) *to solve Problem 2 is* NP *hard.*

*Proof.* We show that solving Problem 2 is equivalent to solving Problem 1 which is previously shown to be NP hard (Guruswami & Raghavendra, 2009; Feldman et al., 2012). Given a dataset $\{\boldsymbol{x}_i, y_i\}_{i=1}^{N}$ for Problem 1 over $n$-bit Boolean strings, we map this to a dataset over graphs $\{\boldsymbol{A}_i, \boldsymbol{x}_i', y_i'\}_{i=1}^{N}$ of $\sum_{k=1}^{n} k = n(n+1)/2$ nodes where $\boldsymbol{x}_i' = -\mathbf{1}$ for all inputs and $y_i = y_i'$ are the same for both datasets. Adjacency matrix $\boldsymbol{A}_i$ is constructed by including a disconnected $k$-clique (including self loops) in the graph whenever $[\boldsymbol{x}_i]_k = 1$. Since there are $n(n+1)/2$ nodes, there are sufficient nodes to construct any such graph. E.g., for $\boldsymbol{x} = (1, 0, 1)^\top$, a corresponding adjacency matrix is

$$\boldsymbol{A} = \begin{bmatrix} 1 & 0 & 0 & 0 & 0 & 0 \\ 0 & 0 & 0 & 0 & 0 & 0 \\ 0 & 0 & 0 & 0 & 0 & 0 \\ 0 & 0 & 0 & 1 & 1 & 1 \\ 0 & 0 & 0 & 1 & 1 & 1 \\ 0 & 0 & 0 & 1 & 1 & 1 \end{bmatrix}. \tag{24}$$

We also assume that the GNNs have $k = n$ channels. In this setting, we have that for any $i \in [N]$

$$f_{\boldsymbol{a}, \boldsymbol{b}, c}(\boldsymbol{x}_i', \boldsymbol{A}_i) = c + \sum_{j=1}^{n} \mathbf{1}^\top \operatorname{relu}(\boldsymbol{x}_i' + a_j \boldsymbol{A}_i \boldsymbol{x}_i') b_j$$

$$= c + \sum_{j=1}^{n} b_j \sum_{\ell=1}^{n} \ell \mathbb{1}[\exists\, \ell - \text{clique} \in \boldsymbol{A}_i] \operatorname{relu}(-1 - \ell a_j), \tag{25}$$

where $\mathbb{1}[\exists\, \ell - \text{clique} \in \boldsymbol{A}_i]$ is the indicator function for whether or not a $\ell$-clique is in the graph $\boldsymbol{A}_i$. The above is clearly a linear function in the vector $\boldsymbol{r}_i \in \{0,1\}^n$ where $[\boldsymbol{r}_i]_\ell = \mathbb{1}[\exists\, \ell - \text{clique} \in \boldsymbol{A}_i]$. Furthermore, the graph is chosen so that $\boldsymbol{r}_i = \boldsymbol{x}_i$. Thus, any choice of weights $\boldsymbol{a}, \boldsymbol{b}, c$ corresponds to a halfspace for $\boldsymbol{x}_i$.

It only remains to be shown that any such halfspace can be written in the form of Eq. (25). Note that

$$f_{\boldsymbol{a}, \boldsymbol{b}, c}(\boldsymbol{x}_i, \boldsymbol{A}_i) = c' + \begin{bmatrix} b_1 & b_2 & \cdots & b_n \end{bmatrix} \boldsymbol{M} \boldsymbol{r}_i, \tag{26}$$

where $c'$ is an updated constant and

$$\boldsymbol{M} = \begin{bmatrix} \operatorname{relu}(-1 - a_1) & 2\operatorname{relu}(-1 - 2a_1) & \cdots & n\operatorname{relu}(-1 - na_1) \\ \operatorname{relu}(-1 - a_2) & 2\operatorname{relu}(-1 - 2a_2) & \cdots & n\operatorname{relu}(-1 - na_2) \\ \vdots & \vdots & \ddots & \vdots \\ \operatorname{relu}(-1 - a_n) & 2\operatorname{relu}(-1 - 2a_n) & \cdots & n\operatorname{relu}(-1 - na_n) \end{bmatrix}. \tag{27}$$

Setting $a_k = (1/2 - k)^{-1}$ makes the above matrix have non-zero entries on the diagonal and every entry above the diagonal. Such a matrix is invertible by Gaussian elimination so every possible halfspace can be constructed by proper choice of the weights $b_1, \ldots, b_n, c'$. $\quad\square$

# E GNN HARDNESS PROOFS

## E.1 LOWER BOUND FOR ERDŐS–RÉNYI DISTRIBUTED GRAPHS

Here, we will consider a model over graphs of $n$ nodes where adjacency matrices $\boldsymbol{A} \sim \text{Unif}(\{0,1\}^{n \times n})$ are drawn uniformly from all possible directed graphs. This distribution can be viewed as an Erdős–Rényi $G(n, p)$ over directed graphs including self edges with $p = 0.5$. Node features are fixed in this model to $\boldsymbol{x} \in \mathbb{R}^n$ where we set $\boldsymbol{x} = \mathbf{1}$ in all instances (i.e., node features are trivial). Note, that we consider Boolean functions here w.l.o.g. as $\{0, 1\}$ valued as opposed to $\{-1, +1\}$ valued as in previous sections since using $\{0, 1\}$ valued functions in this section makes the story easier to follow.

Our aim is to prove Theorem 3 copied below.

**Theorem 3** (SQ hardness of $\mathcal{H}_{ER,n}$). *Any SQ learner capable of learning $\mathcal{H}_{ER,n}$ up to classification error probability $\epsilon$ sufficiently small ($\epsilon < 1/4$ suffices) with queries of tolerance $\tau$ requires at least $\Omega\left(\tau^2 \exp(n^{\Omega(1)})\right)$ queries.*

The function class $\mathcal{H}_{ER,n}$ that achieves the desired lower bound are Boolean valued functions that depend on the counts of the degrees or number of neighbors of the nodes. To describe this class, we first introduce some notation. For a given graph $\boldsymbol{A} \in \{0, 1\}^{n \times n}$, let $\boldsymbol{d_A} \in [n]^n$ be a vector with entry $[\boldsymbol{d_A}]_i$ equal to the number of outgoing edges (or degree) for node $i$. From here, let $\boldsymbol{c_A} \in [n]^{n+1}$ denote the count of the degrees where entry $[\boldsymbol{c_A}]_i$ counts the number of nodes that have $i - 1$ outgoing edges in $\boldsymbol{A}$ or more formally

$$[\boldsymbol{c_A}]_i = \sum_{k=1}^{n} \mathbb{1}\left[[\boldsymbol{d_A}]_k = i - 1\right]. \tag{28}$$

Note that $\boldsymbol{c_A}$ is a length $n + 1$ vector since the number of outgoing edges can be anything from $\{0, 1, \ldots, n\}$. Also note that this vector is an invariant vector as permuting the nodes leaves the counts invariant. Given a subset $S \subseteq [n + 1]$ and a bit $b \in \{0, 1\}$, the functions we aim to learn take the form

$$g_{S,b}(\boldsymbol{A}) = b + \sum_{i \in S}[\boldsymbol{c_A}]_i \mod 2, \tag{29}$$

which are a sort of parity function supported over the elements of the degree counts $\boldsymbol{c_A}$ in $\mathcal{S}$. Therefore, our hypothesis class $\mathcal{H}$ consists of $2^{n+2}$ functions:

$$\mathcal{H}_{ER,n} = \{g_{S,b} : S \subseteq [n+1], b \in \{0,1\}\}. \tag{30}$$

First, we will show below that the class $\mathcal{H}_{ER,n}$ is exponentially hard to learn in the SQ setting. Later in Lemma 15, we show that the functions in $\mathcal{H}$ can be constructed by the GNN architecture thus completing our proof.

*Proof of Theorem 3.* To simplify the function $g_{S,b}$ even further, let $\widehat{\boldsymbol{c}}_{\boldsymbol{A}} \in \{0, 1\}^{n+1}$ be a boolean valued version of $\boldsymbol{c_A}$ where entry $[\widehat{\boldsymbol{c}}_{\boldsymbol{A}}]_i = [\boldsymbol{c_A}]_i \mod 2$. Then the function $g_{S,b}$ is equivalently a parity function in $\widehat{\boldsymbol{c}}_{\boldsymbol{A}}$:

$$g_{S,b}(\boldsymbol{A}) = b + \sum_{i \in S}[\boldsymbol{c_A}]_i \mod 2 = (-1)^b \prod_{i \in S}(-1)^{[\widehat{\boldsymbol{c}}_{\boldsymbol{A}}]_i}. \tag{31}$$

We will prove an SQ lower bound using Theorem 13 based on the $(1 - \eta)$-pairwise independence property of the function class $\mathcal{H}_{ER,n}$ in Definition 12. Namely, note that for two graphs $\boldsymbol{A}, \boldsymbol{A}' \in \{0, 1\}^{n \times n}$, if $\widehat{\boldsymbol{c}}_{\boldsymbol{A}} \neq \widehat{\boldsymbol{c}}_{\boldsymbol{A}'}$, then the distribution of $(g_{S,b}(\boldsymbol{A}), g_{S,b}(\boldsymbol{A}'))$ for $g_{S,b}$ drawn uniformly from $\mathcal{H}_{ER,n}$ is equal to the product distribution $\text{Unif}(\{0, 1\}^2)$. This follows from the fact that $g_{S,b}(\boldsymbol{A}) = (-1)^b \prod_{i \in S}(-1)^{[\widehat{\boldsymbol{c}}_{\boldsymbol{A}}]_i}$ is a parity function of the entries in $\widehat{\boldsymbol{c}}_{\boldsymbol{A}}$. Since $\mathcal{H}_{ER,n}$ spans uniformly over all possible parity functions in $\{0, 1\}^{n+1}$, the pairwise independence properties follows from this whenever $\widehat{\boldsymbol{c}}_{\boldsymbol{A}} \neq \widehat{\boldsymbol{c}}_{\boldsymbol{A}'}$.

Thus, the parameter $1 - \eta$ in the pairwise independence is equal to the probability that $\widehat{\boldsymbol{c}}_{\boldsymbol{A}} \neq \widehat{\boldsymbol{c}}_{\boldsymbol{A}'}$. We will establish that this probability is at least $1 - O(\exp(-n^{\Omega(1)}))$ at the end of this proof thus

guaranteeing $\eta^{-1} = \Omega\left(\exp(n^{\Omega(1)})\right)$. Once this is established, we can apply Theorem 13 and note that for a given function $f^* \in \mathcal{H}_{ER,n}$, to distinguish $\mathcal{D}_{f^*}$ from $\mathcal{D}_{\text{Unif}(\mathcal{H}_\rho)}$ as in Theorem 13, it suffices to return a function $f$ that has classification error at most $1/4$ with respect to $f^*$. Since distinguishing the two distributions requires $\frac{\tau^2}{2\eta}$ queries, we arrive at the final result by noting that $\eta^{-1} = \Omega\left(\exp(n^{\Omega(1)})\right)$.

To finish this proof, we must establish that

$$\mathbb{P}_{\boldsymbol{A} \sim \text{Unif}(\{0,1\}^{n \times n})}[\widehat{\boldsymbol{c}}_{\boldsymbol{A}} \neq \widehat{\boldsymbol{c}}_{\boldsymbol{A}'}] \geq 1 - O(\exp(-n^{\Omega(1)})). \tag{32}$$

We will show this result by proving that for some constant $C > 0$ and $p < 1/2$, for large enough $n$, there are at least $Cn^p$ entries of $\widehat{\boldsymbol{c}}_{\boldsymbol{A}'}$ such that the probability of those entry being even or odd is at most $1/2 + o(1)$. To continue, let us denote this region

$$\Delta_p = \{0, 1, \ldots, n\} \cap [n/2 - Cn^p, n/2 + Cn^p], \quad p < 1/2, \tag{33}$$

as the set of integers within the range $[n/2 - Cn^p, n/2 + Cn^p]$. This range is chosen to coincide with the largest entries of $\boldsymbol{c}_{\boldsymbol{A}}$.

The degree of any node is independently distributed as binomial with distribution $\text{bin}(n, 0.5)$. To asymptotically expand $i$ around the peak of the binomial coefficient, let us index $i = n/2 + r$ for given $r$ such that $i \in \Delta_p$. Therefore, for any such $n/2 + r = i \in \Delta_p$, we have that any given node $j \in [n]$ has degree $\boldsymbol{d}_k = i$ with probability

$$\mathbb{P}[\boldsymbol{d}_k = i] = 2^{-n}\binom{n}{n/2 + r} = \frac{2}{\sqrt{\pi n}}(1 + O(1/n)). \tag{34}$$

In the above, we use the asymptotic approximation (Spencer, 2014) for the binomial coefficient around its peak $\binom{n}{n/2}$ of

$$\binom{n}{\frac{n+r}{2}} = (1 + O(1/n))\sqrt{\frac{2}{\pi n}} 2^n \exp\left(-\frac{r^2}{2n} + O(r^3/n^2)\right). \tag{35}$$

Thus, the marginal distribution of any entry $i \in \Delta_p$ of $[\boldsymbol{c}_{\boldsymbol{A}}]_i \sim \text{bin}\left(n, \frac{2}{\sqrt{\pi n}}(1 + O(1/n))\right)$ is also binomial distributed with probability converging to $\frac{2}{\sqrt{\pi n}}$. For entry $i \in \Delta_p$, this implies that $\mathbb{E}[\boldsymbol{c}_{\boldsymbol{A}}]_i = O(\sqrt{n})$. From the Chernoff concentration bound for the binomial distribution, we also have that for any $i \in \Delta_p$ and for any $\delta > 0$:

$$\mathbb{P}\left[|[\boldsymbol{c}_{\boldsymbol{A}}]_i - \mathbb{E}[\boldsymbol{c}_{\boldsymbol{A}}]_i| \geq n^{1/4+\delta}\right] \leq 2\exp\left(-\frac{n^{1/2+2\delta}(\mathbb{E}[\boldsymbol{c}_{\boldsymbol{A}}]_i)^{-1}}{2}\right) = 2\exp\left(-\Omega(n^{2\delta})\right). \tag{36}$$

By a union bound over all $i \in \Delta_p$, we therefore have that with exponentially high probability, $|[\boldsymbol{c}_{\boldsymbol{A}}]_i - \mathbb{E}[\boldsymbol{c}_{\boldsymbol{A}}]_i| \leq n^{1/4+\delta}$ for all $i \in \Delta_p$.

Summarizing the previous results, we have that there are $2Cn^p$ entries of $[\boldsymbol{c}_{\boldsymbol{A}}]$ where for any entry $i \in \Delta_p$, it holds that for some arbitrarily small $\delta > 0$

$$\frac{2}{\sqrt{\pi}}\sqrt{n}(1 + O(1/n)) - n^{1/4+\delta} \leq [\boldsymbol{c}_{\boldsymbol{A}}]_i \leq \frac{2}{\sqrt{\pi}}\sqrt{n}(1 + O(1/n)) + n^{1/4+\delta}. \tag{37}$$

Since the probability that $[\boldsymbol{c}_{\boldsymbol{A}}]_i$ is equal to any number $m$ is at most $O(n^{-1/4})$, the probability that $[\boldsymbol{c}_{\boldsymbol{A}}]_i$ is odd is at most $1/2 + O(n^{-1/4})$. We now aim to show that conditioned on other entries of $\boldsymbol{c}_{\boldsymbol{A}}$, the probability that a given entry of $\boldsymbol{c}_{\boldsymbol{A}}$ is odd or even is roughly independent of this conditioning.

Given a set of known values $S_{known} \subset \Delta_p$ equal to $[\boldsymbol{c}_{\boldsymbol{A}}]_k = m_k$ for $k \in S_{known}$, the marginal distribution of any entry of $[\boldsymbol{c}_{\boldsymbol{A}}]_j$ for $j \notin S_{known}$ conditioned on $[\boldsymbol{c}_{\boldsymbol{A}}]_k$ for $k \in S_{known}$ is also binomial. We now estimate the parameters of this binomial distribution. Given the convergence guarantee in Eq. (37), $\sum_{k \in S_{known}}[\boldsymbol{c}_{\boldsymbol{A}}]_k = O(|S_{known}|n^{1/2}) = O(n^{1/2+p})$ which is $o(n)$ since $p < 1/2$. Therefore for any set $S_{known} \subseteq \Delta_p$, there are at least $n - o(n)$ remaining nodes whose degree or number of outgoing edges are left to distribute. These remaining nodes cannot have degrees whose values fall in $S_{known}$, but note that since $|S_{known}| \leq O(n^p)$ and $p < 1/2$, there are

still $\Omega(\sqrt{n})$ buckets (values of the possible degrees) remaining each with probability proportional to $\Theta(1/\sqrt{n})$ by Eq. (35). Therefore, from the convergence guarantee of Eq. (37), we have

$$([c_A]_j \mid [c_A]_k = m_k, k \in S_{known}) \sim \text{bin}\left(n - o(n), \Theta(1/\sqrt{n})\right). \tag{38}$$

Thus, the conditional probability that $[c_A]_j$ is equal to any number $m$ is at most $O(n^{-1/4})$, and this implies that the probability that $[c_A]_j$ is odd is at most $1/2 + O(n^{-1/4})$ even when conditioned on entries in $S_{known}$.

Finally, let us calculate the desired probability that $\hat{c}_A = \hat{c}_{A'}$. For ease of notation, let us index the elements of $\Delta_p$ as $r_1, \ldots, r_N$. We have for large enough $n$ that

$$
\begin{aligned}
\mathbb{P}[\hat{c}_A = \hat{c}_{A'}] &\leq \mathbb{P}\left[[\hat{c}_A]_{r_1} = [\hat{c}_{A'}]_{r_1}, \ldots, [\hat{c}_A]_{r_N} = [\hat{c}_{A'}]_{r_N}\right] \\
&= \mathbb{P}\left[[\hat{c}_A]_{r_1} = [\hat{c}_{A'}]_{r_1}\right] \mathbb{P}\left[[\hat{c}_A]_{r_2} = [\hat{c}_{A'}]_{r_2} \mid [\hat{c}_A]_{r_1} = [\hat{c}_{A'}]_{r_1}\right] \times \\
&\quad \times \cdots \mathbb{P}\left[[\hat{c}_A]_{r_N} = [\hat{c}_{A'}]_{r_N} \mid [\hat{c}_A]_{r_1} = [\hat{c}_{A'}]_{r_1}, \ldots, [\hat{c}_A]_{r_{N-1}} = [\hat{c}_{A'}]_{r_{N-1}}\right] \\
&\leq \left(\frac{1}{2} + O(n^{-1/4})\right)^{|\Delta_p|} \\
&= O(\exp(-\Omega(|\Delta_p|))) \\
&= O(\exp(-\Omega(n^p))).
\end{aligned}
\tag{39}
$$

Since $\mathbb{P}_{A \sim \text{Unif}(\{0,1\}^{n \times n})}[\hat{c}_A \neq \hat{c}_{A'}] = 1 - \mathbb{P}_{A \sim \text{Unif}(\{0,1\}^{n \times n})}[\hat{c}_A = \hat{c}_{A'}]$, this completes the proof. $\qquad\square$

Finally, we need to show that the functions $g_{S,b}$ can be constructed as ReLU networks. Before proceeding to give this construction, we should note that it is known that any Boolean function that can be computed in time $O(T(n))$ can also be expressed by a neural network of size $O(T(n^2))$ (Parberry, 1994; Shamir, 2018). Our construction will be specific to the GNN class we consider and show that the boolean functions in $g_{S,b}$ are similarly efficiently constructible.

**Lemma 15** ($g_{S,b}$ as GNN). *For a GNN $f(\cdot)$ of the form of Eq. (5) with hidden layer widths equal to $k_1 = O(n)$ and $k_2 = O(n)$, there exist weights $a, b \in \mathbb{R}^{k_1}, u, v \in \mathbb{R}^{k_2}, W \in \mathbb{R}^{k_2 \times k_1}$ such that for any $S \subseteq [n+1], b \in \{0,1\}, g_{S,b}(A) = f(A; a, b, u, v, W)$ for all inputs $A \in \{0,1\}^{n \times n}$.*

*Proof.* Let us fix a given $S \subseteq [n+1], b \in \{0,1\}$. The function we want to represent is

$$g_{S,b}(A) = b + \sum_{i \in S} [c_A]_i \mod 2, \tag{40}$$

where as a reminder, $[c_A]_i$ counts the number of nodes that have $i - 1$ outgoing edges (see Eq. (28)).

For the construction, we will set $k_1 = k_2 = n + 1$. As a reminder, the two layers of the GNN take the form:

$$
\begin{aligned}
[f_{a,b}^{(1)}(A)]_i &= \mathbf{1}_n^\top \sigma\left(a_i + b_i A x\right) \quad \text{(output of channel } i \in [k_1]) \\
f_{u,v,W}^{(2)}(h) &= \sum_{i=1}^{k_2} u_i \sigma(\langle W_{:,i}, h \rangle + v_i).
\end{aligned}
\tag{41}
$$

Throughout this construction, the node features $x = \mathbf{1}$ are constant.

For the first layer, we will choose $a_i = 2 - i$ and $b_i = 1$ for all $i$. This first layer can only be a function of $d_A = A\mathbf{1}$ where entry $i$ is a count of the number of outgoing edges. Since entries in $d_A$ take values in $\{0, 1, \ldots, n\}$, we can equivalently write this first layer output as a function of the counts $c_A$. In fact, this first layer is linear as a function of $c_A$ and equal to

$$
h = M c_A = \begin{bmatrix} 1 & 2 & \cdots & n & n+1 \\ 0 & 1 & \cdots & n-1 & n \\ \vdots & \vdots & \ddots & \vdots & \vdots \\ 0 & 0 & \cdots & 1 & 2 \\ 0 & 0 & \cdots & 0 & 1 \end{bmatrix} c_A,
\tag{42}
$$

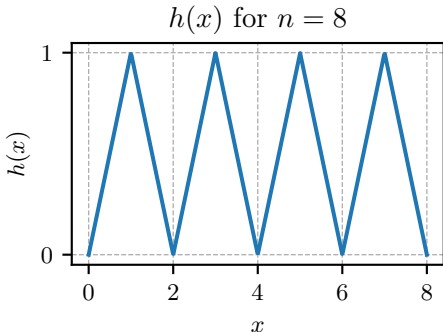

Figure 2: Sample form of function $h(x)$ used in constructing $g_{S,b}(\boldsymbol{A})$ as a GNN. For the construction, we will have $x = \sum_{i \in S} [\boldsymbol{c_A}]_i$.

where the matrix $\boldsymbol{M}$ is clearly full rank and invertible since it strictly positive along the diagonal and in entries above the diagonal.

Our aim in the second layer is to take the sum $b + \sum_{i \in S} [\boldsymbol{c_A}]_i$ and take its parity (i.e. $\mod 2$). To achieve this via sums of ReLUs $\sigma$ we can consider the function $h(\cdot)$ of the form below:

$$h(x) = \sigma(x+b) - 2\sigma(x+b-1) + 2\sigma(x+b-2) - \cdots + (-1)^n \sigma(x+b-n). \quad (43)$$

A sample version of this plot is in Fig. 2. Setting $x = b + \sum_{i \in S}[\boldsymbol{c_A}]_i$ achieves the end goal with $O(n)$ hidden nodes.

To provide actual values for the second layer parameters, let $\mathbf{1}_S \in {0,1}^{n+1}$ denote the vector where entry $i$ is equal to $1$ if $i \in S$ and equal to $0$ otherwise. Then, in the second layer, we have

- $\boldsymbol{W}_{:,i} = (\boldsymbol{M}^{-1})^\top \mathbf{1}_S$ for all columns $i \in [n+1]$

- $a_1 = 1$ and $a_i = 2(-1)^{i-1}$ for all $i \in \{2, \ldots, n+1\}$

- $v_i = b - i + 1$ for all $i \in [n+1]$

Using the values above, we have

$$f^{(2)}_{\boldsymbol{u},\boldsymbol{v},\boldsymbol{W}}(\boldsymbol{h}) = \sigma(\langle (\boldsymbol{M}^{-1})^\top \mathbf{1}_S, \boldsymbol{Mc_A}\rangle + b) + \sum_{i=2}^{k_2}(-1)^{i-1}2\sigma(\langle(\boldsymbol{M}^{-1})^\top \mathbf{1}_S, \boldsymbol{Mc_A}\rangle + b - i + 1)$$

$$= \sigma\left(b + \sum_{i \in S}[\boldsymbol{c_A}]_i\right) + \sum_{i=2}^{k_2}(-1)^{i-1}2\sigma\left(\sum_{i \in S}[\boldsymbol{c_A}]_i + b - i + 1\right)$$

$$= h\left([\boldsymbol{c_A}]_i\right).$$

$$(44)$$

As $\sum_{i \in S}[\boldsymbol{c_A}]_i$ only takes values in $\{0, 1, \ldots, n\}$, the above is supported on all of its inputs and this completes the proof. $\qquad \square$

### E.2 INVARIANT HERMITE POLYNOMIAL

**Multivariate Hermite polynomials**  To handle the graph setting with scaling feature dimension, we first extend the technical machinery related to Hermite polynomials to the invariant settings we consider.

Recall from Diakonikolas et al. (2020) that the $d$-variate normalized Hermite polynomial is defined for every multi-index $J = (J_1, \ldots, J_d) \in \mathbb{N}^d$ as:

$$H_J : \mathbb{R}^d \to \mathbb{R} : (x_1, x_2, \ldots, x_d) \mapsto \prod_{j=1}^{d} H_{J_j}(x_j), \qquad H_i : \mathbb{R} \to \mathbb{R} : x \mapsto \frac{H_{e_i}(x)}{\sqrt{i!}}, i \in \mathbb{N}, \quad (45)$$

where $H_{e_i}$ is the usual probabilist Hermite polynomial.

**Graph-invariant Hermite polynomial** Fix a graph $\mathcal{G}$ on $n$ vertices and let $\boldsymbol{A} := \boldsymbol{A}(\mathcal{G}) \in \mathbb{R}^{n \times n}$ be a graph shift operator (say the Laplacian). Define, for any multi-index $J \in \mathbb{N}^d$:

$$H_J^{\boldsymbol{A}} : \mathbb{R}^{n \times d} \to \mathbb{R} : \boldsymbol{X} \mapsto \frac{1}{\sqrt{n}} \sum_{v=1}^{n} H_J((\boldsymbol{A}\boldsymbol{X})_v). \tag{46}$$

We have the following orthogonality property:

**Lemma 16** (Orthogonality of invariant Hermite polynomial)**.** *For a fixed $\boldsymbol{A} \in \mathbb{R}^{n \times n}$ such that $\sum_{u=1}^{n} \boldsymbol{A}_{vu}^2 = 1$ for all $v \in [n]$, $\langle H_J^{\boldsymbol{A}}, H_{J'}^{\boldsymbol{A}} \rangle_{L^2(\mathcal{N})} = \delta_{J=J'} \cdot c_J$ for all $J, J' \in \mathbb{N}^d \backslash \{0\}$ for some $c_J \in \Theta(1)$ that depends only on $\boldsymbol{A}$.*

*Proof.* Let $\Sigma = \Sigma(\rho) = \begin{bmatrix} 1 & \rho \\ \rho & 1 \end{bmatrix}$ for some $\rho \in (-1, 1)$. We will also use Mehler's formula ([Kibble, 1945](#)) which states:

$$\frac{p_{\mathcal{N}(0,\Sigma)}(x,y)}{p_{\mathcal{N}}(x)p_{\mathcal{N}}(y)} = \sum_{m=0}^{\infty} \rho^m H_m(x) H_m(y), \tag{47}$$

where $p_{\mathcal{N}(0,\Sigma)}$ is the pdf of the bivariate normal distribution with mean 0 and covariance $\Sigma$ and $p_{\mathcal{N}}$ is the pdf of the univariate standard normal distribution.

We first show orthogonality of Hermite polynomials evaluated on dependent random variables. For all $i, j \in \mathbb{N}$:

$$\mathbb{E}_{(x,y) \sim \mathcal{N}_2(0,\Sigma)}[H_i(x)H_j(y)] = \int_{\mathbb{R}} \int_{\mathbb{R}} H_i(x)H_j(y) p_{\mathcal{N}(0,\Sigma)}(x,y) \mathrm{d}x\mathrm{d}y \tag{48}$$

$$= \int_{\mathbb{R}} \int_{\mathbb{R}} H_i(x)H_j(y) p_{\mathcal{N}}(x)p_{\mathcal{N}}(y) \cdot \frac{p_{\mathcal{N}(0,\Sigma)}(x,y)}{p_{\mathcal{N}}(x)p_{\mathcal{N}}(y)} \mathrm{d}x\mathrm{d}y \tag{49}$$

$$= \int_{\mathbb{R}} \int_{\mathbb{R}} H_i(x)H_j(y) \sum_{m=0}^{\infty} \rho^m H_m(x) H_m(y) \mathrm{d}P(x)\mathrm{d}P(y) \tag{50}$$

$$= \sum_{m=0}^{\infty} \rho^m \left( \int_{\mathbb{R}} H_i(x)H_m(x)\mathrm{d}P(x) \right) \left( \int_{\mathbb{R}} H_j(y)H_m(y)\mathrm{d}P(y) \right) \tag{51}$$

$$= \sum_{m=0}^{\infty} \rho^m \delta_{i=m} \delta_{j=m} \tag{52}$$

$$= \rho^i \delta_{i=j}. \tag{53}$$

We are now ready to present the proof of Lemma 16. Pick $J, J' \in \mathbb{N}^d \backslash \{0\}$. We have:

$$\langle H_J^{\boldsymbol{A}}, H_{J'}^{\boldsymbol{A}} \rangle_{L^2(\mathcal{N})} = \frac{1}{n} \sum_{v,v' \in [n]} \mathbb{E}_{\boldsymbol{X} \sim \mathcal{N}^{n \times d}}[H_J((\boldsymbol{A}\boldsymbol{X})_v) H_{J'}((\boldsymbol{A}\boldsymbol{X})_{v'})] \tag{54}$$

$$= \frac{1}{n} \sum_{v,v' \in [n]} \mathbb{E}_{\boldsymbol{Z} \sim P}[H_J(\boldsymbol{z}_v) H_{J'}(\boldsymbol{z}_{v'})], \tag{55}$$

where $\boldsymbol{z}_v$ is the $v$-th row of $\boldsymbol{Z} \in \mathbb{R}^{n \times d}$ and $P$ is a multivariate normal distribution over $\mathbb{R}^{n \times d}$ with mean 0 and covariance:

$$\mathrm{Cov}(\boldsymbol{Z}_{vj}, \boldsymbol{Z}_{v'j'}) = \begin{cases} 0, & \text{if } j \neq j' \\ \sum_{u=1}^{n} \boldsymbol{A}_{vu} \boldsymbol{A}_{v'u} =: \rho_{vv'}, & \text{otherwise.} \end{cases} \tag{56}$$

Furthermore, for any $v$ and any $j \neq j'$, $\boldsymbol{Z}_{vj}$ is independent of $\boldsymbol{Z}_{v'j'}$. This is because columns of $\boldsymbol{X}$ are independent from one another. Note that when $v = v'$ and $j = j'$ the variance is $\sum_{u=1}^{n} \boldsymbol{A}_{vu}^2 = 1$ by our assumption on $\boldsymbol{A}$.

By the dependence structure of $P$, one can write:

$$\mathbb{E}_{\boldsymbol{Z} \sim P}[H_J(\boldsymbol{z}_v) H_{J'}(\boldsymbol{z}_{v'})] = \prod_{j=1}^d \mathbb{E}_{\boldsymbol{Z} \sim P}[H_{J_j}(\boldsymbol{Z}_{vj}) H_{J'_j}(\boldsymbol{Z}_{v'j})] \tag{57}$$

$$= \prod_{j=1}^d \mathbb{E}_{(\boldsymbol{Z}_{vj}, \boldsymbol{Z}_{v'j}) \sim \mathcal{N}(0, \Sigma(\rho_{vv'}))}[H_{J_j}(\boldsymbol{Z}_{vj}) H_{J'_j}(\boldsymbol{Z}_{v'j})] \tag{58}$$

$$= \prod_{j=1}^d \rho_{vv'}^{J_i} \delta_{J_i = J'_i} \tag{59}$$

$$= \rho_{vv'}^{|J|} \delta_{J=J'}. \tag{60}$$

Therefore,

$$\langle H_J^{\boldsymbol{A}}, H_{J'}^{\boldsymbol{A}} \rangle_{L^2(\mathcal{N})} = \delta_{J=J'} \cdot \frac{1}{n} \sum_{v,v' \in [n]} \rho_{vv'}^{|J|} = \delta_{J=J'} \cdot c_J, \tag{61}$$

where $c_J = \frac{1}{n} \sum_{v,v' \in [n]} \rho_{vv'}^{|J|}$. $\qquad\qquad\qquad\qquad\qquad\qquad\square$

**Remark 17.** The quantity $\rho_{vv'}$ deserves an explanation. Essentially, $\rho_{vv'}$ measures how similar two nodes $v$ and $v'$ are, in term of composition of its incoming edges. Algebraically, it is the $(vv')$-th entry of the gram matrix $\boldsymbol{A}\boldsymbol{A}^\top$. We can get the correct normalization of $\boldsymbol{A}$ in the premise of Lemma 16 by starting with a graph $\mathcal{G}$, adding a self-loop to each node of $\mathcal{G}$ and set $\boldsymbol{A} = (\boldsymbol{D}^\dagger)^{\frac{1}{2}} \boldsymbol{M}$ where $\boldsymbol{M}$ is the adjacency matrix of $\mathcal{G}$ (with self-loops) and $\boldsymbol{D}$ is the in-degree matrix, we can ensure that $\rho_{vv'} \in [0, 1]$ for each $v \neq v'$:

$$\rho_{vv'} = \sum_{u=1}^n \boldsymbol{A}_{vu} \boldsymbol{A}_{v'u} = \frac{|N_v \cap N_{v'}|}{\sqrt{\deg(v) \deg(v')}} \leq 1, \tag{62}$$

where $N_v$ is the set of vertices with an incoming edge to $v \in [n]$.

This immediately gives a bound $\max_J c_J \leq n^2$. Finally, it is also worth noting that $c_J$ does not vanish under the above normalization, since adding self-loops before the normalization to $\mathcal{G}$ ensures that $\rho_{vv} = 1$ for all $v \in [n]$ even if $v$ was an isolated vertex in the original graph. Therefore, without loss of generality, we can assume that $c_J \in \Theta(1)$.

It is possible to use a different normalization, such as the Laplacian, as long as one can ensure that $\rho_{vv'}, c_J = \Theta(1)$. The requirement that rows of $\boldsymbol{A}$ sums to 1 is also not strictly necessary but does greatly simplify our exposition.

Note that we are not claiming a complete orthonormal basis for the space $L^2(\mathbb{R}^{n \times d}, \mathcal{N}^{n \times d})$. However, all of the functions we care about will lie in $\mathcal{H}^{(d)} := \text{span}(H_J^{\boldsymbol{A}})_{J \in \mathbb{N}^d}$. For a fixed $n$ and a function $f \in \mathcal{H}^{(d)}$, we define $\hat{f}_J = \langle f, H_J^{\boldsymbol{A}} \rangle_{L^2(\mathcal{N})}$ and the degree $m$ part of $f$: $f^{[m]}(\boldsymbol{X}) = \sum_{|J|=m} \hat{f}_J H_J^{\boldsymbol{A}}(\boldsymbol{X})$.

Facts about gradients of Hermite polynomial also carry over to invariant Hermite polynomials:

**Lemma 18.** *For a fixed graph $\mathcal{G}$ on $n$ vertices and corresponding $\boldsymbol{A}$, we have:*

1. $\frac{\partial}{\partial \boldsymbol{X}_{vj}} H_M^{\boldsymbol{A}}(X) = \frac{1}{\sqrt{n}} \sum_{v'=1}^n \sqrt{M_j} H_{M-E_j}((\boldsymbol{A}\boldsymbol{X})_{v'}) \boldsymbol{A}_{v'v}$, *for all* $v \in [n], j \in [d], M \in \mathbb{N}^d$, *where $E_j$ is the vector with 1 at the $j'$-th position and 0 everywhere else.*

2. $\mathbb{E}_{\boldsymbol{X} \sim \mathcal{N}^{n \times d}}[\langle \nabla H_M^{\boldsymbol{A}}(\boldsymbol{X}), \nabla H_L^{\boldsymbol{A}}(\boldsymbol{X}) \rangle] = \delta_{M=L} \cdot |M| c_M$.

3. $\mathbb{E}_{\boldsymbol{X} \sim \mathcal{N}^{n \times d}}[\langle \nabla^l H_M^{\boldsymbol{A}}(\boldsymbol{X}), \nabla^l H_L^{\boldsymbol{A}}(\boldsymbol{X}) \rangle] = \delta_{M=L} \cdot |M|(|M|-1) \ldots (|M|-l+1) c_M$.

*In the above, $c_M = n^{-1} \sum_{v',v''=1}^n \rho_{v'v''}^{|M|}$.*

*Proof.*     1. We have:

$$\frac{\partial}{\partial \boldsymbol{X}_{vj}} H_M^{\boldsymbol{A}}(\boldsymbol{X}) = \frac{1}{\sqrt{n}} \sum_{v'=1}^n (\nabla H_M((\boldsymbol{A}\boldsymbol{X})_{v'}))_j \boldsymbol{A}_{v'v} \tag{63}$$

$$= \frac{1}{\sqrt{n}} \sum_{v'=1}^n \sqrt{M_j} H_{M-E_j}((\boldsymbol{A}\boldsymbol{X})_{v'}) \boldsymbol{A}_{v'v}, \tag{64}$$

where $E_{j'}$ is the vector with 1 at the $j'$-th position and 0 everywhere else. The second equality is a known fact for the multivariate Hermite polynomials (See Appendix C of Diakonikolas et al. (2020)).

2. Sum the above over all pairs $v, v'$ and take expectation to get:

$$\mathbb{E}_{\boldsymbol{X} \sim \mathcal{N}^{n \times d}}[\langle \nabla H_M^{\boldsymbol{A}}(\boldsymbol{X}), \nabla H_L^{\boldsymbol{A}}(\boldsymbol{X}) \rangle] \tag{65}$$

$$= \sum_{v=1}^n \sum_{j=1}^d \frac{1}{n} \mathbb{E}_{\boldsymbol{X} \sim \mathcal{N}^{n \times d}} \left[ \sum_{v',v''=1}^n (\nabla H_M((\boldsymbol{A}\boldsymbol{X})_{v'}))_j \boldsymbol{A}_{v'v} (\nabla H_L((\boldsymbol{A}\boldsymbol{X})_{v''}))_j \boldsymbol{A}_{v''v} \right]. \tag{66}$$

By the orthogonality property of Hermite multivariate polynomial under dependent multivariate Gaussian (equation 60), we can compute:

$$\mathbb{E}_{\boldsymbol{X} \sim \mathcal{N}^{n \times d}}[H_{M-E_j}((\boldsymbol{A}\boldsymbol{X})_{v'}) H_{L-E_j}((\boldsymbol{A}\boldsymbol{X})_{v''})] = \rho_{v'v''}^{|M|-1} \delta_{M=L}, \tag{67}$$

and,

$$\mathbb{E}_{\boldsymbol{X} \sim \mathcal{N}^{n \times d}}[\langle \nabla H_M^{\boldsymbol{A}}(\boldsymbol{X}), \nabla H_L^{\boldsymbol{A}}(\boldsymbol{X}) \rangle] = \frac{1}{n} \sum_{v,v',v''=1}^n \boldsymbol{A}_{v'v} \boldsymbol{A}_{v''v} \rho_{v'v''}^{|M|-1} \delta_{M=L} \sum_{j=1}^d M_j \tag{68}$$

$$= \delta_{M=L} \cdot \frac{|M|}{n} \sum_{v',v''=1}^n \rho_{v'v''}^{|M|-1} \sum_{v=1}^n \boldsymbol{A}_{v'v} \boldsymbol{A}_{v''v} \tag{69}$$

$$= \delta_{M=L} \cdot \frac{|M|}{n} \sum_{v',v''=1}^n \rho_{v'v''}^{|M|}. \tag{70}$$

3. Recursively apply the first part to get:

$$\frac{\partial^l}{\partial \boldsymbol{X}_{v_1 j_1}, \dots, \boldsymbol{X}_{v_l j_l}} H_M^G(\boldsymbol{X}) \tag{71}$$

$$= \frac{1}{\sqrt{n}} \sum_{v'=1}^n \left( \prod_{p=1}^l \boldsymbol{A}_{v'v_p} \sqrt{\left( M - \sum_{p'=1}^p E_{j_{p'}} \right)_{j_p}} \right) H_{M-\sum_{p=1}^l e_{j_p}}((\boldsymbol{A}\boldsymbol{X})_{v'}). \tag{72}$$

Summing indices and passing through the expectation to get:

$$\mathbb{E}_{\boldsymbol{X} \sim \mathcal{N}^{n \times d}}[\langle \nabla^l H_M^{\boldsymbol{A}}(\boldsymbol{X}), \nabla^l H_L^{\boldsymbol{A}}(\boldsymbol{X}) \rangle] \tag{73}$$

$$= \frac{\delta_{M=L}}{n} \left( \sum_{j_1,\dots,j_l} \left( M - \sum_{p'=1}^p E_{j_{p'}} \right)_{j_p} \right) \sum_{v',v''} \left( \sum_{v=1}^n \boldsymbol{A}_{v'v} \boldsymbol{A}_{v''v} \right)^l \rho_{v'v''}^{|M|-l} \tag{74}$$

$$= \delta_{M=L} |M|(|M|-1) \dots (|M|-l) \cdot \frac{1}{n} \sum_{v',v''=1}^n \rho_{v'v''}^{|M|}. \tag{75}$$

$\square$

**Corollary 19.** *Fix $k, d, e \in \mathbb{N}$. Let $p, q$ be linear combinations of $H_M^{\boldsymbol{A}} : \mathbb{R}^d \to \mathbb{R}$ with $M \in \mathbb{N}^d$ varying among all multi-indices of size $k = |M|$. Then,*

$$\langle \nabla^k p(\boldsymbol{X}), \nabla^k q(\boldsymbol{X}) \rangle = k! \cdot \mathbb{E}_{\boldsymbol{X} \sim \mathcal{N}^{n \times d}}[p(\boldsymbol{X})q(\boldsymbol{X})]. \tag{76}$$

*Proof.* Let $p(\boldsymbol{X}) = \sum_{M:|M|=k} b_M H_M^{\boldsymbol{A}}(\boldsymbol{X})$ and $q(X) = \sum_{K:|K|=k} f_K H_K^{\boldsymbol{A}}(\boldsymbol{X})$. Then:

$$\mathbb{E}_{\boldsymbol{X} \sim \mathcal{N}^{n \times d}}[p(\boldsymbol{X})q(\boldsymbol{X})] = \sum_{M:|M|=k} f_M b_M c_M, \tag{77}$$

where as before, $c_M = n^{-1} \sum_{v', v''=1}^{n} \rho_{v'v''}^{|M|}$.

From the previous lemma,

$$\langle \nabla^k p(\boldsymbol{X}), \nabla^k q(\boldsymbol{X}) \rangle = \mathbb{E}_{\boldsymbol{X} \sim \mathcal{N}^{n \times d}}\left[ \left\langle \sum_{M:|M|=k} b_M \nabla^k H_M^{\boldsymbol{A}}(\boldsymbol{X}), \sum_{K:|K|=k} f_M \nabla^k H_K^{\boldsymbol{A}}(\boldsymbol{X}) \right\rangle \right] \tag{78}$$

$$= \sum_{M,K:|M|=|K|=k} b_M f_M k! \delta_{M=K} c_M \tag{79}$$

$$= k! \sum_{M:|M|=k} b_M f_M c_M \tag{80}$$

$$= k! \cdot \mathbb{E}_{\boldsymbol{X} \sim \mathcal{N}^{n \times d}}[p(\boldsymbol{X})q(\boldsymbol{X})]. \tag{81}$$

$\square$

### E.3  PROOF OF THEOREM 4

The proof of Theorem 4 is derived from the construction in Diakonikolas et al. (2020). Hence, much of the setup will be taken from their construction. However, care must be taken to use the proper invariant Hermite polynomial basis we derived in the previous subsection. For completeness, we extend their formalism here as needed and note where proofs are directly derived from their work.

We first verify that the orthogonal system of invariant Hermite polynomial subsumes 1-hidden-layer GNNs. We formally show this below to continue the proof later on in the basis of the invariant Hermite polynomials. As a reminder, we defined $\mathcal{H}^{(d)} := \text{span}\left(H_J^{\boldsymbol{A}}\right)_{J \in \mathbb{N}^d}$.

**Lemma 20** (Inner product decomposition). *For every $k \in \mathbb{N}$ and $\mathcal{G} \in \mathbb{G}_n$, $F_{\mathcal{G}}^{d,k} \subset \mathcal{H}^{(d)}$.*

*Proof.* Pick $f \in F_{\mathcal{G}}^{d,k}$, with a corresponding $\boldsymbol{W}$ and $\boldsymbol{a}$. Let $g : \mathbb{R}^d \to \mathbb{R} : \boldsymbol{x} \mapsto \sum_{c=1}^{2k} a_c \sigma(\langle \boldsymbol{x}, \boldsymbol{w}_c \rangle)$ where $\boldsymbol{w}_c$ is the $c$-th column of $\boldsymbol{W}$. Then we have $f(\boldsymbol{X}) = \sum_{v=1}^{n} g((\boldsymbol{A}\boldsymbol{X})_v)$ where the subscript represents the row of the corresponding matrix. Since $\rho \in L^2(\mathcal{N})$, $g$ is in $L^2(\mathcal{N})$ and we can write uniquely:

$$g(\boldsymbol{x}) = \sum_{J \in \mathbb{N}^d} \hat{g}_J H_J(\boldsymbol{x}). \tag{82}$$

Therefore,

$$f(\boldsymbol{X}) = \sum_{v=1}^{n} g((\boldsymbol{A}\boldsymbol{X})_v) = \sum_{v=1}^{n} \sum_{J \in \mathbb{N}^d} \hat{g}_J H_J((\boldsymbol{A}\boldsymbol{X})_v) = \sum_{J \in \mathbb{N}^d} (\hat{g}_J \sqrt{n}) H_J^{\boldsymbol{A}}(\boldsymbol{X}) \in \mathcal{H}^{(d)}. \tag{83}$$

$\square$

Now we verify that the special function $f_{\mathcal{G}}$ in our construction of the family of hard functions has vanishing low degree moments:

**Lemma 21** (Low degree moment vanishes). *For any $k \in \mathbb{N}$, for any $\mathcal{G}$ and appropriate $\boldsymbol{A}$, $f_{\mathcal{G}} \in \mathcal{H}^{(2)}$. Furthermore, $(\hat{f}_{\mathcal{G}})_J = 0$ for every $|J| < k$.*

*Proof.* The first part follows from specializing Lemma 20 for $d = 2$. The second part follows from the fact that the corresponding $g$ function in the proof of Lemma 20 for $f$ is the function $f_{\varphi,\sigma}$ in Diakonikolas et al. (2020), whose low degree moments vanish. $\square$

It is also easy to verify that $C_{\mathcal{G}}^{\mathcal{B}} \subset F_{\mathcal{G}}^{d,k}$:

**Lemma 22.** *For any $k, d, n \in \mathbb{N}$, for some activation $\sigma$ that is not a low-degree polynomial, $C_{\mathcal{G}}^{\mathcal{B}} \subset F_{\mathcal{G}}^{d,k}$.*

*Proof.* Pick $\boldsymbol{B} \in \mathcal{B}$ arbitrarily and let $h(\boldsymbol{X}) = f_{\mathcal{G}}(\boldsymbol{X}\boldsymbol{B})$. We have:
$$h(\boldsymbol{X}) = f_{\mathcal{G}}(\boldsymbol{X}\boldsymbol{B}) = \mathbf{1}_n^\top \sigma\left(\boldsymbol{A}(\boldsymbol{X}\boldsymbol{B})\boldsymbol{W}^*\right)\boldsymbol{a}^* = \mathbf{1}_n^\top \sigma\left(\boldsymbol{A}\boldsymbol{X}(\boldsymbol{B}\boldsymbol{W}^*)\right)\boldsymbol{a}^*. \tag{84}$$

Since $\boldsymbol{B}\boldsymbol{W}^*$ is in $\mathbb{R}^{d \times 2k}$ and $\boldsymbol{a}^*$ is in $\mathbb{R}^{2k}$, we have $h \in F_{\mathcal{G}}^{d,k}$. Functions in $\mathcal{F}_k^{\mathcal{B}}$ also do not vanish, as long as $\boldsymbol{A}$ does not have vanishing rank and the activation function $\sigma$ is not a low-degree polynomial, by the same argument as in Diakonikolas et al. (2020). $\square$

From Diakonikolas et al. (2020), we know the following facts about matrices in $\mathbb{R}^{d \times 2}$:

**Lemma 23** (Lemma 4.7 in Diakonikolas et al. (2020)). *Fix $c \in (0, \frac{1}{2})$. Then there is a set $S \subset \mathbb{R}^{d \times 2}$ of size at least $2^{\Omega(d^c)}$ such that for each $\boldsymbol{A}, \boldsymbol{B} \in S$, $\|\boldsymbol{A}^\top\boldsymbol{B}\|_2 \leq O(d^{c-\frac{1}{2}})$.*

The following Lemma generalizes Lemma 4.6 in Diakonikolas et al. (2020):

**Lemma 24.** *Fix a function $\mathcal{H}^{(2)} \ni p : \mathbb{R}^{n \times 2} \to \mathbb{R}$ and $\boldsymbol{U}, \boldsymbol{V} \in \mathbb{R}^{d \times 2}$ such that $\boldsymbol{U}^\top\boldsymbol{U} = \boldsymbol{V}^\top\boldsymbol{V} = \mathbf{1}_2$. Then:*
$$\left|\langle p(\cdot\boldsymbol{U})p(\cdot\boldsymbol{V})\rangle_{L^2(\mathcal{N})}\right| \leq \sum_{m=0}^{\infty} \left(2\sqrt{n}\|\boldsymbol{U}^\top\boldsymbol{V}\|_2\right)^m \cdot \|p^{[m]}\|_{L^2(\mathcal{N})}^2, \tag{85}$$

*where $p^{[m]}$ is the degree $m$ part of $p$.*

*Proof.* Let $f, g : \mathbb{R}^{n \times d} \to \mathbb{R}$ be defined as $f(\boldsymbol{X}) = p(\boldsymbol{X}\boldsymbol{U})$ and $g(\boldsymbol{X}) := p(\boldsymbol{X}\boldsymbol{V})$. Let $f^{[m]}, g^{[m]}$ and $p^{[m]}$ be the degree $m$ part of the corresponding function.

From Corollary 19,
$$\sum_{m=0}^{\infty} \frac{1}{m!}\langle \nabla^m f^{[m]}(\boldsymbol{X}), \nabla^m g^{[m]}(\boldsymbol{X})\rangle = \sum_{m=0}^{\infty} \mathbb{E}_{\boldsymbol{X} \sim \mathcal{N}^{n \times d}}[f^{[m]}(\boldsymbol{X})g^{[m]}(\boldsymbol{X})] \tag{86}$$
$$= \mathbb{E}_{\boldsymbol{X} \sim \mathcal{N}^{n \times d}}[f(\boldsymbol{X})g(\boldsymbol{X})] \tag{87}$$
$$= \mathbb{E}_{\boldsymbol{X} \sim \mathcal{N}^{n \times 2}}[p(\boldsymbol{X}\boldsymbol{U})p(\boldsymbol{X}\boldsymbol{V})] \tag{88}$$
$$= \sum_{m=0}^{\infty} \mathbb{E}_{\boldsymbol{X} \sim \mathcal{N}^{n \times 2}}[p^{[m]}(\boldsymbol{X}\boldsymbol{U})p^{[m]}(\boldsymbol{X}\boldsymbol{V})] \tag{89}$$
$$= \sum_{m=0}^{\infty} \frac{1}{m!}\langle \nabla_{\boldsymbol{X}}^m p^{[m]}(\boldsymbol{X}\boldsymbol{U}), \nabla_{\boldsymbol{X}}^m p^{[m]}(\boldsymbol{X}\boldsymbol{V})\rangle. \tag{90}$$

Fix $m \in \mathbb{R}$, for each indices $v_1, \ldots, v_m \in [n], s_1, \ldots, s_m \in [d]$, in Einstein notation,
$$(\partial_{v_1,s_1} \ldots \partial_{v_m,s_m} f^{[m]})(\boldsymbol{X})(\partial_{v_1,s_1} \ldots \partial_{v_m,s_m} g^{[m]})(\boldsymbol{X}) \tag{91}$$
$$= \partial_{v_1,j_1} \ldots \partial_{v_m,j_m} p^{[m]}(\boldsymbol{X}\boldsymbol{U})\partial_{v_1,j_1'} \ldots \partial_{v_m,j_m'} p^{[m]}(\boldsymbol{X}\boldsymbol{V})\boldsymbol{U}_{s_1 j_1} \ldots \boldsymbol{U}_{s_m j_m}\boldsymbol{V}_{s_1 j_1'} \ldots \boldsymbol{V}_{s_m j_m'} \tag{92}$$
$$\tag{93}$$

Taking the sum over all $v_1, \ldots, v_m \in [n], s_1, \ldots, s_m \in [d]$, while still in Einstein notation, to get:
$$|\langle \nabla_{\boldsymbol{X}}^m p^{[m]}(\boldsymbol{X}\boldsymbol{U}), \nabla_{\boldsymbol{X}}^m p^{[m]}(\boldsymbol{X}\boldsymbol{V})\rangle| \tag{94}$$
$$= |\partial_{v_1,j_1} \ldots \partial_{v_m,j_m} p^{[m]}(\boldsymbol{X}\boldsymbol{U})\partial_{v_1,j_1'} \ldots \partial_{v_m,j_m'} p^{[m]}(\boldsymbol{X}\boldsymbol{V})(\boldsymbol{U}^\top\boldsymbol{V})_{j_1 j_1'} \ldots (\boldsymbol{U}^\top\boldsymbol{V})_{j_m j_m'}| \tag{95}$$
$$\leq 2^m\|\nabla_{\boldsymbol{X}}^m p^{[m]}(\boldsymbol{X}\boldsymbol{U})\|_2^2 \cdot \sqrt{n^m}\|\boldsymbol{U}^\top\boldsymbol{V}\|_2^m \tag{96}$$
$$= m! \cdot \mathbb{E}_{\boldsymbol{X} \sim \mathcal{N}^{n \times d}}[(p^{[m]}(\boldsymbol{X}))^2] \cdot \left(2\sqrt{n}\|\boldsymbol{U}^\top\boldsymbol{V}\|_2\right)^m, \tag{97}$$

where we used 3-variable Cauchy-Schwarz inequality $|a_{ijk}b_{ijk}c_{ijk}| \leq \sqrt{a_{i'j'k'}^2 b_{i''j''k''}^2 c_{i'''j'''k'''}^2}$ to obtain the inequality and Corollary 19 to obtain the last line. $\square$

Notice that in the previous derivation, inequality only occurs at the Cauchy-Schwarz inequality step. We can do a more careful analysis to get:

**Corollary 25.** *In the same setting as Lemma 24, we have:*

$$\langle p(\cdot \boldsymbol{U})p(\cdot \boldsymbol{V})\rangle_{L^2(\mathcal{N})} \tag{98}$$

$$= \sum_{m=0}^{\infty} \frac{1}{m!} \sum_{v_1,\dots,v_m=1}^{n} \sum_{j_1,j_1',\dots,j_m,j_m'=1}^{2} \partial_{v_1,j_1} \dots \partial_{v_m,j_m} p^{[m]}(\boldsymbol{XU}) \partial_{v_1,j_1'} \dots \partial_{v_m,j_m'} p^{[m]}(\boldsymbol{XV})(\boldsymbol{U}^\top \boldsymbol{V})_{j_1 j_1'} \dots (\boldsymbol{U}^\top \boldsymbol{V})_{j_m j_m'}. \tag{99}$$

As the last ingredient, we need the following lemma that connects the family of hard functions to lower bounds on CSQ algorithms:

**Lemma 26** (Lemma 15 of Diakonikolas et al. (2020)). *Let $\mathcal{X}$ be the input space and $\mathcal{D}$ some distribution over $\mathcal{X}$. For a finite set of functions $C = \{f : X \to \mathbb{R}\}$, define:*

$$\rho(C) := 1/|C|^2 \sum_{f,g \in C} \langle f, g \rangle_{L^2(\mathcal{D})}. \tag{100}$$

*The correlational statistical query dimension is defined as:*

$$\text{SDA}(\text{C}, \mathcal{D}, \tau) := \max\{\text{m} \mid \rho(\text{C}') \leq \tau, \forall \text{C}' \subset \text{C}, |\text{C}'|/|\text{C}| \geq 1/\text{m}\}. \tag{101}$$

*Suppose that $\text{SDA}(\text{C}, \mathcal{D}, \tau) \geq \text{m}$ for some $C, \mathcal{D}, \tau$ and $m$, then there is a small enough $\epsilon$ such that any CSQ algorithm that queries from an oracle of $f \in C$ and learn a hypothesis $h \in C$ with $\langle f - h \rangle_{L^2(\mathcal{D})} \leq \epsilon$ either requires $\Omega(m)$ queries or at least one query with precision more than $\sqrt{\tau}$.*

We can now proceed to the proof of the main theorem:

**Theorem 27.** *Any correlational SQ algorithm that, for every concept $g \in C_{\mathcal{G}}^{\mathcal{B}}$ learns a hypothesis $h$ such that $\|g - h\|_{L^2(\mathcal{N})} \leq \epsilon$ for some small $\epsilon > 0$ requires either $2^{d^{\Omega(1)}}$ queries or queries of tolerance at most $d^{-k\Omega(1)} + 2^{-d^{\Omega(1)}}$.*

*Proof.* This proof mirrors that in Diakonikolas et al. (2020). Fix $c \in (0, 1/2)$. To upper bound pairwise correlation between different functions in $C_{\mathcal{G}}^{\mathcal{B}}$, consider for any $\boldsymbol{B}_i \neq \boldsymbol{B}_j \in \mathcal{B}$:

$$\left|\langle f_{\mathcal{G}}(\cdot \boldsymbol{B}_i), f_{\mathcal{G}}(\cdot \boldsymbol{B}_j)\rangle_{L^2(\mathcal{N})}\right| \leq \sum_{m=0}^{\infty} (2\sqrt{n}\|\boldsymbol{B}_i^\top \boldsymbol{B}_j\|_2)^m \cdot \|f_{\mathcal{G}}^{[m]}\|_{L^2(\mathcal{N})}^2 \tag{102}$$

$$= \sum_{m>k}^{\infty} (2\sqrt{n}\|\boldsymbol{B}_i^\top \boldsymbol{B}_j\|_2)^m \cdot \|f_{\mathcal{G}}^{[m]}\|_{L^2(\mathcal{N})}^2 \tag{103}$$

$$\leq (2\sqrt{n}\|\boldsymbol{B}_i^\top \boldsymbol{B}_j\|_2)^k \sum_{m>k}^{\infty} \|f_{\mathcal{G}}^{[m]}\|_{L^2(\mathcal{N})}^2 \tag{104}$$

$$\leq O(d^{k(c-1/2)})\|f_{\mathcal{G}}\|_{L^2(\mathcal{N})}^2. \tag{105}$$

In the above derivation, the first line uses Lemma 24; the second line uses Lemma 21; and the last two lines use Lemma 23 and the fact that we can choose an appropriate parameter to make $\|\boldsymbol{B}_i^\top \boldsymbol{B}_j\|_2 < 1$. Dividing both sides by $\|f_{\mathcal{G}}\|_{L^2(\mathcal{N})}$ gives

$$\left|\langle g_{\mathcal{G}}^{\boldsymbol{B}_i}, g_{\mathcal{G}}^{\boldsymbol{B}_j}\rangle_{L^2(\mathcal{N})}\right| \leq O(d^{k(c-1/2)}). \tag{106}$$

Now, since $\boldsymbol{B}_i^\top \boldsymbol{B}_i = \mathbf{1}_2$, we have $\|f_{\mathcal{G}}(\cdot \boldsymbol{B}_i)\|_{L^2(\mathcal{N})} = \|f_{\mathcal{G}}\|_{L^2(\mathcal{N})}$ and therefore $\|g_{\mathcal{G}}^{\boldsymbol{B}_i}\|_{L^2(\mathcal{N})} = 1$.

We are in a position to apply Lemma 26. Let $\tau := \max_{\boldsymbol{B}_i \neq \boldsymbol{B}_j} \langle g_{\mathcal{G}}^{\boldsymbol{B}_i}, g_{\mathcal{G}}^{\boldsymbol{B}_j} \rangle_{L^2(\mathcal{N})}$. Computing average pairwise correlation of $C_{\mathcal{G}}^{\mathcal{B}}$ yields:

$$\rho(C_{\mathcal{G}}^{\mathcal{B}}) = 1/|\mathcal{B}| + (|\mathcal{B}| - 1)\tau/(2|\mathcal{B}|) \leq 1/|\mathcal{B}| + \tau. \tag{107}$$

Finally, setting $\tau' := 1/|\mathcal{B}| + \tau$ gives $\mathrm{SDA}(C_{\mathcal{G}}^{\mathcal{B}}, \mathcal{N}, \tau') = 2^{\Omega(\mathrm{d}^{\mathrm{c}})}$ and an application of Lemma 26 completes the proof for $F_{\mathcal{G}}^{d,k}$.

To get the proof for $F_n^{d,k}$, by law of total expectation:

$$\left| \mathbb{E}_{(\boldsymbol{X},\mathcal{G}) \sim \mathcal{N} \times \mathcal{E}}[f_n(\boldsymbol{X}\boldsymbol{B}_i, \mathcal{G})f_n(\boldsymbol{X}\boldsymbol{B}_j, \mathcal{G})] \right| \tag{108}$$

$$= |\mathbb{E}_{\mathcal{E}}\mathbb{E}_{\mathcal{N} \times \mathcal{E}}[f_n(\boldsymbol{X}\boldsymbol{B}_i, \mathcal{G})f_n(\boldsymbol{X}\boldsymbol{B}_j, \mathcal{G})|\mathcal{G}]| \tag{109}$$

$$\leq \sum_{\mathcal{G} \in \mathbb{G}_n} \mathrm{Pr}_{\mathcal{E}}(\mathcal{G}) \left| \mathbb{E}_{\mathcal{N}}[f_{\mathcal{G}}(\boldsymbol{X}\boldsymbol{B}_i)f_{\mathcal{G}}(\boldsymbol{X}\boldsymbol{B}_j)] \right| \tag{110}$$

$$\leq \sum_{\mathcal{G} \in \mathbb{G}_n} \mathrm{Pr}_{\mathcal{E}}(\mathcal{G}) \cdot \left( \max_{\boldsymbol{B}_i \neq \boldsymbol{B}_j \in \mathcal{B}} (2\sqrt{n}\|\boldsymbol{B}_i^\top \boldsymbol{B}_j\|_2)^k \right) \cdot \|f_{\mathcal{G}}\|_{L^2(\mathcal{N})}^2 \tag{111}$$

$$\leq O(d^{k(c-1/2)})\|f_n\|_{L^2(\mathcal{N} \times \mathcal{E})}^2, \tag{112}$$

where in the third line, we use the fact that $f_{\mathcal{G}}(\boldsymbol{Z}) = f_n(\boldsymbol{Z}, \mathcal{G})$ and triangle inequality; in the fourth line, we use Eq. (105) and in the last line, we use another instance of total expectation to get $\|f_n\|_{L^2(\mathcal{N} \times \mathcal{E})} = \mathbb{E}_{\mathcal{G} \sim \mathcal{E}}\|f_{\mathcal{G}}\|_{L^2(\mathcal{N})}$. It is worth pointing out that the only instance of $\mathcal{G}$ for which we cannot apply Eq. (105) in the fourth line (assuming we use the normalized Laplacian as $\boldsymbol{A}(\mathcal{G})$) is when $\mathcal{G}$ is the empty graph. However, in that case, $\|f_{\mathcal{G}}\|_{L^2(\mathcal{N})} = 0$ since $f_{\mathcal{G}} \equiv 0$ and the inequality in the fourth line still holds.

Lastly, proceed identically to the proof for $F_{\mathcal{G}}^{d,k}$ to use Lemma 26 and get the final bound. □

## F FRAME-AVERAGING HARDNESS PROOFS

**Frame averaging.** Let $G$ be a subgroup of the permutation group $S_n$ that acts on $\mathbb{R}^{n \times d}$ by permuting the rows and $e$ its identity element. Let $\rho : G \to \mathrm{GL}(n)$ with $\rho(g)$ the representation of $g$ as a matrix in $\mathbb{R}^{n \times n}$. For any group element $g \in G$ and $\boldsymbol{X} \in \mathbb{R}^{n \times d}$, we will write $g\boldsymbol{X}$ as a shorthand for $\rho(g)\boldsymbol{X}$ where the product is matrix multiplication. A frame is a function $\mathcal{F} : \mathbb{R}^{n \times d} \to 2^G \setminus \{\emptyset\}$ satisfying $\mathcal{F}(g\boldsymbol{X}) = g\mathcal{F}(\boldsymbol{X})$ set-wise. For given frames $\mathcal{F}$, we will show a hardness result for the following class of functions:

$$\mathcal{H}_{\mathcal{F}} := \left\{ f : \mathbb{R}^{n \times d} \to \mathbb{R} \text{ with } f(\boldsymbol{X}) = \frac{1}{\sqrt{|\mathcal{F}(\boldsymbol{X})|}} \sum_{g \in \mathcal{F}(\boldsymbol{X})} \boldsymbol{a}^\top \sigma(\boldsymbol{W}^\top (g^{-1}\boldsymbol{X}))\mathbf{1}_d \mid \right.$$

$$\left. \boldsymbol{W} \in \mathbb{R}^{n \times k}, \boldsymbol{a} \in \mathbb{R}^k \right\}, \tag{113}$$

for given nonlinearity $\sigma$ that is not a polynomial of low degree.

### F.1 EXPONENTIAL LOWER BOUND FOR CONSTANT FRAME AND REYNOLDS OPERATOR

**Lemma 28** (Constant frame must be whole group). *If $\mathcal{F}(\boldsymbol{X})$ is a constant function, i.e. $\mathcal{F}(\boldsymbol{X})$ is independent of $\boldsymbol{X}$, then $\mathcal{F}(\boldsymbol{X}) = G \,\forall \boldsymbol{X}$.*

*Proof.* When $\mathcal{F}(\boldsymbol{X})$ is a constant function, say $F \subseteq G$ for all $\boldsymbol{X}$, then the frame property mandates that $F \neq \emptyset$ and there is some $f \in F$. The frame property also tells us that since $f^{-1} \in G$ (and $G$ is a group), $f^{-1}F = F$ and thus $e \in F$. Pick any element $g \in G$ and observe that since $e \in F$ and $gF = F$, $g$ must also be in $F$. As this is true for all elements $g \in G$, $F$ must be the whole group $G$ itself. □

In the following, we will therefore assume that $G$ is a polynomial-sized (in $n$) subgroup of $S_n$. We will slightly modify Lemma 17 of (Diakonikolas et al., 2020) to show:

**Lemma 29.** *For any $0 < c < 1/2$, and small $\epsilon \in (0, 1)$, for any polynomial-sized subset $F \subset S_n$, there exists a set $\mathcal{B}'$ of at least $2^{\Omega(n^c)}/|F|^2$ matrices in $\mathbb{R}^{n \times 2}$ such that for each pair $\boldsymbol{B}, \boldsymbol{B}' \in \mathcal{B}'$ and for each $g, g' \in F$, it holds that:*

1. $(g\boldsymbol{B})^\top(g\boldsymbol{B}) = \boldsymbol{I}$;

2. $\|(g\boldsymbol{B})^\top(g'\boldsymbol{B}')\|_2 \leq O(n^{c-1/2})$ *when $\boldsymbol{B} \neq \boldsymbol{B}'$; and*

3. *when $g \neq g'$, $(g\boldsymbol{B})^\top(g'\boldsymbol{B}))_{ij} =: \begin{bmatrix} a_1 & b_1 \\ b_2 & a_2 \end{bmatrix}$ where $b_i \leq O(n^{c-1/2})$ and,*

$$a_i - \frac{F_{g,g'}}{n\sqrt{1-\epsilon}} \leq O(n^{c-1/2}), \tag{114}$$

$$a_i - \sqrt{1-\epsilon}\frac{F_{g,g'}}{n} \geq -O(n^{c-1/2}), \tag{115}$$

*for $i \in [2]$. Here $F_{g,g'}$ is the number of fixed points of $g^{-1}g'$ (number of cycles of length one).*

The proof is delayed to App. F.3 for a more streamlined read.

**Family of hard functions.** Recall the low-dimensional function from the proof of Theorem 4:

$$f^*_{\exp} : \mathbb{R}^{2 \times d} \to \mathbb{R} \text{ with } f(\boldsymbol{X}) = (\boldsymbol{a}^*)^\top \sigma((\boldsymbol{W}^*)^\top \boldsymbol{X})\mathbf{1}_d \tag{116}$$

for some special parameter $\boldsymbol{a}^* \in \mathbb{R}^{2k}$ and $\boldsymbol{W}^* \in \mathbb{R}^{2 \times 2k}$ in Eq. (11). We now define the family of hard functions, indexed by a set of matrices $\mathcal{B} \subset \mathbb{R}^{n \times 2}$ obtained from Lemma 29:

$$C^\mathcal{B}_\mathcal{F} = \left\{ g_{\boldsymbol{B}} : \mathbb{R}^{n \times d} \to \mathbb{R} \text{ with } g_{\boldsymbol{B}}(\boldsymbol{X}) = \frac{\sum_{g \in G} f^*_{\exp}(\boldsymbol{B}^\top g^{-1} \boldsymbol{X})}{\sqrt{|G| \cdot \|f^*_{\exp}\|_\mathcal{N}}} \mid \boldsymbol{B} \in \mathcal{B}' \subset \mathbb{R}^{n \times 2} \right\}. \tag{117}$$

**Theorem 30.** *For any feature dimension $d = \Theta(1)$ and width parameter $k = \Theta(n)$. Let $G$ be a subgroup of $S_n$ with $|G| = \Theta(\text{poly}(n))$. Pick a set $\mathcal{B}'$[4] $\subset \mathbb{R}^{n \times 2}$ of size at least $2^{\Omega(d^{\Omega(1)})}/|G|^2$ according to Lemma 29 and normalize each function in the hard function family $C^{\mathcal{B}'}_\mathcal{F}$ to have $\|\cdot\|_\mathcal{N}$ norm 1. Let $\epsilon' > 0$ be a sufficiently small error constant. Then for any $g \in C^{\mathcal{B}'}_\mathcal{F}$, any CSQ algorithm that queries from oracles of concept $f \in C^{\mathcal{B}'}_\mathcal{F}$ and outputs a hypothesis $h$ with $\|f - h\|_\mathcal{N} \leq \epsilon'$ requires either $2^{n^{\Omega(1)}}/|G|^2$ queries or at least one query with precision $|G|2^{-n^{\Omega(1)}} + \sqrt{|G|}n^{-\Omega(k)}$.*

*Proof.* First we review properties of the constructed objects. By Lemma 29, $g\boldsymbol{B}$ and $g'\boldsymbol{B}'$ are almost orthogonal for any $\boldsymbol{B} \neq \boldsymbol{B}' \in \mathcal{B}'$ and $g, g' \in G$.

Then by Theorem 27 with matrix $\boldsymbol{A} = \boldsymbol{I}_d$, for any $\boldsymbol{B} \neq \boldsymbol{B}' \in \mathcal{B}'$ and any $g, g' \in G$, noting that $\boldsymbol{B}^\top(g^{-1}\boldsymbol{X}) = (g\boldsymbol{B})^\top \boldsymbol{X}$ since a permutation matrix is orthonormal, we have

$$|\mathbb{E}_{\boldsymbol{X} \sim \mathcal{N}}[f^*_{\exp}((g\boldsymbol{B})^\top \boldsymbol{X})f^*_{\exp}((g'\boldsymbol{B}')^\top \boldsymbol{X})]| \leq O(n^{k(c-1/2)})\|f^*_{\exp}\|^2_\mathcal{N}, \tag{118}$$

for the same $c$ that was chosen in the statement of Lemma 29.

Now we verify that $\|g_{\boldsymbol{B}}\|_\mathcal{N} = \Omega(1)$ for each $g_{\boldsymbol{B}} \in C^{\mathcal{B}'}_{FA(G)}$. Let $\boldsymbol{B} = [\boldsymbol{x} \quad \boldsymbol{y}]$ and $\boldsymbol{B}_g := g\boldsymbol{B}, \boldsymbol{B}_{g'} := (g')\boldsymbol{B}$.

$$\|g_{\boldsymbol{B}}\|^2_\mathcal{N} \tag{119}$$

$$= \frac{1}{\|f^*_{\exp}\|^2_\mathcal{N}} \mathbb{E}_{\boldsymbol{X} \sim \mathcal{N}} \left[ \sum_{g \in G} \frac{(f^*_{\exp}(\boldsymbol{B}^\top_g \boldsymbol{X}))^2}{|G|} + \sum_{g \neq g' \in G} \frac{f^*_{\exp}((\boldsymbol{B}^\top_g \boldsymbol{X})f^*_{\exp}(\boldsymbol{B}^\top_{g'}\boldsymbol{X})}{|G|} \right] \tag{120}$$

$$= 1 + \frac{1}{|G|\|f^*_{\exp}\|^2_\mathcal{N}} \sum_{g \neq g' \in G} \mathbb{E}_{\boldsymbol{X} \sim \mathcal{N}} \left[ f^*_{\exp}(\boldsymbol{B}^\top_g \boldsymbol{X})f^*_{\exp}(\boldsymbol{B}^\top_{g'}\boldsymbol{X}) \right] \tag{121}$$

---

[4]The different notation highlights the fact that we are using a slightly different set of matrices from Lemma 23 and (Diakonikolas et al., 2020).

Now we split the sum into two parts. In the first part, we examine $g \neq g'$ such that $\max(|\langle g\boldsymbol{x}, g'\boldsymbol{x}\rangle|, |\langle g\boldsymbol{y}, g'\boldsymbol{y}\rangle|) < c_1$ for some constant $0 < c_1 < 1/\sqrt{3}$, and leave the remaining cases for the other part. Collect the pairs $g, g'$ in the first part into a set $G_1$. Then

$$\left| \frac{1}{|G| \|f_{\exp}^*\|_{\mathcal{N}}^2} \sum_{g \neq g' \in G_1} \mathbb{E}_{\boldsymbol{X} \sim \mathcal{N}} \left[ f_{\exp}^*(\boldsymbol{B}_g^\top \boldsymbol{X}) f_{\exp}^*(\boldsymbol{B}_{g'}^\top \boldsymbol{X}) \right] \right| \tag{122}$$

$$\leq \frac{1}{|G| \|f_{\exp}^*\|_{\mathcal{N}}^2} \sum_{g \neq g' \in G_1} \left| \mathbb{E}_{\boldsymbol{X} \sim \mathcal{N}} \left[ f_{\exp}^*(\boldsymbol{B}_g^\top \boldsymbol{X}) f_{\exp}^*(\boldsymbol{B}_{g'}^\top \boldsymbol{X}) \right] \right| \tag{123}$$

$$\leq \frac{1}{|G|} \sum_{g \neq g' \in G} \|\boldsymbol{B}_g^\top \boldsymbol{B}_{g'}\|_2^k, \tag{124}$$

where $k$ is the number of neurons in the hidden layer of $f_{\exp}^*$. The second line comes from the triangle inequality and the last line comes from Theorem 27.

We have,

$$\|\boldsymbol{B}_g^\top \boldsymbol{B}_{g'}\|_2^k \leq \|\boldsymbol{B}_g^\top \boldsymbol{B}_{g'}\|_F^k \tag{125}$$

$$= \left( 2z^2 + \langle g\boldsymbol{x}, (g')\boldsymbol{x}\rangle^2 + \langle g\boldsymbol{y}, (g')\boldsymbol{y}\rangle^2 \right)^{k/2} \tag{126}$$

$$\leq 3^{k/2-1} \left( (2z)^k + |\langle g\boldsymbol{x}, (g')\boldsymbol{x}\rangle|^k + |\langle g\boldsymbol{y}, (g')\boldsymbol{y}\rangle|^k \right). \tag{127}$$

The last line uses power mean inequality. By our assumptions,

$$3^{k/2-1} \sum_{g \neq g' \in G} |\langle g\boldsymbol{x}, (g')\boldsymbol{x}\rangle|^k < |G|(|G|-1) \frac{(c_1\sqrt{3})^k}{3} < |G|(|G|-1)(c_1\sqrt{3})^k \tag{128}$$

and similarly $3^{k/2-1} \sum_{g \neq g' \in G} |\langle g\boldsymbol{y}, (g')\boldsymbol{y}\rangle|^k \leq |G|(|G|-1)(c_1\sqrt{3})^k$.

Plugging back,

$$\frac{1}{|G| \|f_{\exp}^*\|_{\mathcal{N}}^2} \left| \sum_{g \neq g' \in G_1} \mathbb{E}_{\boldsymbol{X} \sim \mathcal{N}} \left[ f_{\exp}^*(\boldsymbol{B}_g^\top \boldsymbol{X}) f_{\exp}^*(\boldsymbol{B}_{g'}^\top \boldsymbol{X}) \right] \right| \leq (|G|-1)(c_1\sqrt{3})^k. \tag{129}$$

Since $|G| = \mathrm{poly}(n)$ and $k = \Theta(n)$, the final expression is $o(1)$ in $n$.

We now consider the remaining part which is reproduced here:

$$\frac{1}{|G| \|f_{\exp}^*\|_{\mathcal{N}}^2} \sum_{g \neq g' \in G^2 \setminus G_1} \mathbb{E}_{\boldsymbol{X} \sim \mathcal{N}} \left[ f_{\exp}^*((g\boldsymbol{B})^\top \boldsymbol{X}) f_{\exp}^*((g'\boldsymbol{B})^\top \boldsymbol{X}) \right] \tag{130}$$

Zooming in, we have:

$$\sum_{g \neq g' \in G^2 \setminus G_1} \mathbb{E}_{\boldsymbol{X} \sim \mathcal{N}} \left[ f_{\exp}^*(\boldsymbol{B}_g^\top \boldsymbol{X}) f_{\exp}^*(\boldsymbol{B}_{g'}^\top \boldsymbol{X}) \right] \tag{131}$$

$$= \sum_{g \neq g' \in G^2 \setminus G_1} \sum_{m=0}^{\infty} \frac{1}{m!} \sum_{v_1, \ldots, v_m = 1}^{d} \sum_{j_1, j_1', \ldots, j_m, j_m' = 1}^{2}$$
$$\partial_{v_1, j_1} \ldots \partial_{v_m, j_m} (f_{\exp}^*)^{[m]}(\cdot) \partial_{v_1, j_1'} \ldots \partial_{v_m, j_m'} (f_{\exp}^*)^{[m]}(\cdot) (\boldsymbol{B}_g^\top \boldsymbol{B}_{g'})_{j_1 j_1'} \ldots (\boldsymbol{B}_g^\top \boldsymbol{B}_{g'})_{j_m j_m'} \tag{132}$$

$$= \sum_{g \neq g' \in G^2 \setminus G_1} \sum_{m=k}^{\infty} \frac{1}{m!} \sum_{v_1, \ldots, v_m = 1}^{d} \sum_{j_1, j_1', \ldots, j_m, j_m' = 1}^{2}$$
$$\partial_{v_1, j_1} \ldots \partial_{v_m, j_m} (f_{\exp}^*)^{[m]}(\cdot) \partial_{v_1, j_1'} \ldots \partial_{v_m, j_m'} (f_{\exp}^*)^{[m]}(\cdot) (\boldsymbol{B}_g^\top \boldsymbol{B}_{g'})_{j_1 j_1'} \ldots (\boldsymbol{B}_g^\top \boldsymbol{B}_{g'})_{j_m j_m'}, \tag{133}$$

where the second line comes from Corollary 25 and the last line is because all low-degree moment of $f_{\exp}^*$ vanishes.

Further zooming in, for each $g \neq g' \in G^2 \backslash G_1$, each $m \geq k$ and each $v_1, \ldots, v_m \in [2]$, denote $p_J$ as $\partial_{v_1, J_1} \ldots \partial_{v_m, J_m}(f_{\exp}^*)^{[m]}(\cdot)$ for each $J \in [2]^m$, and the corresponding vector $p \in \mathbb{R}^{2^m}$. We have, using Einstein notation to sum over $j_1, \ldots j_m, j_1', \ldots, j_m'$:

$$\partial_{v_1, j_1} \ldots \partial_{v_m, j_m}(f_{\exp}^*)^{[m]}(\cdot)\partial_{v_1, j_1'} \ldots \partial_{v_m, j_m'}(f_{\exp}^*)^{[m]}(\cdot)(\boldsymbol{B}_g^\top \boldsymbol{B}_{g'})_{j_1 j_1'} \ldots (\boldsymbol{B}_g^\top \boldsymbol{B}_{g'})_{j_m j_m'}, = \boldsymbol{p}^\top (\boldsymbol{B}_g^\top \boldsymbol{B}_{g'})^{\otimes m} \boldsymbol{p} \tag{134}$$

By Corollary 43, it suffices to show that $\boldsymbol{C} := \boldsymbol{B}_g^\top \boldsymbol{B}_{g'}$ satisfies $\boldsymbol{z}^\top \boldsymbol{C} \boldsymbol{z} \geq 0$ for all $\boldsymbol{z} \in \mathbb{R}^2$. However, note that by concentration results of Lemma 29, for any small $\epsilon \in (0, 1)$,

$$\langle g\boldsymbol{x}, g'\boldsymbol{x} \rangle \in \left[ \sqrt{1-\epsilon}\bar{F}_{g,g'} - \delta_{\boldsymbol{x}}, \frac{\bar{F}_{g,g'}}{\sqrt{1-\epsilon}} + \delta_{\boldsymbol{x}} \right] \tag{135}$$

$$\langle g\boldsymbol{y}, g'\boldsymbol{y} \rangle \in \left[ \sqrt{1-\epsilon}\bar{F}_{g,g'} - \delta_{\boldsymbol{y}}, \frac{\bar{F}_{g,g'}}{\sqrt{1-\epsilon}} + \delta_{\boldsymbol{y}} \right] \tag{136}$$

$$\langle g\boldsymbol{x}, g'\boldsymbol{y} \rangle = \epsilon_1 \tag{137}$$

$$\langle g\boldsymbol{y}, g'\boldsymbol{x} \rangle = \epsilon_2, \tag{138}$$

for $\delta_{\boldsymbol{x}}, \delta_{\boldsymbol{y}} \geq 0, \max(\delta_{\boldsymbol{x}}, \delta_{\boldsymbol{y}}, |\epsilon_1|, |\epsilon_2|) < O(n^{c-1/2})$ and $\bar{F}_{g,g'} = F_{g,g'}/n$ with $F_{g,g'}$ denotes the number of fixed points of $g^{-1}g'$ as in Lemma 29. Note that $1 \geq \bar{F}_{g,g'} > c_1$ is bounded away from 0 (by definition of $G_1$ and concentration result in Lemma 29). Choose $\epsilon$ small enough that $\sqrt{1-\epsilon}\bar{F}_{g,g'} =: c_2, c_2 - \delta_{\boldsymbol{x}}, c_2 - \delta_{\boldsymbol{y}}$ are all bounded away from 0 when $n$ is large enough. Thus, for any $\boldsymbol{z} \in \mathbb{R}^2$,

$$\boldsymbol{z}^\top \boldsymbol{C} \boldsymbol{z} \geq \boldsymbol{z}_1^2(c_2 - \delta_{\boldsymbol{x}}) + \boldsymbol{z}_2^2(c_2 - \delta_{\boldsymbol{y}}) + \boldsymbol{z}_1 \boldsymbol{z}_2(\epsilon_1 + \epsilon_2) \tag{139}$$

$$= (c_2 - \delta_{\boldsymbol{x}})\left( \boldsymbol{z}_1 - \frac{\epsilon_1 + \epsilon_2}{c_2 - \delta_{\boldsymbol{x}}}\boldsymbol{z}_2 \right)^2 + \boldsymbol{z}_2^2 \left( c_2 - \delta_{\boldsymbol{y}} - \frac{(\epsilon_1 + \epsilon_2)^2}{c_2 - \delta_{\boldsymbol{x}}} \right), \tag{140}$$

which is clearly nonnegative for large enough $n$ as $c_2$ is bounded away from 0. We conclude that $\boldsymbol{p}^\top (\boldsymbol{B}_g^\top \boldsymbol{B}_{g'})^{\otimes m} \boldsymbol{p}$ is nonnegative and thus removing these terms leads to a lower bound on $\|g_{\boldsymbol{B}}\|_{\mathcal{N}}^2$.

For pairwise correlation, we have:

$$|\langle g_{\boldsymbol{B}}, g_{\boldsymbol{B}'} \rangle_{\mathcal{N}}| \leq \frac{1}{|G| \cdot \|f_{\exp}^*\|_{\mathcal{N}}^2} \left| \mathbb{E}_{\boldsymbol{X} \sim \mathcal{N}} \sum_{g, g' \in G} f_{\exp}^*((g\boldsymbol{B})^\top \boldsymbol{X}) f_{\exp}^*((g'\boldsymbol{B}')^\top \boldsymbol{X}) \right| \tag{141}$$

$$\leq \frac{1}{|G| \cdot \|f_{\exp}^*\|_{\mathcal{N}}^2} \sum_{g, g' \in G} |\mathbb{E}_{\boldsymbol{X} \sim \mathcal{N}} f_{\exp}^*((g\boldsymbol{B})^\top \boldsymbol{X}) f_{\exp}^*((g'\boldsymbol{B}')^\top \boldsymbol{X})| \tag{142}$$

$$\leq O(n^{k(c-1/2)})|G|. \tag{143}$$

Now we replace each $g_{\boldsymbol{B}}$ by its normalization to make $\|g_{\boldsymbol{B}}\|_{\mathcal{N}}^2 = 1$. Note that this only decreases (or increases by at most a constant factor) the correlation between different hard functions since $\|g_{\boldsymbol{B}}\|_{\mathcal{N}}^2 = \Omega(1)$. We are in position to apply Lemma 26. Let $\tau := \max_{\boldsymbol{B}_i \neq \boldsymbol{B}_j} |\langle g_{\boldsymbol{B}_i}, g_{\boldsymbol{B}_j} \rangle_{L^2(\mathcal{N})}|$. Computing average pairwise correlation of $C_{\mathcal{G}}^{\mathcal{B}'}$ yields:

$$\rho(C_{\mathcal{G}}^{\mathcal{B}'}) = 1/|\mathcal{B}'| + (1 - 1/|\mathcal{B}'|)\tau \leq 2/|\mathcal{B}'| + \tau. \tag{144}$$

Finally, setting $\tau' := 2/|\mathcal{B}'| + \tau$ gives $\text{SDA}(C_{\mathcal{G}}^{\mathcal{B}'}, \mathcal{N}, \tau') = 2^{\Omega(n^c)}/|G|^2$ and an application of Lemma 26 completes the proof.

$\square$

## F.2 SUPERPOLYNOMIAL LOWER BOUNDS

Throughout, we will use the notation $\subseteq_k$ to denote a subset of size $k$. The technique from this part comes mainly from (Goel et al., 2020). Define:

$$f_S : \mathbb{R}^{n \times d} \to \mathbb{R}, \text{ where } f_S(\boldsymbol{X}) = \boldsymbol{1}_d^\top \left( \sum_{\boldsymbol{w} \in \{-1,1\}^k} \chi(\boldsymbol{w}) \sigma(\langle \boldsymbol{w}, \boldsymbol{X}_S \rangle / \sqrt{k}) \right), \qquad S \subseteq_k [n].$$
(145)

where $\chi$ is the parity function $\chi(\boldsymbol{w}) = \prod_i w_i$, and $\boldsymbol{X}_S \in \mathcal{X}^k$, or alternatively $\boldsymbol{X}_S \in \mathbb{R}^{k \times d}$, denotes the rows of $\boldsymbol{X}$ indexed by $S$. Here, $\langle \boldsymbol{w}, \boldsymbol{X}_S \rangle := \sum_{i=1}^k w_i (\boldsymbol{X}_S)_{i,:} = w \cdot \boldsymbol{X}_S$ and $\sigma : \mathbb{R}^d \to \mathbb{R}^d$ is an activation function. Throughout, we will set $k \leq \log_2 n$ so that the above network has at most $O(n)$ hidden nodes. For any $\boldsymbol{z} \in \{-1,1\}^n$, let $\boldsymbol{X} \circ \boldsymbol{z}$ be the matrix in $\mathbb{R}^{n \times d}$ whose $v$-th row is $z_v \boldsymbol{X}_{v,:}$. We will use the following lemma, which is a basic consequence of properties proven in Goel et al. (2020), but include the proofs here to be self-contained.

**Lemma 31.** *For all $S \neq T \subseteq_k [n]$, $g \in G$, and $\boldsymbol{z} \in \{-1,1\}^n$, we have:*

1. $f_S(g \cdot (\boldsymbol{X} \circ \boldsymbol{z})) = \chi_{g^{-1} \cdot S}(\boldsymbol{z}) f_S(g \cdot \boldsymbol{X}) = \chi_{g^{-1} \cdot S}(\boldsymbol{z}) f_{g^{-1} \cdot S}(\boldsymbol{X})$.

2. $\|f_S\|_{L^2(\mathcal{N})} \geq \Omega(e^{-\Theta(k)})$ *and* $\langle f_S, f_T \rangle_{L^2(\mathcal{N})} = 0$.

*Proof.* Fix $S \neq T \subseteq_k [n]$.

1. By definition:

$$f_S(\boldsymbol{X}) = \sum_{j=1}^d \sum_{\boldsymbol{w} \in \{-1,1\}^k} \chi(\boldsymbol{w}) \sigma(\langle \boldsymbol{w}, (\boldsymbol{X}_{:,j})_S \rangle / \sqrt{k})$$
(146)

where $\boldsymbol{X}_{:,j}$ is the $j$-th column of $\boldsymbol{X}$. $G$ acts on the rows of $\boldsymbol{X}$, i.e. $(g\boldsymbol{X})_{i,:} = \boldsymbol{X}_{g^{-1}i,:}$. Therefore, letting the order of $S$ be fixed, we have $(g\boldsymbol{X})_S = \boldsymbol{X}_{g^{-1}S,:}$.

We have, for any $g \in G$ and any $\boldsymbol{z} \in \{-1,1\}^n$:

$$f_S(g \cdot (\boldsymbol{X} \circ \boldsymbol{z})) = \sum_{j=1}^d \sum_{\boldsymbol{w} \in \{-1,1\}^k} \chi(\boldsymbol{w}) \sigma(\langle \boldsymbol{w}, ((g \cdot (\boldsymbol{X} \circ \boldsymbol{z}))_{:,j})_S \rangle / \sqrt{k})$$
(147)

$$= \sum_{j=1}^d \sum_{\boldsymbol{w} \in \{-1,1\}^k} \chi(\boldsymbol{w}) \sigma(\langle \boldsymbol{w}, ((\boldsymbol{X} \circ \boldsymbol{z})_{:,j})_{g^{-1} \cdot S} \rangle / \sqrt{k})$$
(148)

$$= \sum_{j=1}^d \sum_{\boldsymbol{w} \in \{-1,1\}^k} \chi(\boldsymbol{w}) \sigma(\langle \boldsymbol{w}, (\boldsymbol{X}_{:,j})_{g^{-1} \cdot S} \circ \boldsymbol{z}_{g^{-1} \cdot S} \rangle / \sqrt{k})$$
(149)

$$= \chi_{g^{-1} \cdot S}(\boldsymbol{z}) \sum_{j=1}^d \sum_{\boldsymbol{w} \in \{-1,1\}^k} \chi(\boldsymbol{w} \circ \boldsymbol{z}_{g^{-1} \cdot S}) \sigma(\langle \boldsymbol{w} \circ \boldsymbol{z}_{g^{-1} \cdot S}, (\boldsymbol{X}_{:,j})_{g^{-1} \cdot S} \rangle / \sqrt{k})$$
(150)

$$= \chi_{g^{-1} \cdot S}(\boldsymbol{z}) \sum_{j=1}^d \sum_{\boldsymbol{w} \in \{-1,1\}^k} \chi(\boldsymbol{w}) \sigma(\langle \boldsymbol{w}, (\boldsymbol{X}_{:,j})_{g^{-1} \cdot S} \rangle / \sqrt{k})$$
(151)

$$= \chi_{g^{-1} \cdot S}(\boldsymbol{z}) f_{g^{-1} \cdot S}(\boldsymbol{X})$$
(152)

$$= \chi_{g^{-1} \cdot S}(\boldsymbol{z}) f_S(g \cdot \boldsymbol{X}).$$
(153)

In the above derivation, we use the definition of $f_S$ to get the first line. The second line comes from the fact that permuting by $g$, then taking the subset $S$ is equivalent to taking the subset $g^{-1} \cdot S$. The third line comes from the fact that each element of $\boldsymbol{z}$ is a scalar. The fourth line comes from the fact that $\chi(\boldsymbol{w}) = \chi_{g^{-1} \cdot S}(\boldsymbol{z}) \cdot \chi_{g^{-1} \cdot S}(\boldsymbol{w} \circ \boldsymbol{z}_{g^{-1} \cdot S})$. The fifth line is a change of variable $\boldsymbol{w} \mapsto \boldsymbol{w} \circ \boldsymbol{z}_{g^{-1} \cdot S}$. Finally, the last line comes from the definition of $f_S$, and noting that the dependence of $f_S$ on $S$ only comes from which row of $\boldsymbol{X}$ is selected by $S$; thus, $f_{g^{-1} \cdot S}(\boldsymbol{X}) = f_S(g \cdot \boldsymbol{X})$.

2. We have, for any $S \neq T$:

$$\langle f_S, f_T \rangle_{\mathcal{L}^2(\mathcal{N})} = \mathbb{E}_{\boldsymbol{X} \sim \mathcal{L}^2(\mathcal{N})}[f_S(\boldsymbol{X}) f_T(\boldsymbol{X})] \tag{154}$$

$$= \mathbb{E}_{\boldsymbol{z} \sim \mathrm{Unif}(\{-1,1\}^n)} \mathbb{E}_{\boldsymbol{X}}[f_S(\boldsymbol{X} \circ \boldsymbol{z}) f_T(\boldsymbol{X} \circ \boldsymbol{z})] \tag{155}$$

$$= \mathbb{E}_{\boldsymbol{z} \sim \mathrm{Unif}(\{-1,1\}^n)} \mathbb{E}_{\boldsymbol{X}}[f_S(\boldsymbol{X}) f_T(\boldsymbol{X}) \chi_S(\boldsymbol{z}) \chi_T(\boldsymbol{z})] \tag{156}$$

$$= \mathbb{E}_{\boldsymbol{X}}[f_S(\boldsymbol{X}) f_T(\boldsymbol{X})] \mathbb{E}_{\boldsymbol{z} \sim \mathrm{Unif}(\{-1,1\}^n)}[\chi_S(\boldsymbol{z}) \chi_T(\boldsymbol{z})] \tag{157}$$

$$= 0 \tag{158}$$

by sign-invariance of $\mathcal{N}$ and orthogonality of parity functions under $\mathrm{Unif}(\{-1,1\}^n)$. When $S = T$, the same argument as Goel et al. (2020) Theorem 3.8 shows that $f_S$ does not vanish.

$\square$

Now, define the family of hard functions as:

$$\mathcal{H}_{\mathcal{S}, \mathcal{F}} := \left\{ \tilde{f}_S : \boldsymbol{X} \mapsto \frac{1}{\sqrt{|\mathcal{F}(\boldsymbol{X})|}} \sum_{g \in \mathcal{F}(\boldsymbol{X})} f_S(g \cdot \boldsymbol{X}) \mid [n] \supseteq_k S \in \mathcal{S} \right\}, \tag{159}$$

for some set $\mathcal{S}$ depending on $\mathcal{F}$ and $G$. The subsets in $\mathcal{S}$ such that these hard functions are orthogonal will determine the size of the family of hard functions, and therefore the CSQ dimension lower bound.

**Definition 32** (Sign-invariant frame). A frame $\mathcal{F}$ is **sign-invariant** if $\mathcal{F}(\boldsymbol{X}) = \mathcal{F}(\boldsymbol{X} \circ \boldsymbol{z})$ for almost all $\boldsymbol{X}$ (under the given data distribution) and for all $\boldsymbol{z} \in \{-1, 1\}^n$.

**Lemma 33.** *If $\mathcal{F}$ is any sign-invariant frame, then there exists $\mathcal{S}$ with $|\mathcal{S}| \geq \Omega(n^{\Omega(\log n)})/M(n)$ such that the functions in $\mathcal{H}_{\mathcal{S}, \mathcal{F}}$ are pairwise orthogonal, where $M(n)$ is the size of the largest orbit of $\binom{[n]}{m}$ where $m = \Omega(\log n)$ under $G$. If moreover $|\mathcal{F}(\boldsymbol{X})| = 1$ for almost all $x$, then $|\mathcal{S}| = \Omega(n^{\Omega(\log n)})$.*

*Proof.* We have for any $S \neq T$:

$$\langle \tilde{f}_S, \tilde{f}_T \rangle_{L^2(\mathcal{N})} = \mathbb{E}_{\boldsymbol{X} \sim \mathcal{N}} \frac{1}{|\mathcal{F}(\boldsymbol{X})|} \sum_{g, g' \in \mathcal{F}(\boldsymbol{X})} f_S(g \cdot \boldsymbol{X}) f_T(g' \cdot \boldsymbol{X}) \tag{160}$$

$$= \mathbb{E}_{\boldsymbol{z} \sim \mathrm{Unif}\{-1,1\}^n} \mathbb{E}_{\boldsymbol{X} \sim \mathcal{N}} \frac{1}{|\mathcal{F}(\boldsymbol{X} \circ \boldsymbol{z})|} \sum_{g, g' \in \mathcal{F}(\boldsymbol{X} \circ \boldsymbol{z})} f_S(g \cdot (\boldsymbol{X} \circ \boldsymbol{z})) f_T(g' \cdot (\boldsymbol{X} \circ \boldsymbol{z})) \tag{161}$$

$$= \mathbb{E}_{\boldsymbol{z} \sim \mathrm{Unif}\{-1,1\}^n} \mathbb{E}_{\boldsymbol{X} \sim \mathcal{N}} \frac{1}{|\mathcal{F}(\boldsymbol{X} \circ \boldsymbol{z})|}$$
$$\sum_{g, g' \in \mathcal{F}(\boldsymbol{X} \circ \boldsymbol{z})} \chi_{g^{-1} \cdot S}(\boldsymbol{z}) f_{g^{-1} \cdot S}(\boldsymbol{X}) \chi_{(g')^{-1} \cdot T}(\boldsymbol{z}) f_{(g')^{-1} \cdot T}(\boldsymbol{X}) \tag{162}$$

$$= \mathbb{E}_{\boldsymbol{z} \sim \mathrm{Unif}\{-1,1\}^n} \mathbb{E}_{\boldsymbol{X} \sim \mathcal{N}} \frac{1}{|\mathcal{F}(\boldsymbol{X})|}$$
$$\sum_{g, g' \in \mathcal{F}(\boldsymbol{X})} \chi_{g^{-1} \cdot S}(\boldsymbol{z}) f_{g^{-1} \cdot S}(\boldsymbol{X}) \chi_{(g')^{-1} \cdot T}(\boldsymbol{z}) f_{(g')^{-1} \cdot T}(\boldsymbol{X}) \tag{163}$$

$$= \mathbb{E}_{\boldsymbol{X} \sim \mathcal{N}} \frac{1}{|\mathcal{F}(\boldsymbol{X})|}$$
$$f_{g^{-1} \cdot S}(\boldsymbol{X}) f_{(g')^{-1} \cdot T}(\boldsymbol{X}) \sum_{g, g' \in \mathcal{F}(\boldsymbol{X})} \mathbb{E}_{\boldsymbol{z} \sim \mathrm{Unif}\{-1,1\}^n} \chi_{g^{-1} \cdot S}(\boldsymbol{z}) \chi_{(g')^{-1} \cdot T}(\boldsymbol{z}) \tag{164}$$

Here, we have used that the frame is sign-invariant to replace $\mathcal{F}(\boldsymbol{X} \circ \boldsymbol{z})$ with simply $\mathcal{F}(\boldsymbol{X})$, and rearrange the expectations accordingly. Now, if we have no further assumptions on $\mathcal{F}$, then we

require $S$ and $T$ to be $m = \Omega(\log n)$-sized subsets in **different** orbits (in other words, $S \neq gT$ for all $g \in G$). Then, by Lemma 31, $g^{-1}S \neq (g')^{-1}T$ for any $g$ and $g'$, and the inner expectation over $z$ (and therefore the entire expression) is 0. This yields a lower bound on $|\mathcal{S}|$ of $\Omega(n^{\Omega(\log n)})/M(n)$, from choosing one $\log n$-sized subset from each orbit.

If we moreover have that $\mathcal{F}$ has size one almost everywhere, then $S \neq T$ suffices to ensure orthogonality, and we can set $|\mathcal{S}| = n^m$ where $m = \Theta(\log n)$ by letting $\mathcal{S}$ be all $\Theta(\log n)$-sized subsets of $[n]$. Note that if subsets are size $\Theta(\log n)$, the hidden width of the network is polynomial and furthermore, if $m \leq \log_2 n$ then the width is $O(n)$. □

**Remark 34.** Note that the definition of frame always allows one to choose a singleton frame $\mathcal{F}(X)$ for any $X$ whose stabilizer is trivial under the action of $G$ on $\mathcal{R}^{n \times d}$. For any other $X$ with nontrivial stabilizer, since $G$ is a subgroup of the permutation group, $X$ must have at least 2 equal entries and thus lies in a manifold of dimensions strictly smaller than $n \times d$. Density of $\mathcal{N}$ thus ensures that $\Pr_{X \sim \mathcal{N}}[|\mathcal{F}(X)| = 1] = 1$ as long as we choose the singleton frame at every $X$ without self-symmetry. A practical choice of frame in this case, which is also sign-invariant, is the frame $\mathcal{F}(X)$ which is the set of all permutations that lexicographically sort $X$ by absolute value. It is not hard to see that with probability 1 under $\mathcal{N}$, there is only 1 permutation that sorts $X$.

**Remark 35.** The case of sign-invariant frames includes $\mathcal{F}(X) = G$, the Reynolds operator.

**Corollary 36.** *Let $\mathcal{F}$ be a sign-invariant frame. For any $c > 0$, any CSQ algorithm that queries from an oracle of concept $f \in \mathcal{H}_{\mathcal{S},\mathcal{F}}$ and outputs a hypothesis $h$ with $\|f - h\|_{\mathcal{N}} \leq \Theta(1)$ needs at least $\Omega(n^{\Omega(\log n)})/M(n))$ queries or at least one query with precision $o(n^{-c/2})$.*

**Corollary 37.** *Let $\mathcal{F}$ be a sign-invariant singleton frame. For any $c > 0$, any CSQ algorithm that queries from an oracle of concept $f \in \mathcal{H}_{\mathcal{S},\mathcal{F}}$ and outputs a hypothesis $h$ with $\|f - h\|_{\mathcal{N}} \leq \Theta(1)$ needs at least $\Omega(n^{\Omega(\log n)})$ queries or at least one query with precision $o(n^{-c/2})$.*

*Proof.* These corollaries are immediate applications of Theorem 4.1 and Lemma 2.6 from Goel et al. (2020) to Lemma 33. □

It remains to show that the norms of the hard functions in $\mathcal{H}_{\mathcal{S},\mathcal{F}}$ do not vanish. We show this below for sign-invariance, almost surely polynomial frames.

**Lemma 38.** *Let $\mathcal{H}_{\mathcal{S},\mathcal{F}}$ be the hard functions derived in Lemma 33 from the class of hard functions $\mathcal{H}_{\mathcal{S}}$ in Eq. (145) (which comes from Goel et al. (2020), but with an extra input dimension). Let $\langle \mathbb{E}[f_S^2] \rangle_{\mathrm{Unif}(\mathcal{H}_{\mathcal{S}})}$ denote the average squared norm of functions uniformly drawn from $\mathcal{H}_{\mathcal{S}}$ and let $M_{\mathcal{H}_{\mathcal{S},\mathcal{F}}} = \max_{\tilde{f}_S \in \mathcal{H}_{\mathcal{S},\mathcal{F}}} \mathbb{E}[\tilde{f}_S(X)^2]$ denote the maximum squared norm of the hard functions in $\mathcal{H}_{\mathcal{S},\mathcal{F}}$. If $M_{\mathcal{H}_{\mathcal{S},\mathcal{F}}}/\langle \mathbb{E}[f_S^2] \rangle_{\mathrm{Unif}(\mathcal{H}_{\mathcal{S}})} = O(\mathrm{poly}(n))$, then there exists a constant $c > 0$ and subset $\mathcal{C} \subset \mathcal{H}_{\mathcal{S},\mathcal{F}}$ of size $|\mathcal{C}| = \Omega(|\mathcal{H}_{\mathcal{S},\mathcal{F}}|/\mathrm{poly}(n))$ many functions whose norms are at least $c\langle \mathbb{E}[f_S^2] \rangle_{\mathrm{Unif}(\mathcal{H}_{\mathcal{S}})}$.*

*Proof.* We use a second moment method argument to prove the above statement. Let the random variable $Z$ denote the squared norm $\mathbb{E}[(\tilde{f}_S(X))^2]$ of a function drawn from the distribution $\mathrm{Unif}(\mathcal{H}_{\mathcal{S},\mathcal{F}})$ which is uniform over all functions in $\mathcal{H}_{\mathcal{S},\mathcal{F}}$. The Paley-Zygmund inequality states that for any given $\theta \in [0, 1]$

$$\mathbb{P}(Z > \theta \mathbb{E}[Z]) \geq (1 - \theta)^2 \frac{\mathbb{E}[Z]^2}{\mathbb{E}[Z^2]}. \tag{165}$$

The probability above calculates the portion of functions in $\mathrm{Unif}(\mathcal{H}_{\mathcal{S},\mathcal{F}})$ which have non-vanishing norm.

Now, we calculate the first two moments of $Z$ over the $|\mathcal{H}_{\mathcal{S},\mathcal{F}}| = \binom{n}{k}$ functions in the class where $k = \Theta(\log n)$ as in the construction in Goel et al. (2020). For the first moment,

$$\mathbb{E}_{\boldsymbol{X}\sim\mathcal{N}}[Z] = \frac{1}{|\mathcal{H}_{\mathcal{S},\mathcal{F}}|} \sum_{\tilde{f}_S \in \mathcal{H}_{\mathcal{S},\mathcal{F}}} \mathbb{E}_{\boldsymbol{X}\sim\mathcal{N}}[\tilde{f}_S(\boldsymbol{X})^2] \tag{166}$$

$$= \binom{n}{k}^{-1} \mathbb{E}_{\boldsymbol{X}\sim\mathcal{N}} \left[ \sum_{S\subseteq_k[n]} \frac{1}{|\mathcal{F}(\boldsymbol{X})|} \sum_{g,g'\in\mathcal{F}(\boldsymbol{X})} f_S(g\cdot\boldsymbol{X})f_S(g'\cdot\boldsymbol{X}) \right] \tag{167}$$

$$= \binom{n}{k}^{-1} \mathbb{E}_{\boldsymbol{X}\sim\mathcal{N}} \left[ \frac{1}{|\mathcal{F}(\boldsymbol{X})|} \sum_{S\subseteq_k[n]} \left( \sum_{g\in\mathcal{F}(\boldsymbol{X})} f_S(\boldsymbol{X})^2 + \sum_{g\neq g'\in\mathcal{F}(\boldsymbol{X})} f_S(g\cdot\boldsymbol{X})f_S(g'\cdot\boldsymbol{X}) \right) \right] \tag{168}$$

$$\geq \binom{n}{k}^{-1} \mathbb{E}_{\boldsymbol{X}\sim\mathcal{N}} \left[ \frac{1}{|\mathcal{F}(\boldsymbol{X})|} \sum_{g\in\mathcal{F}(\boldsymbol{X})} \sum_{S\subseteq_k[n]} f_S(g\cdot\boldsymbol{X})^2 \right]$$
$$- \binom{n}{k}^{-1} \left| \sum_{S\subseteq_k[n]} \mathbb{E}_{\boldsymbol{X}\sim\mathcal{N}} \left[ \frac{1}{|\mathcal{F}(\boldsymbol{X})|} \sum_{\substack{g,g'\in\mathcal{F}(\boldsymbol{X}) \\ g^{-1}S\neq(g')^{-1}S}} f_{g^{-1}S}(\boldsymbol{X})f_{(g')^{-1}S}(\boldsymbol{X}) \right] \right| \tag{169}$$

$$= \binom{n}{k}^{-1} \mathbb{E}_{\boldsymbol{X}\sim\mathcal{N}} \left[ \frac{1}{|\mathcal{F}(\boldsymbol{X})|} \sum_{g\in\mathcal{F}(\boldsymbol{X})} \sum_{S\subseteq_k[n]} f_S(g\cdot\boldsymbol{X})^2 \right]$$
$$- \binom{n}{k}^{-1} \left| \sum_{S\subseteq_k[n]} \sum_{\substack{g,g'\in G \\ g^{-1}S\neq(g')^{-1}S}} \mathbb{E}_{\boldsymbol{X}\sim\mathcal{N}} \left[ \frac{\delta_{g,g'\in\mathcal{F}(\boldsymbol{X})}}{|\mathcal{F}(\boldsymbol{X})|} f_{g^{-1}S}(\boldsymbol{X})f_{(g')^{-1}S}(\boldsymbol{X}) \right] \right|, \tag{170}$$

where in the third line, we had the inequality $\geq$ since we ignore $g\neq g'$ such that $g^{-1}S = (g')^{-1}S$ (the summands in this case is $f_{g^{-1}S}(\boldsymbol{X})^2 \geq 0$ so ignoring them gives a lower bound) and then use reverse triangle inequality $|a+b| \geq |a| - |b|$.

The first term is

$$\binom{n}{k}^{-1} \mathbb{E}_{\boldsymbol{X}\sim\mathcal{N}} \left[ \frac{1}{|\mathcal{F}(\boldsymbol{X})|} \sum_{g\in\mathcal{F}(\boldsymbol{X})} \sum_{S\subseteq_k[n]} f_S(g\cdot\boldsymbol{X})^2 \right] = \binom{n}{k}^{-1} \mathbb{E}_{\boldsymbol{X}\sim\mathcal{N}} \left[ \sum_{S\subseteq_k[n]} f_S(\boldsymbol{X})^2 \right] = \langle\mathbb{E}[f_S^2]\rangle_{\mathrm{Unif}(\mathcal{H}_{\mathcal{S}})}. \tag{171}$$

Now we show that the second term is 0. For a fix $S \subset_k [n], g, g' \in G$ such that $g^{-1}S \neq (g')^{-1}S$, we have:

$$\mathbb{E}_{\boldsymbol{X}\sim\mathcal{N}} \left[ \frac{\delta_{g,g'\in\mathcal{F}(\boldsymbol{X})}}{|\mathcal{F}(\boldsymbol{X})|} f_{g^{-1}S}(\boldsymbol{X})f_{(g')^{-1}S}(\boldsymbol{X}) \right]$$
$$= \mathbb{E}_{\boldsymbol{z}\sim\mathrm{Unif}(\{-1,1\}^n)} \left[ \mathbb{E}_{\boldsymbol{X}\sim\mathcal{N}} \left[ \frac{\delta_{g,g'\in\mathcal{F}(\boldsymbol{X}\circ\boldsymbol{z})}}{|\mathcal{F}(\boldsymbol{X}\circ\boldsymbol{z})|} f_{g^{-1}S}(\boldsymbol{X}\circ\boldsymbol{z})f_{(g')^{-1}S}(\boldsymbol{X}\circ\boldsymbol{z}) \right] \right]$$
$$= \mathbb{E}_{\boldsymbol{X}\sim\mathcal{N}} \left[ \frac{\delta_{g,g'\in\mathcal{F}(\boldsymbol{X})}}{|\mathcal{F}(\boldsymbol{X})|} f_{g^{-1}S}(\boldsymbol{X})f_{(g')^{-1}S}(\boldsymbol{X})\mathbb{E}_{\boldsymbol{z}}\left[\chi_{g^{-1}S}(\boldsymbol{z})\chi_{(g')^{-1}S}(\boldsymbol{z})\right] \right]$$
$$= 0,$$

where we have used the fact that the frame $\mathcal{F}$ is sign-invariant and the decomposition result from Equation (1) of (Goel et al., 2020). Thus the second term is 0.

For the second moment, we have

$$
\begin{aligned}
\mathbb{E}[Z^2] &= \frac{1}{|\mathcal{H}_{\mathcal{S},\mathcal{F}}|} \sum_{\tilde{f}_S \in \mathcal{H}_{\mathcal{S},\mathcal{F}}} \mathbb{E}[\tilde{f}_S(\boldsymbol{X})^2]^2 \\
&\leq \frac{1}{|\mathcal{H}_{\mathcal{S},\mathcal{F}}|} \sum_{\tilde{f}_S \in \mathcal{H}_{\mathcal{S},\mathcal{F}}} \max_{\tilde{f}_S \in \mathcal{H}_{\mathcal{S},\mathcal{F}}} \mathbb{E}[\tilde{f}_S(\boldsymbol{X})^2]^2 \\
&= M_{\mathcal{H}_{\mathcal{S},\mathcal{F}}}^2.
\end{aligned}
\tag{172}
$$

Thus, applying Paley-Zygmund, we have

$$
\mathbb{P}\left(Z > \theta \left\langle \mathbb{E}[f_S^2] \right\rangle_{\mathrm{Unif}(\mathcal{H}_S)}\right) \geq (1-\theta)^2 \frac{\mathbb{E}[Z]^2}{\mathbb{E}[Z^2]} \geq (1-\theta)^2 \left\langle \mathbb{E}[f_S^2] \right\rangle_{\mathrm{Unif}(\mathcal{H}_S)}^2 M_{\mathcal{H}_{\mathcal{S},\mathcal{F}}}^{-2}. \tag{173}
$$

Setting $c = (1-\theta)^2$ completes the proof. $\qquad\square$

**Corollary 39.** *Assuming that the activation is ReLU or sigmoid, the subset of non-vanishing functions in $\mathcal{H}_{\mathcal{S},\mathcal{F}}$ is superpolynomial in $n$.*

*Proof.* It suffices to show that $M_{\mathcal{H}_{\mathcal{S},\mathcal{F}}} / \left\langle \mathbb{E}[f_S^2] \right\rangle_{\mathrm{Unif}(\mathcal{H}_S)}$ is $O(\mathrm{poly}(n))$.

Theorem 3.8 of Goel et al. (2020) guarantees that $\left\langle \mathbb{E}[f_S^2] \right\rangle_{\mathrm{Unif}(\mathcal{H}_S)} \geq \Omega\left(e^{-\Theta(k)}\right) = \Omega\left(\mathrm{poly}(n)^{-1}\right)$ (since $k = \Theta(\log n)$).

To upper bound $M_{\mathcal{H}_{\mathcal{S},\mathcal{F}}}$, we uniformly bound, for each $\tilde{f}_S \in \mathcal{H}_{\mathcal{S},\mathcal{F}}$,

$$
\mathbb{E}_{\boldsymbol{X}\sim\mathcal{N}}[\tilde{f}_S(\boldsymbol{X})^2] = \mathbb{E}_{\boldsymbol{X}\sim\mathcal{N}}\left[ \left( \frac{1}{\sqrt{|\mathcal{F}(\boldsymbol{X})|}} \sum_{g\in\mathcal{F}(\boldsymbol{x})} \sum_{j=1}^{d} \sum_{w\in\{-1,1\}^k} \chi(w)\sigma\left( \frac{\langle w, (g\cdot\boldsymbol{X}_{:,j})_S\rangle}{\sqrt{k}} \right) \right)^2 \right]
\tag{174}
$$

$$
\leq \mathbb{E}_{\boldsymbol{X}\sim\mathcal{N}}\left[ \sqrt{|\mathcal{F}(\boldsymbol{X})|}2^k \sum_{g\in\mathcal{F}(\boldsymbol{X})} \sum_{j=1}^{d} \sum_{w\in\{-1,1\}^k} \left( \chi(w)\sigma\left( \frac{\langle w, (g\cdot\boldsymbol{X}_{:,j})_S\rangle}{\sqrt{k}} \right) \right)^2 \right]
\tag{175}
$$

$$
\leq \mathbb{E}_{\boldsymbol{X}\sim\mathcal{N}}\left[ \sqrt{|\mathcal{F}(\boldsymbol{X})|}2^k \sum_{g\in\mathcal{F}(\boldsymbol{X})} \sum_{j=1}^{d} \sum_{w\in\{-1,1\}^k} \frac{(\langle w, (g\cdot\boldsymbol{X}_{:,j})_S\rangle)^2}{k} \right]
\tag{176}
$$

$$
\leq \mathbb{E}_{\boldsymbol{X}\sim\mathcal{N}}\left[ \sqrt{|\mathcal{F}(\boldsymbol{X})|}2^k \sum_{g\in\mathcal{F}(\boldsymbol{X})} \sum_{j=1}^{d} \sum_{w\in\{-1,1\}^k} \sum_{s\in S}(g\cdot\boldsymbol{X}_{:,j})_s^2 \right]
\tag{177}
$$

$$
\leq \mathbb{E}_{\boldsymbol{X}\sim\mathcal{N}}\left[ |\mathcal{F}(\boldsymbol{X})|^{3/2}2^{2k}dk\boldsymbol{x}_{[1]}^2 \right] \leq \mathrm{poly}(n)\mathbb{E}_{\boldsymbol{X}}[\boldsymbol{x}_{[1]}^2]
\tag{178}
$$

where $\boldsymbol{x}_{[1]}$ is the largest entry in absolute value of $\boldsymbol{X}$. In the above, we use Cauchy-Schwarz inequality for the second and fourth line, while the third line uses the fact that $\mathrm{ReLU}(x)^2 \leq x^2$ for all $x \in \mathbb{R}$. Since with probability 1, $|\mathcal{F}(\boldsymbol{X})|$ is at most polynomial in $n$ by our assumption, it suffices to show that the order (either largest or smallest) statistic $\boldsymbol{x}_{[1]}$ has small moment when $\boldsymbol{X}$ is a iid Gaussian random vector of length $n \times d$.

Bounding this moment is a classical study in concentration inequalities. By an inequality of Talagrand (Chatterjee, 2014; Talagrand, 1994), we know that $\mathbb{E}_{\boldsymbol{x}}[\boldsymbol{x}_{[1]}^2] - (\mathbb{E}_{\boldsymbol{x}}[\boldsymbol{x}_{[1]}])^2 \leq C/\log(nd)$ for some constant $C$. The first moment is also bounded by $O(\sqrt{\log(nd)})$ and thus $\mathbb{E}_{\boldsymbol{x}}[\boldsymbol{x}_{[1]}^2] \leq O(\log(nd))$.

Thus $M_{\mathcal{H}_{\mathcal{S},\mathcal{F}}} = O(\mathrm{poly}(n))$ and our conclusions follow. $\qquad\square$

### F.3 Proof of Extra Tools

**Lemma 40** (Sub-Gaussian concentration for inner-product with permuted vector). *Let $G \leq S_n$ with at least 2 elements and fix arbitrary $g \neq g' \in G$. Then for any $c \in (0, 1/2)$:*

$$\Pr_{\boldsymbol{x}}\left[\left|\frac{\langle g\boldsymbol{x}, g'\boldsymbol{x}\rangle}{\|\boldsymbol{x}\|} - \mu_{g,g'}\right| \geq \Omega(n^c)\right] \leq \exp(-\Omega(n^{2c})), \tag{179}$$

*for some $\mu_{g,g'} \in \left[\frac{F_{g,g'}}{\sqrt{n+1}}, \frac{F_{g,g'}}{\sqrt{n}}\right]$, i.e. $\mu_{g,g'} = \Theta(F_{g,g'} n^{-1/2})$, and where $F_{g,g'}$ is the number of fixed points in $g^{-1}g'$ (i.e. the number of cycles of length 1 in its cycle representation).*

*Proof.* Let

$$f(\boldsymbol{x}) = \frac{\langle g\boldsymbol{x}, g'\boldsymbol{x}\rangle}{\|\boldsymbol{x}\|}, \tag{180}$$

where $\boldsymbol{x} \sim \mathcal{N}(0, \boldsymbol{I}_n)$.

Note that

$$\nabla f(\boldsymbol{x}) = \frac{1}{\|\boldsymbol{x}\|}(\boldsymbol{x}_h + \boldsymbol{x}_{h^{-1}}) - \frac{1}{\|\boldsymbol{x}\|}\frac{\langle g\boldsymbol{x}, g'\boldsymbol{x}\rangle}{\langle \boldsymbol{x}, \boldsymbol{x}\rangle}\boldsymbol{x}, \tag{181}$$

where $\boldsymbol{x}_h$ consists of $\boldsymbol{x}$ with entries reordered by $h = g^{-1}g'$, i.e. $[\boldsymbol{x}_h]_i = [\boldsymbol{x}]_{h^{-1}(i)}$. Via triangle inequality,

$$\|\nabla f(\boldsymbol{x})\| \leq \frac{3}{\|\boldsymbol{x}\|}\|\boldsymbol{x}\| \leq 3. \tag{182}$$

Thus, by Gaussian Lipschitz concentration (Lemma 2.2 of (Pisier, 1986)), we have that:

$$\Pr_{\boldsymbol{x}}\left[\left|\frac{\langle g\boldsymbol{x}, g'\boldsymbol{x}\rangle}{\|\boldsymbol{x}\|} - \mathbb{E}f(\boldsymbol{x})\right| \geq \Omega(n^c)\right] \leq \exp(-\Omega(n^{2c})). \tag{183}$$

Finally, we compute $\mu := \mathbb{E}_{\boldsymbol{x}}f(\boldsymbol{x})$. To do this, note that for every $i \neq j \in [n]$,

$$\mathbb{E}_{\boldsymbol{x}}\frac{\boldsymbol{x}_i\boldsymbol{x}_j}{\|\boldsymbol{x}\|_2} = \mathbb{E}_{\boldsymbol{x}}\frac{(-\boldsymbol{x}_i)\boldsymbol{x}_j}{\|\boldsymbol{x}\|_2} = 0, \tag{184}$$

by a change of variable formula $\boldsymbol{x} \mapsto (\boldsymbol{x}_1, \ldots, \boldsymbol{x}_{i-1}, -\boldsymbol{x}_i, \boldsymbol{x}_{i+1}, \ldots, \boldsymbol{x}_n)$. $\qquad\square$

At the same time, for every $i \neq j \in [n]$,

$$\mathbb{E}_{\boldsymbol{x}}\frac{\boldsymbol{x}_i^2}{\|\boldsymbol{x}\|_2} = \mathbb{E}_{\boldsymbol{x}}\frac{\boldsymbol{x}_j^2}{\|\boldsymbol{x}\|_2} = \frac{1}{n}\sum_{k=1}^n \mathbb{E}_{\boldsymbol{x}}\frac{\boldsymbol{x}_k^2}{\|\boldsymbol{x}\|_2} = \frac{1}{n}\mathbb{E}_{\boldsymbol{x}}\|\boldsymbol{x}\|_2, \tag{185}$$

by a change of variable that swaps $\boldsymbol{x}_i$ and $\boldsymbol{x}_j$ using a permutation Jacobian matrix with determinant $\pm 1$.

Thus $\mu = \frac{F_{g,g'}}{n}\mathbb{E}_{\boldsymbol{x}}\|\boldsymbol{x}\|_2$, where $F_{g,g'}$ is the number of fixed points of $g^{-1}g'$. We appeal to elementary analysis of Gaussian vectors (for instance, Chandrasekaran et al. (2012)) to obtain:

$$\mathbb{E}_{\boldsymbol{x}}\|\boldsymbol{x}\|_2 = \sqrt{2}\frac{\Gamma((n+1)/2)}{\Gamma(n/2)} \in [n/\sqrt{n+1}, \sqrt{n}], \tag{186}$$

where $\Gamma$ is Euler's Gamma function. Thus, we have

$$\mu \in \left[\frac{F_{g,g'}}{\sqrt{n+1}}, \frac{F_{g,g'}}{\sqrt{n}}\right] \tag{187}$$

**Lemma 41.** *Let $\boldsymbol{x} \in \mathbb{R}^n$ be a unit vector iid uniformly distributed over the $n$-dimensional unit sphere. Let $G \leq S_n$ and $g \neq g' \in G$. Let $F_{g,g}$ be the number of fixed points of $h := g^{-1}g'$'s action on $[n]$. Then for all $c \in (0, 1/2)$ and small $\epsilon \in (0, 1)$:*

$$\Pr_{\boldsymbol{x}}\left[\langle g\boldsymbol{x}, g'\boldsymbol{x}\rangle - \frac{F_{g,g'}}{n\sqrt{1-\epsilon}} \geq \Omega(n^{c-1/2})\right] \leq \exp(-\Omega(n^{2c})), \tag{188}$$

$$\Pr_{\boldsymbol{x}}\left[\langle g\boldsymbol{x}, g'\boldsymbol{x}\rangle - \sqrt{1-\epsilon}\frac{F_{g,g'}}{n} \leq -\Omega(n^{c-1/2})\right] \leq \exp(-\Omega(n^{2c})). \tag{189}$$

*Proof.* Recall that an alternative way to sample unit vectors i.i.d. uniformly from the hypersphere is to first sample $z \sim \mathcal{N}(0, I_n)$ and then set $x := \frac{z}{\|z\|_2}$. Thus, we study $\frac{\langle gz, g'z \rangle}{\|z\|_2^2}$.

To do this, note that concentration of $\|z\|_2$ is well studied (Proposition 2.2 and Corollary 2.3 of Barvinok (2005)), for any $0 < \epsilon < 1$:

$$\max \left( \Pr_z \left( \|z\|_2 \geq \sqrt{\frac{n}{1-\epsilon}} \right), \Pr_z \left( \|z\|_2 \leq \sqrt{(1-\epsilon)n} \right) \right) \leq \exp(-\epsilon^2 n/4). \tag{190}$$

Concentration of the random variable $\langle gz, g'z \rangle / \|z\|_2$ was studied in Lemma 40 as:

$$\Pr_{z \sim \mathcal{N}^n} \left[ |\langle gz, g'z \rangle / \|z\|_2 - \mu_{g,g'}| \geq \Omega(n^{c+1/2}) \right] \leq \exp(-\Omega(n^{2c})), \tag{191}$$

for some $\mu_{g,g'} \in [F_{g,g'}/\sqrt{n+1}, F_{g,g'}/\sqrt{n}]$. Since $\frac{1}{\sqrt{n}} - \frac{1}{\sqrt{n+1}} \in O(n^{-3/2})$ and $F_{g,g'} \leq n$, we can also write:

$$\Pr_x \left[ \frac{\langle gx, g'x \rangle}{\|x\|} - \frac{F_{g,g'}}{\sqrt{n}} \geq \Omega(n^c) \right] \leq \exp(-\Omega(n^{2c})), \tag{192}$$

and

$$\Pr_x \left[ \frac{\langle gx, g'x \rangle}{\|x\|} - \frac{F_{g,g'}}{\sqrt{n}} \leq -\Omega(n^c) \right] \leq \exp(-\Omega(n^{2c})). \tag{193}$$

We have by union bound:

$$\Pr \left[ \frac{\langle gz, g'z \rangle}{\|z\|_2^2} - \frac{F_{g,g'}}{n\sqrt{1-\epsilon}} > \Omega(n^{c-1/2}) \right] \leq \Pr \left[ \frac{\langle gz, g'z \rangle}{\|z\|_2} > \frac{F_{g,g'}}{\sqrt{n}} + \Omega(n^c) \right] + \Pr[\|z\|_2 < \sqrt{(1-\epsilon)n}]$$
$$\tag{194}$$

$$\leq \exp(-\Omega(n^{2c})) + \exp(-\epsilon^2 n/4) \tag{195}$$

$$\leq \exp(-\Omega(n^{2c})). \tag{196}$$

On the other side,

$$\Pr \left[ \frac{\langle gz, g'z \rangle}{\|z\|_2^2} - \sqrt{1-\epsilon} \frac{F_{g,g'}}{n} < -\Omega(n^{c-1/2}) \right] \leq \Pr \left[ \frac{\langle gz, g'z \rangle}{\|z\|_2} < \frac{F_{g,g'}}{\sqrt{n}} - \Omega(n^c) \right] + \Pr \left[ \|z\|_2 > \sqrt{\frac{n}{1-\epsilon}} \right]$$
$$\tag{197}$$

$$\leq \exp(-\Omega(n^{2c})) + \exp(-\epsilon^2 n/4) \tag{198}$$

$$\leq \exp(-\Omega(n^{2c})). \tag{199}$$

$\square$

**Lemma 42** (Kronecker product of PSD matrices is PSD). *For some $m, n \in \mathbb{N}$. Fix $A \in \mathbb{R}^{m \times m}$ and $B \in \mathbb{R}^{n \times n}$ not necessarily symmetric. If for all $x \in \mathbb{R}^m, x^\top A x \geq 0$ and for all $y \in \mathbb{R}^n, y^\top B y \geq 0$ then for all $z \in \mathbb{R}^{mn}, z^\top (A \otimes B) z \geq 0$, where $\otimes$ is the Kronecker product.*

*Proof.* Write $z = (z^{[1]}, \ldots, z^{[m]})$ where $z^{[i]} \in \mathbb{R}^n$ for each $i \in [m]$, then we have:

$$z^\top (A \otimes B) z = \sum_{i,i'=1}^m \sum_{j,j'=1}^n z_j^{[i]} (A_{i,i'} B_{j,j'}) z_{j'}^{[i']} \tag{200}$$

$$= \sum_{i,i'=1}^m A_{i,i'} \sum_{j,j'=1}^n z_j^{[i]} B_{j,j'} z_{j'}^{[i']} \tag{201}$$

$$= \sum_{i,i'=1}^m A_{i,i'} C_{i,i'}, \tag{202}$$

where $C_{i,i'} = \sum_{j,j'=1}^n z_j^{[i]} B_{j,j'} z_{j'}^{[i']}$. Now it is not hard to see that $C$ is symmetric. At the same time, for any $w \in \mathbb{R}^m$,

$$w^\top C w = \sum_{i,i'=1}^n w_i w_{i'} \sum_{j,j'=1}^n z_j^{[i]} B_{j,j'} z_{j'}^{[i']} = \sum_{i,i'=1}^n \sum_{j,j'=1}^n (w_i z_j^{[i]}) B_{j,j'} (w_{i'} z_{j'}^{[i']}). \tag{203}$$

By property of $\boldsymbol{B}$, the inner sum is nonnegative and thus $\boldsymbol{w}^\top \boldsymbol{C} \boldsymbol{w}$ is nonnegative. Therefore we conclude that $\boldsymbol{C}$ is symmetric positive semi-definite and thus admits a spectral decomposition:

$$\boldsymbol{C} = \sum_{k=1}^{r} \lambda_k \boldsymbol{v}^{[k]} (\boldsymbol{v}^{[k]})^\top, \tag{204}$$

for some set of vectors $\boldsymbol{v}^{[k]}$ and nonnegative $\lambda_k \geq 0$, $k \in [r]$ for some $r$.

Thus,

$$\sum_{i,i'=1}^{m} \boldsymbol{A}_{i,i'} \boldsymbol{C}_{i,i'} = \sum_{i,i'=1}^{m} \boldsymbol{A}_{i,i'} \sum_{k=1}^{r} \lambda_k \boldsymbol{v}_i^{[k]} \boldsymbol{v}_{i'}^{[k]} = \sum_{i,i'=1}^{m} \sum_{k=1}^{r} \lambda_k \boldsymbol{v}_i^{[k]} \boldsymbol{A}_{i,i'} \boldsymbol{v}_{i'}^{[k]}. \tag{205}$$

By property of $\boldsymbol{A}$ and the fact that $\lambda_k$ are nonnegative, the inner sum is nonnegative and we get our conclusion. $\qquad\square$

**Corollary 43** (PSD of tensor power). *For any $m, n \in \mathbb{N}$, if $\boldsymbol{A} \in \mathbb{R}^{m \times m}$ is such that $\boldsymbol{x}^\top \boldsymbol{A} \boldsymbol{x} > 0$ for all $\boldsymbol{x} \in \mathbb{R}^m$, then $\boldsymbol{A}^{\otimes n}$ satisfies: $\boldsymbol{z}^\top \boldsymbol{A}^{\otimes n} \boldsymbol{z} > 0$ for all $\boldsymbol{z} \in \mathbb{R}^{m^n}$*

*Proof.* Apply Lemma 42 $n-1$ times. $\qquad\square$

*Proof of Lemma 29.* We use Corollary D.3 from (Diakonikolas et al., 2017) copied below:

**Lemma 44** (Corollary D.3 from (Diakonikolas et al., 2017)). *Let $\boldsymbol{x}, \boldsymbol{y} \in \mathbb{R}^n$ be two unit vectors iid uniformly distributed over the $n$-dimensional unit sphere. Then:*

$$\Pr_{\boldsymbol{x},\boldsymbol{y}}[|\langle \boldsymbol{x}, \boldsymbol{y} \rangle| \geq \Omega(n^{c-1/2})] \leq \exp(-\Omega(n^{2c})), \tag{206}$$

*for any $c \in (0, 1/2)$.*

Let $g, g' \in F$ be two arbitrary permutations that act on $\mathbb{R}^n$ by permuting the vector. Since the uniform distribution over the $n$-dimensional unit sphere is invariant to permutation of the indices, its pushforward via $g$ and $g'$ is still uniform. Therefore, we have, for any $g, g' \in F$:

$$\Pr_{\boldsymbol{x},\boldsymbol{y}}[|\langle g\boldsymbol{x}, g'\boldsymbol{y} \rangle| \geq \Omega(n^{c-1/2})] \leq \exp(-\Omega(n^{2c})). \tag{207}$$

Draw $T = 2^{\Omega(n^c)}/|F|^2$ such iid vectors to form a set $S \subset \mathbb{R}^n$. We have, by union bound:

$$\Pr_{\boldsymbol{x}_1,\ldots,\boldsymbol{x}_T}[\exists i \neq j \in [T], \exists g, g' \in F, |\langle g\boldsymbol{x}_i, g'\boldsymbol{x}_j \rangle| \geq \Omega(n^{c-1/2})] \tag{208}$$

$$\leq \sum_{i,j \in [T]} \sum_{g,g' \in F} \Pr_{\boldsymbol{x}_i,\boldsymbol{x}_j}[|\langle g\boldsymbol{x}_i, g'\boldsymbol{x}_j \rangle| \geq \Omega(n^{c-1/2})] \tag{209}$$

$$\leq 2^{\Omega(n^c)} \exp(-\Omega(n^{2c})) \tag{210}$$

$$\leq \exp(-\Omega(n^{2c})). \tag{211}$$

On the other hand, let $F_{g,g'}$ be the number of fixed points of $g^{-1}g'$ for some $g \neq g' \in F$, we also want to union bound using Lemma 41, for any small constant $\epsilon \in (0, 1)$:

$$\Pr_{\boldsymbol{x}_1,\ldots,\boldsymbol{x}_T}\left[\exists i \in [T], \exists g \neq g' \in F, \langle g\boldsymbol{x}_i, g'\boldsymbol{x}_i \rangle - \frac{F_{g,g'}}{n\sqrt{1-\epsilon}} \geq \Omega(n^{c-1/2})\right] \tag{212}$$

$$\leq \sum_{i \in [T]} \sum_{g \neq g' \in F} \Pr_{\boldsymbol{x}_i}\left[\langle g\boldsymbol{x}_i, g'\boldsymbol{x}_i \rangle - \frac{F_{g,g'}}{n\sqrt{1-\epsilon}} \geq \Omega(n^{c-1/2})\right] \geq \Omega(n^{c-1/2})] \tag{213}$$

$$\leq 3 \cdot 2^{\Omega(n^c)} \exp(-\Omega(n^{2c})) \tag{214}$$

$$\leq \exp(-\Omega(n^{2c})). \tag{215}$$

Similarly,

$$\Pr_{\boldsymbol{x}_1,\ldots,\boldsymbol{x}_T}\left[\exists i \in [T], \exists g \neq g' \in F, \langle g\boldsymbol{x}_i, g'\boldsymbol{x}_i\rangle - \sqrt{1-\epsilon}\frac{F_{g,g'}}{n} \leq -\Omega(n^{c-1/2})\right] \leq \exp(-\Omega(n^{2c})) \tag{216}$$

Therefore, for every small $\epsilon \in (0,1)$, there exists a set $S \subset \mathbb{R}^n$ of size $T = 2^{\Omega(n^c)}/|F|^2$ such that for every $\boldsymbol{x} \neq \boldsymbol{y} \in S$, for every $g \neq g' \in F$, we have:

$$\|g\boldsymbol{x}\|_2 = 1 \tag{217}$$

$$\max(|\langle g\boldsymbol{x}, g\boldsymbol{y}\rangle|, |\langle g\boldsymbol{x}, g'\boldsymbol{y}\rangle|) \leq O(n^{c-1/2}), \tag{218}$$

$$\langle g\boldsymbol{x}_i, g'\boldsymbol{x}_i\rangle - \frac{F_{g,g'}}{n\sqrt{1-\epsilon}} \leq O(n^{c-1/2}), \tag{219}$$

$$\langle g\boldsymbol{x}_i, g'\boldsymbol{x}_i\rangle - \sqrt{1-\epsilon}\frac{F_{g,g'}}{n} \geq -O(n^{c-1/2}). \tag{220}$$

Pair elements of $S$ (as column vectors) together arbitrarily to obtain a set $\mathcal{B}'$ of matrices in $\mathbb{R}^{n\times 2}$ of size $T/2$. Each element of $\mathcal{B}'$ has two columns, each one of these columns is a underline{distinct} vector in $S$. We have, for any $\boldsymbol{B} =: (\boldsymbol{x}_{1,1}, \boldsymbol{x}_{1,2}), \boldsymbol{B}' =: (\boldsymbol{x}_{2,1}, \boldsymbol{x}_{2,2}) \in \mathcal{B}'$, and for any $g, g' \in F$,

$$\|(g\boldsymbol{B})^\top(g'\boldsymbol{B}')\|_2 \leq \|(g\boldsymbol{B})(g'\boldsymbol{B}')^\top\|_F = \sqrt{\sum_{i,j\in[2]}(\langle g\boldsymbol{x}_{1i}, g'\boldsymbol{x}_{2j}\rangle)^2} \leq O(n^{c-1/2}). \tag{221}$$

The remaining conditions in Lemma 29 are also straightforward to check. $\square$

## G  LEARNING INVARIANT POLYNOMIALS

Here, we aim to extend the algorithm of Andoni et al. (2014) to the setting of invariant polynomials. This algorithm is an SQ algorithm, but not a CSQ algorithm.

First, let us review the GROWING-BASIS algorithm. Assume inputs $\boldsymbol{x} \in \mathbb{R}^n$ are distributed according to some product distribution $\mathcal{D} = \mu_1 \times \cdots \times \mu_n$. The goal of the algorithm is to learn the coefficients of some polynomial $f^*(\boldsymbol{x}) = \sum_S a_S H_S(\boldsymbol{x})$ where $S = (S_1, S_2, \ldots, S_n)$ is a tuple of positive integers which indicate the degree of the polynomial in the $i$-th variable. The monomials in the polynomial take the form

$$f^*(\boldsymbol{x}) = \sum_S a_S H_S(\boldsymbol{x}), \text{ where } H_S(\boldsymbol{x}) = \prod_{i=1}^n H_{S_i}(\boldsymbol{x}_i), \tag{222}$$

where $H_{S_i}$ is the $S_i$-th degree polynomial in an class of polynomials orthogonal under the distribution $\mathcal{D}$: $\langle H_i, H_j\rangle = \int H_i(x)H_j(x)\mu_i(x)dx = 0$ for all $i \neq j$. An example of such an orthogonal set of polynomials are the Legendre polynomials which are orthogonal under the uniform distribution over $[-1,1]$.

Next, to rigorously obtain a separation on invariant function classes, we need to define a few objects. We will often simplify notation or definitions for ease of readability; the presentation of this material in its full mathematical rigor can be found in various textbooks, e.g. Derksen & Kemper (2015).

**Independent generators**  Let $g_1, \ldots, g_r$ be invariant functions and K a field, either $\mathbb{R}$ or $\mathbb{C}$. The ring of polynomial $\mathrm{K}[g_1, \ldots, g_r]$ is the set of all sums $\sum_{\alpha\in\mathbb{N}^r} c_\alpha g^\alpha$ where $g^\alpha = \prod_{i=1}^r g_i^{\alpha_i}$. Assume that $\{g_i\}_{i=1}^r$ are independent in the sense that for all $i \in [r]$, there is no polynomial $P \in \mathrm{K}[g_1, \ldots, g_{i-1}, g_{i+1}, \ldots, g_r]$ that agrees with $g_i$ as a function $\mathrm{K}^n \to \mathrm{K}$. It is not hard to see that if a member of $\mathrm{K}[g_1, \ldots, g_r]$ can be evaluated as a function $\mathrm{K}^n \to \mathrm{K}$ then they are also invariant functions. The converse is not true, in general, as some invariant functions may not be written as polynomials over certain generators. However, some special groups have nice, finitely many generators whose polynomial ring spans almost the whole space of invariant functions. Denote by $\mathcal{F}^{d,r} := \mathrm{K}[g_1, \ldots, g_r]^{\leq d}$ the set of all polynomials in $g_i$'s with degree at most $d$, i.e., $\sum_{\alpha\in\mathbb{N}^r | \sum_i \alpha_i \leq d} c_\alpha g^\alpha$.

**Vector space of bounded degree polynomials generated by a finite set**   A different way of thinking about $\mathcal{F}^{d,r}$ is as a vector space. Let $V^d(g_1, \ldots, g_r)$ be the $(r+1)^d$-dimensional vector space over the field $k$, whose canonical basis is the set $\{e_I\}_{I \in (\{0\} \cup [r])^d}$. It is not hard to show that $V^d(g_1, \ldots, g_r)$ is isomorphic to $\mathrm{K}[g_1, \ldots, g_r]^{\leq d}$ by identifying $e_I$ to the monomial $\prod_{i=1}^d g_{I_i}$ where $g_0 \equiv 1$. This perspective as a vector space allows for a nice calculus tool: let $\mathbb{P}$ be a product distribution over $\mathrm{K}^r$ (tuples of length $r$, with each entry an element in K) and $\langle f, g \rangle_{\mathbb{P}} := \int f\bar{g}\mathrm{d}\mathbb{P}$. Here, we once again think of $g_i$'s as functions $\mathrm{K}^n \to \mathrm{K}$ and so are elements in $\mathcal{F}^{d,r}$. Applying Gram-Schmidt orthogonalization to basis elements of $V^d(g_1, \ldots, g_r)$ generates another set of orthogonal (in $\langle \cdot, \cdot \rangle_{\mathbb{P}}$) elements $H_1, \ldots H_{(r+1)^d}$.[5] Note that there are still $(r+1)^d$ such orthogonal functions since Gram-Schmidt preserves dimensionality of the input and ouput vector spaces. This process in analogous to that constructing the Hermite and Legendre polynomials for Gaussian in uniform distributions respectively. When $\mathbb{P}$ is a product distribution, we can reindex this basis as $\{H_{\boldsymbol{v}}\}_{\boldsymbol{v} \in (\{0\} \cup [r])^d}$.

**Sparse polynomial**   Before giving any formal results, we must state some technical preliminary restrictions that must be set for learning to be possible. Our learning setting restricts $\mathcal{F}^{d,r}$ to only contain polynomials in $g_1, \ldots, g_r$ with at most $k$ terms in the expansion using $\{H_{\boldsymbol{v}}\}_{\boldsymbol{v} \in (\{0\} \cup [r])^d}$. In other words, when viewed as vectors in $V^d(g_1, \ldots, g_r)$ under the basis $\{H_{\boldsymbol{v}}\}_{\boldsymbol{v} \in (\{0\} \cup [r])^d}$, only $k$ elements are non-zero. Call this new set $\mathcal{F}^{d,r,k}$.

We should also note that for technical reasons, we restrict the outputs of the polynomials to be normalized within some bounded range to properly normalize the function with respect to the error metric and allow for the SQ query bounds to also properly query such functions. In the GROWING-BASIS algorithm, one must make queries to estimate $\langle H_{\boldsymbol{v}}, f^* \rangle$ and $\langle H_{\boldsymbol{v}}, (f^*)^2 \rangle$. Given any statistical query function has output bounded in the range $[-1, +1]$, to measure the correlations, we must choose a query function that divides by the maximum value of $|H_{\boldsymbol{v}}(f^*)^2|$ in some high probability region $D \subseteq \mathrm{K}^r$ with measure $\mathbb{P}(D) \geq 1 - o(1)$ and outputs zero elsewhere. This way, statistical query functions whose outputs are bounded in the range $[-1, +1]$ can effectively calculate the proper correlations needed in the algorithm.[6] Thus, we introduce a normalization term $\hat{M}_{d,k} = \max_{\boldsymbol{x} \in D, \boldsymbol{v} \in (\{0\} \cup [r])^d} |H_{\boldsymbol{v}}(f^*)^2|$ and introduce this quantity into the query tolerance to allow for proper estimation of relevant variables. This normalization plays a similar role to the quantity $M_{d,k}$ in Andoni et al. (2014) for their guarantees of sampling complexity. When the target function $f^*$ is properly bounded, it is not too hard to see that this quantity is also easily bounded. For example, when $|f^*| = O(1)$ in the range $[-1, +1]^n$, $\hat{M}_{d,k}$ is at most a constant for inputs drawn from the uniform distribution over the interval $[-1, +1]$ since the Legendre polynomials are also appropriately bounded. In general, functions can always be normalized by dividing by an appropriate constant.

Finally, the GROWING-BASIS algorithm depends on a parameter $\tau_d$ that controls how much higher order terms show up in the square of a polynomial. More formally, let $H_i : \mathrm{K} \to \mathrm{K}$ denote the $i$-th orthogonal polynomial for some given distribution on K. For any such orthogonal polynomial, we have the decomposition (see equation 2.1 of Andoni et al. (2014))

$$H_t(x) = 1 + \sum_{j=1}^{2t} c_{t,j} H_j(x). \tag{223}$$

Let $c_t = c_{t,2t}$, then $\tau_d = \min_{t \leq d} c_t$. This quantity is generally bounded for common distributions, e.g., when $H_t$ are the Hermite or Legendre polynomials then $\tau_d = \Omega(1)$ (Andoni et al., 2014).

With this set up, we are ready to state and prove our main result for this section:

**Theorem 45** (Adapted from theorems 2.1 and 2.2 of Andoni et al. (2014))**.** *Let* K *be a field,* $g_1, \ldots, g_r$ *be independent functions* $\mathrm{K}^n \to \mathrm{K}$ *that are invariant to a group* $G$ *acting alge-*

---

[5]We also note that it is technically more efficient to apply this procedure to each of the generators individually. That is, apply Gram-Schmidt to the space $[1, g_i, \ldots, g_i^d]$. Since there is a product distribution, this procedure constructs orthogonal polynomials in each basis which can then be combined to construct orthogonal polynomials in $\mathrm{K}[g_1, \ldots, g_r]$.

[6]We note that there are likely more effective means of handling this normalization requirement, by e.g., composing multiple queries together to calculate the relevant quantities. However, we do not consider this strengthening here for sake of simplicity.

*braically on* $\mathrm{K}[g_1, \ldots, g_r]$ *as defined above. Let* $\mathbb{P}$ *be a product distribution and define the inner product* $\langle p, q \rangle_{\mathbb{P}} := \mathbb{E}_{\boldsymbol{g}:=(g_1, \ldots, g_r) \sim \mathbb{P}}[p(\boldsymbol{g})\overline{q(\boldsymbol{g})}]$. *Let* $\{H_{\boldsymbol{v}}\}_{\boldsymbol{v} \in (\{0\} \cup [r])^d}$ *be the output of Gram-Schmidt on* $\mathcal{F}^{d,r}$ *with respect to this inner product. Further assume that the polynomial quotient* $H_{2m,0,\ldots,0}/(H_{m,0,\ldots,0})^2$ *(ignoring remainders) is* $\tau_d$ *for every* $m \in [\lfloor d/2 \rfloor]$. *Let* $\mathcal{F}^{d,r,k}$ *be the space of polynomials in* $\mathcal{F}^{d,r}$ *that is* $k$ *sparse in* $H_i$ *basis. Then any CSQ algorithm with oracle access to* $\langle \cdot, f^* \rangle_{\mathbb{P}}$ *for some* $f^* \in \mathcal{F}^{d,r,k}$ *that outputs a hypothesis* $h \in \mathcal{F}^{d,r,k}$ *while achieving mean square error* $\|h - f^*\|_{\mathbb{P}} < O(1)$ *must make either at least* $(r+1)^d$ *calls to the oracle or require precision less than* $O(r^{-c})$ *for any* $c > 0$. *However, GROWING-BASIS can learn* $h$ *with error* $\|h - f^*\|_{\mathbb{P}} < O(1)$ *using only* $O(krd(1/\tau_d)^d)$ *calls to oracles* $\langle \cdot, f^* \rangle_{\mathbb{P}}$ *and* $\langle \cdot, (f^*)^2 \rangle_{\mathbb{P}}$ *with precision* $\Omega(\tau_d^d/(krd\hat{M}_{d,k}))$.

*Proof.* The CSQ lower bound can be obtained by invoking standard correlational statistical queries dimension bound form orthogonal polynomials (Diakonikolas et al., 2020; Goel et al., 2020; Reyzin, 2020; Szörényi, 2009). Specifically, by correctness of Gram-Schmidt orthonormalization, the set of basis elements $\{H_{\boldsymbol{v}}\}_{\boldsymbol{v} \in (\{0\} \cup [r])^d}$ also form a pairwise orthogonal set, which each themselves have norm 1 under $\langle p, q \rangle_{\mathbb{P}}$. thus CSQ dimension of $f^* \in \mathcal{F}^{d,r,k}$ is immediately lower bounded by $\Omega(r^d)$ and the conclusion for CSQ algorithms follow.

The SQ upper bound via GROWING-BASIS can be obtained by observing that Andoni et al. (2014)'s proof still works verbatim when the main ingredients are there: a product distribution, an inner product with an orthornormal basis, and access to oracle $\langle \cdot, (f^*)^2 \rangle_{\mathbb{P}}$. Note that although the GROWING-BASIS algorithm is presented error free, we allow small errors, by making a thresholding in Step 2 and 8 of Andoni et al. (2014) equal to the error tolerance $\epsilon$. This way, we are guaranteed to miss only terms with low correlation and thus bounded effect on the final mean square error. To be more precise, every time we made an error on the cutoff of step 8, we lose as most a fraction of the precision that amounts to $\tau_d^d$ where as stated before $\tau_d$ is the coefficient of $H_{2n,0,\ldots,0}$ in $(H_{n,0,\ldots,0})^2$. Intuitively, when losing a monomial in the final product, the norm of the monomial can multiplicatively compound over each variable. $\tau_d$ is the lowest possible amount each variable can contribute to this loss (See (Andoni et al., 2014), Lemma 2.3). Finally, we also must divide the query tolerance by a factor $\hat{M}_{d,k}$ to account for the normalization of the target function as discussed earlier. □

**Corollary 46.** *In the same setting as Theorem 45, if instead, we are interested in polynomials that are* $k$-*sparse in the canonical basis* $\{\prod_{i=1}^r g_i^{\alpha_i} \mid \sum_i \alpha_i = 1\}$ *(recall that* $\mathcal{F}^{d,r,k}$ *was sparse in the orthogonal basis* $\{H_{\boldsymbol{v}}\}_{\boldsymbol{v} \in (\{0\} \cup [r])^d}$*). Then the query complexity of GROWING-BASIS becomes at most* $O(krd2^d)$ *with precision* $\Omega(1/(krd2^d))$ *whenever* $\tau_d = \Omega(1)$ *and targets are normalized so that* $\hat{M}_{d,k} = O(1)$.

*Proof.* By the same argument as Lemma 2.1 of (Andoni et al., 2014), any change of basis in the vector space of polynomials of bounded degree incur at worst a multiplicative factor $2^d$ in the sparsity parameter. Since the new function class is $k$-sparse in the canonical basis, it is $k2^d$-sparse in the orthogonal basis. Apply Theorem 45 to this new sparsity level to get the desired result. □

This separation is weaker than when there is sparsity in the orthogonal basis. However, there is still a polynomial (in $r$) separation between CSQ and SQ if $d$ and $k$ are set to be constant independent of $r$ and a superpolynomial separation when $d = \Theta(\log(n))$.

Finally, for the above theorem to give a meaningful separation, we list some example groups for which there is a simple distribution $\mathcal{D}$ over inputs such that the push-forward distribution $\mathbb{P} = \phi_{\#}\mathcal{D}$ via the map $\phi : \boldsymbol{x} := (x_1, \ldots, x_n) \mapsto (g_1(\boldsymbol{x}), \ldots g_r(\boldsymbol{x}))$ is a product distribution.

**Example 5.** Consider inputs $\boldsymbol{X} \in \mathbb{R}^{n \times 3}$ which correspond to $n$ points in $\mathbb{R}^3$. Ordered point clouds are symmetric under the action $\boldsymbol{X} \cdot \boldsymbol{U}$ for matrix $\boldsymbol{U} \in O(3)$ in the orthogonal group. Let inputs $\boldsymbol{X}$ have entries drawn i.i.d. Gaussian. Consider generators $g_1, \ldots, g_n$ where $g_i(\boldsymbol{x}) = \|\boldsymbol{X}_{i,:}\|^2$, which is the sum of the squares of the $i$-th row of $\boldsymbol{X}$ and clearly invariant to orthogonal transformations. The pushforward distribution after the mapping $\boldsymbol{X} \mapsto [g_1(\boldsymbol{X}), \ldots, g_n(\boldsymbol{X})]$ is a product distribution over Chi-squared random variables $\chi^2(3)^n$.

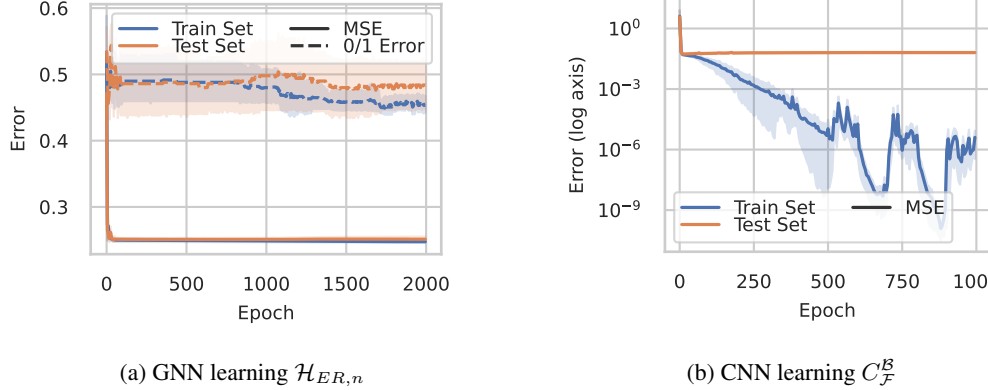

(a) GNN learning $\mathcal{H}_{ER,n}$     (b) CNN learning $C_{\mathcal{F}}^{\mathcal{B}}$

Figure 3: Replication of experiments as in Fig. 1, except here, we consider a minimal architecture consisting of a single layer of graph or cyclic convolution followed by a single hidden layer MLP. This is the minimal number of layers needed to learn the desired function classes for the architectures considered. For the CNN plot, the jumps in the train set MSE are due to perturbations in the loss at very low values near computer precision.

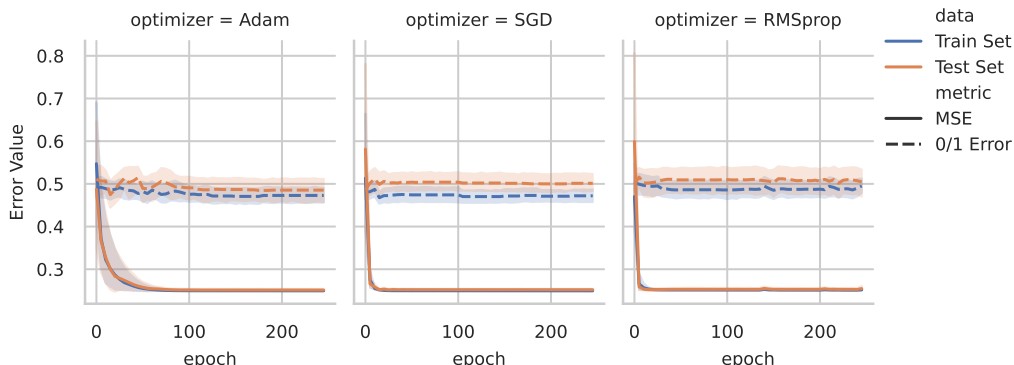

Figure 4: Replication of experiments in Fig. 1a with different optimizers show that the performance of the GNN is virtually the same across the various optimizers. Performance is averaged over 10 runs. For each run, the learning rate is chosen by perturbing the default learning rate by a random multiplicative factor in the range $[0.1, 10]$.

**Example 6.** Consider inputs $x \in \mathbb{C}^n$ drawn from $U[\mathbb{C} \cap \{|z| = 1\}]^n$ uniformly from unit norm complex numbers in each entry. The power sum symmetric polynomials consist of a generating basis $\left[\sum_i x_i, ..., \sum_i x_i^d\right]$. Another way to parameterize each $x_j$ is as $x_j = e^{i\theta}$ where $\theta$ is distributed $U[0, 2\pi]$, for all $j \in [n]$. Thus $x_j^t$ can also be parameterized by $e^{i\theta}$ with the exact same distribution of $\theta \sim U[0, 2\pi]$, for all $j \in [n], t \in [d]$. The distribution of $\left[\sum_i x_i, ..., \sum_i x_i^d\right]$ is therefore a product distribution $(\text{Law}(\sum_{i=1}^n Z_i))^d$ where $Z_i \sim U[\mathbb{C} \cap \{|z| = 1\}]$. These generators and input distributions come from the classical text of Macdonald (1979). Here, as long as $r \le n$, $r$ only depends on $d$ and not $n$. Plugging this $r$ into Theorem 45 gives straightforward bounds that are independent of $n$-the original number of variables, but does not give meaningful separation in Corollary 46.

As a reminder, we also have an example in the main text for the sign group (Lim et al., 2022).

## H   EXPERIMENTAL DETAILS

GNN experiments were performed using the Pytorch Geometric library for implementing geometric architectures (Fey & Lenssen, 2019). The hard functions were drawn from $\mathcal{H}_{ER,n}$ (see Eq. (7))

where subset $S \subseteq [n]$ was drawn uniformly from all the possible subsets of size $\lfloor n/2 \rfloor$ and $b$ was either $0$ or $1$ with equal probability. For our experiments, we set $n = 15$ and trained on $n^2 = 225$ datapoints. The overparameterized GNN used during training consisted of 3 layers of graph convolution followed by a node aggregation average pooling layer and a two layer ReLU MLP with width 64. The graph convolution layers used 32 channels.

For the CNN experiments, we constructed the hard functions in Eq. (17) setting $n = 50$ and $k = 10$. Here, matrices $B \in \mathcal{B}$ were drawn randomly from all possible $n \times 2$ orthogonal matrices by taking a QR decomposition of a random $n \times 2$ matrix with Gaussian i.i.d. elements. For sake of convenience, we did not divide by the norm of the function. The CNN architecture implemented consisted of three layers of fully parameterized cyclic convolution each with 100 output channels. This was followed by a global average pooling layer and and a two layer ReLU MLP of width 100. The network was given $10n = 500$ training samples and was trained with the Adam optimizer with batch size 32.

Experiments in the main text considered architectures which were overparameterized in terms of the number of layers with respect to the target function. For sake of completeness, we include in Fig. 3 results for merely sufficiently parameterized networks which consist of a single convolution layer, pooling and single hidden layer MLP. The results in Fig. 3 are consistent with those observed in Fig. 1. Additionally, to further confirm that these results are robust to hyperparameter choices for the optimizer, we repeated the GNN learning experiments with the choice of optimizer varying between Adam, SGD, and RMSprop (Paszke et al., 2019). Fig. 4 shows the results for these different optimizers. For each optimizer, 10 simulations were performed where for each simulation, the learning rate was chosen by multiplying the default learning rate by a random number in the range $[0.1, 10]$.

All experiments were run using Pytorch on a single GPU (Paszke et al., 2019). Default initialization schemes were used for the initial network weights. In our experiments, we used the Adam optimizer and tuned the learning rate in the range $[0.0001, 0.003]$. For CNN experiments, to increase stability of training in later stages, we added a scheduler that divided the learning rate by two every 200 epochs. Plots are created by combining and averaging five random realizations of each experiment.

