# On the hardness of learning under symmetries

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

}(X)|} \sum_{g \in \mathcal{F}(X)} h(g^{-1}X)$. For instance, setting $\forall X : \mathcal{F}(X) = G$ recovers the Reynolds (or group-averaging) operator. Given a fixed frame $\mathcal{F}$, we will first show CSQ lower bounds for the following class of frame-averaged one-hidden-layer fully connected nets:

$$\mathcal{H}_{\mathcal{F}} := \left\{ f : \mathbb{R}^{n \times d} \to \mathbb{R}, f(X) = \frac{1}{\sqrt{|\mathcal{F}(X)|}} \sum_{g \in \mathcal{F}(X)} a^{\top} \sigma(W^{\top}(g^{-1}X))\mathbf{1}_d \mid W \in \mathbb{R}^{n \times k}, a \in \mathbb{R}^k \right\},$$

for some nonlinearity $\sigma$.

**Example 2** (Frame for CNN). Set $d = 1$ and let $G$ be the cyclic group acting on $\mathbb{R}^n$ via cyclic shifts of its elements with frame $\mathcal{F}(X) = G, \forall X \in \mathbb{R}^n$. Then, $\mathcal{F}_n^d$ consists of CNNs with one convolutional layer and $k$ hidden channels.

**Remark 6** (Difficulty of frames). Since the frame $\mathcal{F}(X)$ may vary by datapoint $X$, the distribution $\text{Unif}(\mathcal{F}(X)^{-1}) \circ X$ can be significantly different from the original distribution over $X$, even if $X \sim gX$ for all $g$. For example, consider the frame $\mathcal{F}(X)$ containing all permutations that sort $X$ lexicographically (where $|\mathcal{F}(X)| > 1$ if $X$ contains a repeated row). Even if $X \sim \mathcal{N}$ with $\mathcal{N}$ invariant to row-permutation, the resultant distribution is not row permutation invariant.

We focus on certain cases where such effects are not too pronounced. For instance, it is simple to check that if $\mathcal{F}(X)$ is constant over $X$, then $\mathcal{F}(X) = G \; \forall X$ (Lemma 27). In the following, we assume that $G$ is a polynomial-sized (in $n$) subgroup of $S_n$.

**Family of hard functions.** Recall the low-dimensional function from the proof of Theorem 4:

$$f_{\exp}^* : \mathbb{R}^{2 \times d} \to \mathbb{R} \text{ with } f(X) = (a^*)^{\top} \sigma((W^*)^{\top}X)\mathbf{1}_d \tag{16}$$

for some special parameter $a^* \in \mathbb{R}^{2k}$ and $W^* \in \mathbb{R}^{2 \times 2k}$ in Eq. (11). We now define the family of hard functions, indexed by a set of matrices $\mathcal{B} \subset \mathbb{R}^{n \times 2}$ obtained from Lemma 28:

$$C_{\mathcal{F}}^{\mathcal{B}} = \left\{ g_B : \mathbb{R}^{n \times d} \to \mathbb{R} \text{ with } g_B(X) = \frac{\sum_{g \in G} f_{\exp}^*(B^{\top}g^{-1}X)}{\sqrt{|G| \cdot \|f_{\exp}^*\|_{\mathcal{N}}}} \mid B \in \mathcal{B} \subset \mathbb{R}^{n \times 2} \right\}. \tag{17}$$

**Theorem 7.** *For feature dimension $d$ independent of inputs $n$ and width parameter $k = \Theta(n)$, let $\epsilon > 0$ be a sufficiently small error constant. Then there exists a set $\mathcal{B}'$ of size at least $2^{\Omega(d^{\Omega(1)})}/|G|^2$ such that: for any target $g \in C_{\mathcal{F}}^{\mathcal{B}'}$, $\|g\|_{\mathcal{N}} = \Theta(1)$, any CSQ algorithm that outputs a hypothesis $h$ with $\|g - h\|_{\mathcal{N}} \leq \epsilon$ requires either $2^{n^{\Omega(1)}}/|G|^2$ queries or at least one query with precision $|G|2^{-n^{\Omega(1)}} + \sqrt{|G|}n^{-\Omega(k)}$.*

*Proof sketch.* Using a union bound argument over the original set $\mathcal{B}$ from Diakonikolas et al. (2020), we obtain from Lemma 28 a set of orthogonal matrices $\mathcal{B}'$ such that not only does $\|B(B')^T\|$ remain small $\forall B, B' \in \mathcal{B}'$, but also $\|gB(g'B')^T\|$ remains small $\forall g, g' \in G$. This construction costs a factor of $|G|^2$ in $|\mathcal{B}'|$. The rest of the proof proceeds similarly; see App. F.1. $\square$

We also obtain superpolynomial lower bounds for more general frames using the technique of Goel et al. (2020). Here, we drop the assumption that $G$ is polynomially-sized. The hard functions are based on parity functions, similar to Goel et al. (2020) (but with an extra input dimension for $X$):

$$f_S : \mathbb{R}^{n \times d} \to \mathbb{R} \text{ with } f_S(X) = \mathbf{1}_d^{\top} \Big( \sum_{w \in \{-1,1\}^m} \chi(w)\sigma(\langle w, X_S \rangle / \sqrt{m}) \Big), \

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

}[\widehat{\boldsymbol{c}}_{\boldsymbol{A}} = \widehat{\boldsymbol{c}}_{\boldsymbol{A}'}] &\leq \mathbb{P}\left[[\widehat{\boldsymbol{c}}_{\boldsymbol{A}}]_{r_1} = [\widehat{\boldsymbol{c}}_{\boldsymbol{A}'}]_{r_1}, \ldots, [\widehat{\boldsymbol{c}}_{\boldsymbol{A}}]_{r_N} = [\widehat{\boldsymbol{c}}_{\boldsymbol{A}'}]_{r_N}\right] \\
&= \mathbb{P}\left[[\widehat{\boldsymbol{c}}_{\boldsymbol{A}}]_{r_1} = [\widehat{\boldsymbol{c}}_{\boldsymbol{A}'}]_{r_1}\right] \mathbb{P}\left[[\widehat{\boldsymbol{c}}_{\boldsymbol{A}}]_{r_2} = [\widehat{\boldsymbol{c}}_{\boldsymbol{A}'}]_{r_2} \mid [\widehat{\boldsymbol{c}}_{\boldsymbol{A}}]_{r_1} = [\widehat{\boldsymbol{c}}_{\boldsymbol{A}'}]_{r_1}\right] \times \\
&\quad \times \cdots \mathbb{P}\left[[\widehat{\boldsymbol{c}}_{\boldsymbol{A}}]_{r_N} = [\widehat{\boldsymbol{c}}_{\boldsymbol{A}'}]_{r_N} \mid [\widehat{\boldsymbol{c}}_{\boldsymbol{A}}]_{r_1} = [\widehat{\boldsymbol{c}}_{\boldsymbol{A}'}]_{r_1}, \ldots, [\widehat{\boldsymbol{c}}_{\boldsymbol{A}}]_{r_{N-1}} = [\widehat{\boldsymbol{c}}_{\boldsymbol{A}'}]_{r_{N-1}}\right] \\
&\leq \left(\frac{1}{2} + O(n^{-1/4})\right)^{|\Delta_p|} \\
&= O(\exp(-\Omega(|\Delta_p|))) \\
&= O(\exp(-\Omega(n^p))).
\end{aligned}
\tag{39}
$$

Since $\mathbb{P}_{\boldsymbol{A} \sim \mathrm{Unif}(\{0,1\}^{n \times n})}[\widehat{\boldsymbol{c}}_{\boldsymbol{A}} \neq \widehat{\boldsymbol{c}}_{\boldsymbol{A}'}] = 1 - \mathbb{P}_{\boldsymbol{A} \sim \mathrm{Unif}(\{0,1\}^{n \times n})}[\widehat{\boldsymbol{c}}_{\boldsymbol{A}} = \widehat{\boldsymbol{c}}_{\boldsymbol{A}'}]$, this completes the proof. $\qquad\square$

Finally, we need to show that the functions $g_{S,b}$ can be constructed as ReLU networks. Before proceeding to give this construction, we should note that it is known that any Boolean function that can be computed in time $O(T(n))$ can also be expressed by a neural network of size $O(T(n^2))$ (Parberry, 1994; Shamir, 2018). Our construction will be specific to the GNN class we consider and show that the boolean functions in $g_{S,b}$ are similarly efficiently constructible.

**Lemma 15** ($g_{S,b}$ as GNN). *For a GNN $f(\cdot)$ of the form of Eq. (5) with hidden layer widths equal to $k_1 = O(n)$ and $k_2 = O(n)$, there exist weights $\boldsymbol{a}, \boldsymbol{b} \in \mathbb{R}^{k_1}, \boldsymbol{u}, \boldsymbol{v} \in \mathbb{R}^{k_2}, \boldsymbol{W} \in \mathbb{R}^{k_2 \times k_1}$ such that for any $S \subseteq [n+1]$, $b \in \{0,1\}$, $g_{S,b}(\boldsymbol{A}) = f(\boldsymbol{A}; \boldsymbol{a}, \boldsymbol{b}, \boldsymbol{u}, \boldsymbol{v}, \boldsymbol{W})$ for all inputs $\boldsymbol{

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

} = g'\boldsymbol{B}'$ iff $g = g'$ and $\boldsymbol{B} = \boldsymbol{B}'$; and $\|(g\boldsymbol{B})(g'\boldsymbol{B}')^\top\|_2 \leq O(d^{c-1/2})$.*

The proof is delayed to App. F.3 for a more streamlined read.

**Family of hard functions.** Recall the low-dimensional function from the proof of Theorem 4:

$$f^*_{\exp} : \mathbb{R}^{2 \times d} \to \mathbb{R} \text{ with } f(\boldsymbol{X}) = (\boldsymbol{a}^*)^\top \sigma((\boldsymbol{W}^*)^\top \boldsymbol{X}) \mathbf{1}_d \tag{112}$$

for some special parameter $\boldsymbol{a}^* \in \mathbb{R}^{2k}$ and $\boldsymbol{W}^* \in \mathbb{R}^{2 \times 2k}$ in Eq. (11). We now define the family of hard functions, indexed by a set of matrices $\mathcal{B} \subset \mathbb{R}^{n \times 2}$ obtained from Lemma 28:

$$C^{\mathcal{B}}_{\mathcal{F}} = \left\{ g_{\boldsymbol{B}} : \mathbb{R}^{n \times d} \to \mathbb{R} \text{ with } g_{\boldsymbol{B}}(\boldsymbol{X}) = \frac{\sum_{g \in G} f^*_{\exp}(\boldsymbol{B}^\top g^{-1} \boldsymbol{X})}{\sqrt{|G| \cdot \|f^*_{\exp}\|_{\mathcal{N}}}} \mid \boldsymbol{B} \in \mathcal{B}' \subset \mathbb{R}^{n \times 2} \right\}. \tag{113}$$

**Theorem 29.** *For any feature dimension $d = \Theta(1)$ and width parameter $k = \Theta(n)$, let $\epsilon > 0$ be a sufficiently small error constant and $\mathcal{F}$ an almost surely polynomial frame. Then there exists a set $\mathcal{B}$ of size at least $2^{\Omega(d^{\Omega(1)})}/|G|^2$ such that: for any $g \in C^{\mathcal{B}'}_{\mathcal{F}}$, $\|g\|_{\mathcal{N}} = \Theta(1)$; and any CSQ algorithm that queries from oracles of concept $f \in C^{\mathcal{B}'}_{\mathcal{F}}$ and outputs a hypothesis $h$ with $\|f - h\|_{\mathcal{N}} \leq \epsilon$ requires either $2^{n^{\Omega(1)}}/|G|^2$ queries or at least one query with precision $|G|2^{-n^{\Omega(1)}} + \sqrt{|G|}n^{-\Omega(k)}$.*

*Proof.* For any $\boldsymbol{B}, \boldsymbol{B}' \in \mathcal{B}'$ and any $g, g' \in F$ such that $g\boldsymbol{B} \neq g'\boldsymbol{B}'$, using Theorem 26 with matrix $\boldsymbol{A} = \boldsymbol{I}_d$ and note that $B^\top(g^{-1}X) = (gB)^\top X$ since a permutation matrix is orthonormal,

$$|\mathbb{E}_{\boldsymbol{X} \sim \mathcal{N}}[f^*_{\exp}((g\boldsymbol{B})^\top \boldsymbol{X}), f^*_{\exp}((g'\boldsymbol{B}')^\top \boldsymbol{X})]| \leq O(n^{k(c-1/2)})\|f^*_{\exp}\|^2_{\mathcal{N}}, \tag{114}$$

for the same $c$ that was chosen in the statement of Lemma 28.

First we verify that $\|g_{\boldsymbol{B}}\|_{\mathcal{N}} = \Theta(1)$ for each $g_{\boldsymbol{B}} \in C^{\mathcal{B}'}_{F A(G)}$:

$$\|g_{\boldsymbol{B}}\|^2_{\mathcal{N}} \tag{115}$$

$$= \frac{1}{\|f^*_{\exp}\|^2_{\mathcal{N}}} \mathbb{E}_{\boldsymbol{X} \sim \mathcal{N}} \left[ \sum_{g \in G} \frac{(f^*_{\exp}((g\boldsymbol{B})^\top \boldsymbol{X}))^2}{|G|} + \sum_{g \neq g' \in G} \frac{f^*_{\exp}((g\boldsymbol{B})^\top \rho(g) \boldsymbol{X}) f^*_{\exp}((g'\boldsymbol{B})^\top \boldsymbol{X})}{|G|} \right] \tag{116}$$

$$\geq 1 - O(n^{k(c-1/2)})(|G| - 1). \tag{117}$$

Since $k$ is of order $n$, the second term is eventually bounded by a constant.

Then, we also have:

$$|\langle g_{\boldsymbol{B}}, g_{\boldsymbol{B}'} \rangle_{\mathcal{N}}| \leq \frac{1}{|G| \cdot \|f^*_{\exp}\|^2_{\mathcal{N}}} \left| \mathbb{E}_{\boldsymbol{X} \sim \mathcal{N}} \sum_{g, g' \in G} f^*(\boldsymbol{B}^\top \rho(g) \boldsymbol{X}) f^*((\boldsymbol{B}')^\top \rho(g') \boldsymbol{X}) \right| \tag{118}$$

$$\leq \frac{1}{|G| \cdot \|f^*_{\exp}\|^2_{\mathcal{N}}} \sum_{g, g' \in G} \left| \mathbb{E}_{\boldsymbol{X} \sim \mathcal{N}} f^*(\boldsymbol{B}^\top \rho(g) \boldsymbol{X}) f^*((\boldsymbol{B}')^\top \rho(g') \boldsymbol{X}) \right| \

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

 $[\sum_i \boldsymbol{x}_i, \ldots, \sum_i \boldsymbol{x}_i^d]$. Another way to parameterize each $\boldsymbol{x}_j$ is as $\boldsymbol{x}_j = e^{i\theta}$ where $\theta$ is distributed $U[0, 2\pi]$, for all $j \in [n]$. Thus $\boldsymbol{x}_j^t$ can also be parameterized by $e^{i\theta}$ with the exact same distribution of $\theta \sim U[0, 2\pi]$, for all $j \in [n], t \in [d]$. The distribution of $[\sum_i \boldsymbol{x}_i, \ldots, \sum_i \boldsymbol{x}_i^d]$ is therefore a product distribution $(\text{Law}(\sum_{i=1}^{n} Z_i))^d$ where $Z_i \sim U[\mathbb{C} \cap \{|z| = 1\}]$. These generators and input distributions come from the classical text of Macdonald (1979). Here, as long as $r \leq n$, $r$ only depends on $d$ and not $n$. Plugging this $r$ into Theorem 38 gives straightforward bounds that are independent of $n$-the original number of variables, but does not give meaningful separation in Corollary 39.

As a reminder, we also have an example in the main text for the sign group (Lim et al., 2022).

## H   EXPERIMENTAL DETAILS

GNN experiments were performed using the Pytorch Geometric library for implementing geometric architectures (Fey & Lenssen, 2019). The hard functions were drawn from $\mathcal{H}_{ER,n}$ (see Eq. (7)) where subset $S \subseteq [n]$ was drawn uniformly from all the possible subsets of size $\lfloor n/2 \rfloor$ and $b$ was either 0 or 1 with equal probability. For our experiments, we set $n = 15$ and trained on $n^2 = 225$ datapoints. The overparameterized GNN used during training consisted of 3 layers of graph convolution followed by a node aggregation average pooling layer and a two layer ReLU MLP with width 64. The graph convolution layers used 32 channels.

For the CNN experiments, we constructed the hard functions in Eq. (17) setting $n = 50$ and $k = 10$. Here, matrices $B \in \mathcal{B}$ were drawn randomly from all possible $n \times 2$ orthogonal matrices by taking a QR decomposition of a random $n \times 2$ matrix with Gaussian i.i.d. elements. For sake of convenience, we did not divide by the norm of the function. The CNN architecture implemented consisted of three

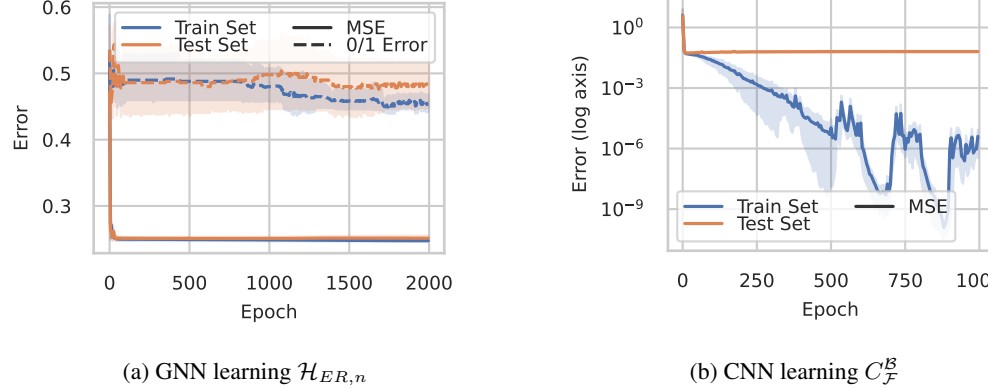

(a) GNN learning $\mathcal{H}_{ER,n}$          (b) CNN learning $C_{\mathcal{F}}^{\mathcal{B}}$

Figure 3: Replication of experiments as in Fig. 1, except here, we consider a minimal architecture consisting of a single layer of graph or cyclic convolution followed by a single hidden layer MLP. This is the minimal number of layers needed to learn the desired function classes for the architectures considered. For the CNN plot, the jumps in the train set MSE are due to perturbations in the loss at very low values near computer precision.

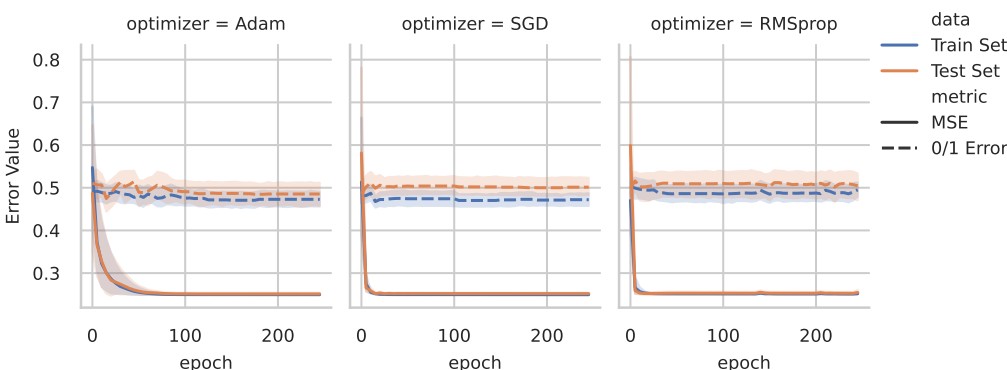

Figure 4: Replication of experiments in Fig. 1a with different optimizers show that the performance of the GNN is virtually the same across the various optimizers. Performance is averaged over 10 runs. For each run, the learning rate is chosen by perturbing the default learning rate by a random multiplicative factor in the range $[0.1, 10]$.

layers of fully parameterized cyclic convolution each with 100 output channels. This was followed by a global average pooling layer and and a two layer ReLU MLP of width 100. The network was given $10n = 500$ training samples and was trained with the Adam optimizer with batch size 32.

Experiments in the main text considered architectures which were overparameterized in terms of the number of layers with respect to the target function. For sake of completeness, we include in Fig. 3 results for merely sufficiently parameterized networks which consist of a single convolution layer, pooling and single hidden layer MLP. The results in Fig. 3 are consistent with those observed in Fig. 1. Additionally, to further confirm that these results are robust to hyperparameter choices for the optimizer, we repeated the GNN learning experiments with the choice of optimizer varying between Adam, SGD, and RMSprop (Paszke et al., 2019). Fig. 4 shows the results for these different optimizers. For each optimizer, 10 simulations were performed where for each simulation, the learning rate was chosen by multiplying the default learning rate by a random number in the range $[0.1, 10]$.

All experiments were run using Pytorch on a single GPU (Paszke et al., 2019). Default initialization schemes were used for the initial network weights. In our experiments, we used the Adam optimizer and tuned the learning rate in the range $[0.0001, 0.003]$. For CNN experiments, to increase stability

of training in later stages, we added a scheduler that divided the learning rate by two every 200 epochs. Plots are created by combining and averaging five random realizations of each experiment.