# OpenReview forum: "On the hardness of learning under symmetries"
_ICLR.cc/2024/Conference — ICLR 2024 spotlight_

### Official Review · Reviewer_kFXh · 2023-10-27

**Soundness:** 4 excellent
**Presentation:** 4 excellent
**Contribution:** 4 excellent
**Rating:** 8
**Confidence:** 3

**Summary:**

The authors study the computational hardness of learning equivariant networks using gradient descent.  They show that enforcing symmetries like permutation invariance does not make learning any substantially easier, and that their hardness results hold even for shallow 1-layer GNNs and CNNs.  They provide statistical query (SQ) lower bounds that scale exponentially with feature dimensions for various architectures.  Additionally, the authors provide an efficient non-gradient based algorithm for learning sparse invariant polynomials, separating SQ and correlational SQ complexity.  Lastly, they perform numerous experiments to verify their results.

**Strengths:**

Originality:

The authors prove numerous new results on the sample complexity of learning in neural networks, and provide ample empirical support for their work.

Quality/clarity:

The authors sketch their proofs using careful, clear technical arguments.  Additionally, their experiments are simple, but clear demonstrations of the practical difficulty of learning networks within the families they authors study.

Significance:

The author's work significantly advances progress on the hardness of learning symmetric networks, opening the door to clear avenues of future, follow-up work.

**Weaknesses:**

I would've liked a _slightly_ more thorough empirical treatment, if only to make sure that the failure to learn was not due to poor hyperparameter choices / poor initialization etc.

**Questions:**

Could the authors comment more on the applicability of "worst-case" reasoning re: the likelihood of these function classes to well-describe nature?  It seems plausible that the worst case could be significantly harder than the typical case for problems that we care about.  In practice, these sorts of hardness results don't seem to impact practitioner's usage of these model classes much at all!

---

> ### Author Response · Authors · 2023-11-16
>
> We thank the reviewer for their positive feedback and insightful comments. We respond to specific points below.
>
> > I would've liked a slightly more thorough empirical treatment, if only to make sure that the failure to learn was not due to poor hyperparameter choices / poor initialization etc.
>
> In our experiments, we found little to no difference in the results based on the choice of hyperparameters. For example, for the varying levels of overparameterization (e.g. more layers) and various learning rates that we tested, the results were essentially the same. Some of this is reported in the appendix, Section H (Figure 3). We have also included an additional plot showing that the performance of the GNN is consistent across three different optimizers (Adam, SGD, RMSprop) and different learning rates for those optimizers.
>
>
> > Could the authors comment more on the applicability of "worst-case" reasoning re: the likelihood of these function classes to well-describe nature? It seems plausible that the worst case could be significantly harder than the typical case for problems that we care about. In practice, these sorts of hardness results don't seem to impact practitioner's usage of these model classes much at all!
>
> The reviewer raises a good question that we hope forms the basis for future studies. Indeed, as pointed out by the reviewer and in the discussion section of our paper (Section 8), the functions we see in the real world are clearly learnable, and definitely not in the form of the rather adversarial ones studied here. Something has to be changed in the theoretical setting to tie it closer to practice; this is also the case for the existing line of work on SQ/CSQ lower bounds for fully-connected neural networks. To proceed formally, we see two paths for future study. First, worst-case settings such as SQ/CSQ can give too much flexibility to produce hard functions, which are often rather contrived. Some ways to make the setting more practical are to change the data input model (e.g. data that falls on a low dimensional manifold), or to introduce noise to the function class (e.g. weights are randomly chosen or subject to random perturbation). Second, we can reduce the size of the function class to rule out impractically adversarial learning tasks, e.g. by placing restrictions on the weights of the architectures to make them more tied to realistic functions. Such restrictions could also include bounds on the condition number, rank, etc. Promising work in these directions has been done for simple architectures, and we hope these ideas are expanded and studied in the symmetric setting as well [1-4].
>
> **References:**\
> [1] Chen, Sitan, et al. "Learning narrow one-hidden-layer relu networks." The Thirty Sixth Annual Conference on Learning Theory. PMLR, 2023.
>
> [2] Diakonikolas, Ilias, and Daniel M. Kane. "Efficiently Learning One-Hidden-Layer ReLU Networks via Schur Polynomials." arXiv preprint arXiv:2307.12840 (2023).
>
> [3] Daniely, Amit, Nathan Srebro, and Gal Vardi. "Most Neural Networks Are Almost Learnable." arXiv preprint arXiv:2305.16508 (2023).
>
> [4] Goel, Surbhi, et al. "Recurrent Convolutional Neural Networks Learn Succinct Learning Algorithms." Advances in Neural Information Processing Systems 35 (2022): 7328-7341.

---

### Official Review · Reviewer_Fgas · 2023-11-15

**Soundness:** 4 excellent
**Presentation:** 4 excellent
**Contribution:** 3 good
**Rating:** 6
**Confidence:** 5

**Summary:**

This work considers the problem of learning symmetric Neural Networks.  The
authors provide hardness results for Statistical Query (SQ) and Correlational
Statistical Query (CSQ) algorithms (and an NP-Hardness result for properly
learning Graph Neural Networks (GNNs)).  An example of a symmetric neural
network is a 2-layer GNN that maps an input graph $A$ to $g(f(A))$, where
$f:\{0, 1\}^{n \times n} \mapsto R^k$ first aggregates $k$ permutation
invariant features of the input graph $A$ and $g$ is a one-hidden layer MLP.

The first result is an SQ hardness result for two-layer GNNs showing that for
the above class of GNNs $\tau^2 2^{n^{\Omega(1)}}$ queries of tolerance $\tau$
are required.  The result follows by designing a 2-layer GNN where the
$i$-output of the first layer counts how many nodes have $i-1$ outgoing edges
and the second layer selects a subset of those counts and computes its parity.
By using properties of GNP graphs, the authors reduce the problem to the
well-known hard problem of learning parity functions over the uniform
distribution on the $n$-dimensional Boolean hypercube.

The second result considers GNNs that take as input a $n \times d$ feature
matrix $X$ and then compute $1_n^T \sigma(A(G) X W) a$, for an adjacency matrix
$A(G) \in \{0,1\}^n$,a weight $d \times 2 k$ matrix $W$ and a $2k$-dimensional
weight vector $a$.  They give a $d^k$ CSQ lower bound for this problem.  This
result follows from adapting the hard instances of the CSQ lower bound
construction of [2].

The third result shows that for CNNs (and more general frame-averaged networks)
of the form $f(X) = 1/|G| \sum_{g \in G} a^T \sigma (W^T g^{-1} X) 1_d$ where
$X$ is a $n \times d$ input matrix, $G$ is a group acting on $R^n$ (e.g., could
be cyclic shifts) either requires $2^{n^{\Omega(1)}}/ |G|^2$ queries or a query
with precision $|G| 2^{-n^{\Omega(1)}} + \sqrt{|G|} n^{-\Omega(k)}$.  The proof
of this results also adapts the construction of [2] For more general
frame-averaged networks, the authors use the techniques developed in [1] to
show a super-polynomial CSQ lower-bound (for any constant c either $n^{\log n}$
queries are needed or a query with accuracy $n^{-c}$).


[1] Surbhi Goel, Aravind Gollakota, Zhihan Jin, Sushrut Karmalkar, and Adam Klivans. Superpolynomial lower bounds for learning one-layer neural networks using gradient descent.
ICML 2020.

[2] Ilias Diakonikolas, Daniel M Kane, Vasilis Kontonis, and Nikos Zarifis. Algorithms and sq lower bounds for pac learning one-hidden-layer relu networks. COLT 2020.

**Strengths:**

1. The problem considered in this work is interesting and well-motivated. Most theoretical prior works on learning neural networks focused on fully connected shallow networks; investigating the learnability of popular and practically relevant classes of neural networks such as GNNs and CNNs (that have more restricted symmetric structure) is a natural
next step.

2. The paper provides hardness results for various classes of ``symmetric'' neural networks in the SQ and CSQ models that are general models of computation capturing, for example, stochastic gradient descent algorithms.

3. I found the paper to be well-organized and written. The authors clearly state what results of prior works they rely on to get their results.

**Weaknesses:**

1. The novelty of the technics and arguments used in the lower bounds provided in this work may be limited in the sense that most of the claimed results rely heavily on machinery developed in the prior works [1,2].

**Questions:**

1. See weaknesses.

2. While the authors clearly state which lemmas and proofs of the prior works they are using, I think a more detailed high-level explanation of the arguments and the differences from prior work should appear in the main body of the paper.

---

> ### Author Response · Authors · 2023-11-16
>
> We thank the reviewer for their insightful and detailed feedback, and respond to their specific questions and criticisms below.
>
>
> > The novelty of the techniques and arguments used in the lower bounds provided in this work may be limited in the sense that most of the claimed results rely heavily on machinery developed in the prior works [1,2].
>
> The reviewer is correct in noting that we based many of our own proofs on the rather useful and elegant proof techniques already in Diakonikolas et al., Goel et al., and others. Indeed, in our efforts to prove hardness results, we found that the techniques of these works were useful backbones to our proofs. Nonetheless, as shown in the appendix, extending these techniques to invariant settings often required careful and detailed changes or enhancements to the existing results. Finally, we should note one exception to our application of previous machinery: we could not prove the lower bound (Theorem 3) for Erdos-Renyi graph inputs via the Diakonikolas et al. or Goel et al. constructions. There, a more customized GNN had to be constructed to achieve orthogonality and SQ hardness, which may be of independent interest.
>
> > While the authors clearly state which lemmas and proofs of the prior works they are using, I think a more detailed high-level explanation of the arguments and the differences from prior work should appear in the main body of the paper.
>
> We thank the reviewer for pointing this out. Noting the space limits, we tried our best to provide high level explanations in the paper of the results describing our techniques early on in the introduction and adding proof sketches to the main text. Nonetheless, we recognize that following along with the arguments can be challenging at times and more details would without a doubt be helpful. For the updated draft, we have incorporated additional explanation in the text under the paragraph **Our contributions**.

---

### Official Review · Reviewer_aiE7 · 2023-11-21

**Soundness:** 4 excellent
**Presentation:** 3 good
**Contribution:** 3 good
**Rating:** 8
**Confidence:** 3

**Summary:**

This paper studies the hardness of learning certain two-layer or one-hidden-layer neural networks under symmetrized architecture/algorithmic designs on Gaussian inputs, via the Statistical Query (SQ) lower bound techniques. It provides several results characterizing the hardness of learning GNNs and CNNs via leveraging correlational SQ (CSQ) lower bounds for learning boolean functions and by connecting them with learning parity functions. It also discussed when CSQ lower bounds can be different than SQ lower bounds.

**Strengths:**

1. The question studied in this paper is closely related to a core question in understanding deep learning, that is: can deep learning benefit from symmetry-inspired algorithmic designs? In this sense I deem the question studied in the paper valuable and this paper's attempt to deal with it respectful.
2. The technical contribution of this paper, although still depended on some prior works, is novel enough to my understanding to be nontrivial. This paper constructed function classes that were not studied before to specifically deal with their problems, and proved hardness of learning these classes, which is a notable effort.
3. This paper covers both GNN and CNNs, and discussed the difference between CSQ and SQ in certain scenarios, which is good for completeness.

**Weaknesses:**

The weaknesses listed below are, in my opinion, secondary to the contributions of this paper.
The approach of this paper in studying the hardness of learning symmetry-enhanced neural networks has certain limitations. It cannot account for all neural architectures at once and requires specific construction whenever the problem formulation changes by a little bit. And the hard function classes, although are well designed for the proof, are not very intuitive in terms of broader impact to people who are outside of the learning theoretic community. Perhaps there could be more illustrative explanation for the intuition behinds the constructions and also its possible impacts outside pure theory.

**Questions:**

None

---

> ### Author Response · Authors · 2023-11-22
>
> We appreciate the positive feedback and thoughtful review.
>
> We acknowledge the reviewer's valid points regarding the limitations of our approach, specifically in terms of the lack of intuitiveness in the constructed function classes. As mentioned in other responses and in our discussion, we believe future work could focus on refining these classes or adapting the learning setting to enhance practicality. Additionally, we appreciate the observation that our approach may require specific constructions when the problem formulation undergoes changes. While we recognize this limitation, we aimed to be as broad as possible in our setting with this work and hope that our methods can serve as a basis for extensions in other settings.

---

### Author Response · Authors · 2023-11-16
**General comment about edits and changes**

We thank the reviewers for their insightful comments and feedback. As an overall point, we have made two changes to the draft in response to the reviewers’ feedback:
- Under the *Our contributions* subsection, we have added some further explanation of our technical arguments and a discussion of the differences with and motivation from prior work.
- In response to  reviewer kFXh’s feedback about the sensitivity of our experiments with respect to hyperparameters, we have added additional experiments in the supplemental material (Appendix H) with different optimization algorithms and hyperparameters to show that the empirical results are consistent across such choices.

---

### Meta-Review · Area_Chair_UzMY · 2023-12-21

**Metareview:**

After careful consideration of the reviewers' feedback and the authors' responses, the consensus is that the paper "Statistical Query Lower Bounds for Symmetric Neural Networks" is a valuable contribution to the field of machine learning theory. The paper addresses an important question regarding whether deep learning can benefit from symmetry-inspired algorithmic designs, focusing on the hardness of learning graph neural networks (GNNs) and convolutional neural networks (CNNs) using Statistical Query (SQ) lower bounds.

The strengths of the paper are clear. It presents novel technical contributions by constructing specific function classes to prove the hardness of learning these classes under symmetrized architecture/algorithmic designs. Moreover, the paper thoroughly explores both GNNs and CNNs and discusses the distinction between correlational SQ (CSQ) and SQ lower bounds in certain contexts, adding to the completeness of the work.

While some reviewers pointed out limitations, such as the approach's potential lack of generality and the non-intuitiveness of the hard function classes for those outside the learning theoretic community, these are outweighed by the paper's core contributions. Additionally, the authors have acknowledged these points and expressed their intent to address them in future work, which demonstrates a commitment to furthering the research in this area.

Questions about the applicability of "worst-case" reasoning were raised, and the authors provided a thoughtful response, suggesting future research directions that could bridge the gap between theory and practice. The authors also addressed concerns about empirical validation by including additional experiments in the supplemental material.

In light of the above, the paper is accepted for publication. The combination of theoretical insights, technical novelty, and a thoughtful discussion of the results' implications make this a good paper that will likely stimulate further research.

**Justification For Why Not Higher Score:**

A higher score was not warranted due to the reviewers' concerns about the limited novelty of the techniques and the heavy reliance on prior work. However, the authors have demonstrated that their work extends and adapts existing techniques in non-trivial ways to tackle the symmetrized settings, which justifies not assigning a lower score.

**Justification For Why Not Lower Score:**

The decision not to assign a lower score is supported by the paper's clear strengths, the authors' thorough responses to reviewer feedback, and the potential impact of the work on the field. The reviewers' confidence in their assessments, the comprehensive review process, and the alignment between the reviewers' evaluations further solidify this decision.

---

### Decision · Program_Chairs · 2024-01-16

Accept (spotlight)